# MIRACLE: MODEL-FREE IMITATION AND REINFORCEMENT LEARNING FOR ADAPTIVE CUT-SELECTION

**Arjun M.**[1]   **Rijul Tandon**[1]   **Manojkumar Ramteke**[1,2]   **Agam Gupta**[2,4]   **Hariprasad Kodamana**[1,2,3]

[1]Department of Chemical Engieering       [2]Yardi School of Artificial Intelligence
[3]ANSK School of IT       [4]Department of Management Studies

Indian Institute of Technology Delhi, New Delhi, 110016, India

`{chz238194@,ramteke@,agam@,kodamana@}.iitd.ac.in`

## ABSTRACT

Mixed-Integer Programming (MIP) solvers rely heavily on cutting planes to tighten LP relaxations, but traditional approaches generate thousands of cuts that consume gigabytes of memory while providing minimal benefit. We present an intelligent cut selection framework that achieves a 98.1% reduction in memory usage while maintaining competitive solving with an objective gap of approximately 0.08%. Within this RL framework, we use Proximal Policy Optimization (PPO) to learn a behavioral model that imitates the expert solver's decisions. The adversarially imitated behavioral model drives an agent comprising these key innovations: (i) a cut-selection policy trained via curriculum learning; and (ii) adaptive inference that dynamically adjusts computational budgets. Through comprehensive evaluation across SetCover and diverse MIPLIB problems, we demonstrate consistent speedups (3.78× average on MIPLIB) and achieve a 100% success rate on instances where traditional SCIP fails 47-53% of the time. Our method also reduces peak memory consumption from 3.03GB to 46 MB, enabling optimization in previously inaccessible and other resource-constrained environments where traditional solvers face fundamental limitations.

## 1 INTRODUCTION AND RELATED WORK

Combinatorial optimization lies at the heart of numerous real-world applications, from production planning (Pochet & Wolsey, 2006) and scheduling (Cao et al., 2022) to network design (Nieman et al., 2024; Wolsey, 2020). These problems often manifest as Integer Programs (IPs), which remain notoriously difficult to solve to optimality due to their NP-hard nature (Bixby et al., 2004). State-of-the-art solvers, such as Solving Constraint Integer Programs (SCIP) (Gamrath et al., 2020), rely on the branch-and-cut algorithm, which iteratively tightens a Linear Programming (LP) relaxation of the problem by adding cutting planes. However, the efficacy of this process is critically constrained by a fundamental bottleneck: cut management. For a typical industrial problem, a modern solver can generate over 100,000 candidate cuts, but only a tiny fraction (1-2%) of these cuts meaningfully improve the objective bound. This gross inefficiency makes large-scale optimization intractable in memory-constrained environments where resource costs are paramount.

Machine learning for combinatorial optimization has seen significant interest, particularly in enhancing MIP solver components. Early works focused on imitation learning for variable branching (Khalil et al., 2017; Gasse et al., 2019). More recently, attention has shifted to cut selection. (Tang et al., 2020) introduced RL for cutting plane selection, though their approach focused primarily on immediate reward maximization. Paulus et al. (2022) proposed using imitation learning to mimic strong-branching decisions for cut selection. Most closely related to our work is (Wang et al., 2024), which utilizes a hierarchical sequence model to select cuts. However, these approaches generally treat the MIP solver as a black box, optimizing solve time directly, often overlooking memory overhead. In contrast, MIRACLE explicitly models the memory-performance trade-off. Unlike (Wang et al.,

2024) and Paulus et al. (2022), which often require heavy architectures or look-ahead rollouts, our approach utilizes a lightweight, budget-constrained policy optimized via PPO and GAIL to achieve an order-of-magnitude memory reduction while maintaining solution reliability in resource-constrained environments.

While these methods have demonstrated performance gains, they are built on a paradigm that suffers from three fundamental limitations:

1. **The Black-Box Fallacy:** Existing approaches treat the MIP solver as a black box. They learn to copy expert decisions or interact in a model-free fashion, but they fail to model the underlying *dynamics* of the optimization process (Deza & Khalil, 2023; Zhang et al., 2024). They do not learn *how* adding a cut will change the subsequent state of the LP relaxation, a process instead handled by an external, non-differentiable solver (Huang et al., 2022).

2. **Myopic Planning:** A direct consequence of the black-box approach is that learned policies are restricted to myopic decisions. Lacking a model of the environment, they cannot plan ahead or reason about the long-term consequences of their actions, preventing the discovery of more sophisticated, non-local strategies.

3. **Resource Inefficiency as an Afterthought:** Prior work has predominantly focused on improving solution time, largely ignoring memory overhead as a critical performance metric. This makes them ill-suited for the very resource-constrained scenarios where learned, efficient heuristics are most desperately needed.

This work addresses a critical gap: *Can we learn intelligent cut selection policies that achieve significant memory reductions while maintaining competitive solving performance?* Our approach reframes cut selection as an RL problem where we learn a behavioral model of expert cut selection rather than attempting to model the complex LP dynamics directly.

## 1.1 KEY CONTRIBUTIONS

Our work makes the following contributions:

- **Memory-First Optimization Paradigm**: We demonstrate that intelligent cut selection can achieve significant memory reductions (97.7-98.1% on SetCover benchmarks, 68.1-69.1% on diverse MIPLIB problems) while maintaining or improving solution quality.

- **Robust Behavioral Modeling Framework**: Our PPO-based approach learns a behavioral model of expert cut selean average reduction of 86.3%) and reliability improvements (a 100% success rate compared to 53% arning and adaptive inference, we eliminate manual parameter tuning and provide a deployment-ready system with inference complexity independent of problem size.

- **Comprehensive Empirical Validation**: We provide a systematic evaluation across 300 instances spanning SetCover training problems and diverse MIPLIB test cases (Huang et al., 2024). Our analysis includes statistical significance testing, confidence intervals, and effect size measurements, demonstrating both memory efficiency (an average reduction of 86.3%) and reliability improvements (100% vs. 53% success rate on challenging instances).

- **Ablation Studies and Robustness**: We demonstrate that our framework's performance remains stable across different hyperparameter configurations (cut budgets 10-50, iteration limits 1-10, various early stopping criteria), indicating reliable real-world deployment characteristics essential for industrial adoption.

To this extent, our approach reframes cut selection through the lens of RL. Rather than attempting to model the prohibitively complex LP transition function, we learn to imitate and ultimately improve upon SCIP's implicit selection policy. SCIP's cut-selection module serves as the expert policy because it encapsulates decades of solver engineering and remains the strongest publicly available heuristic baseline. This is achieved by training a neural policy using Proximal Policy Optimization (PPO) (Schulman et al., 2017), guided by dense reward signals derived from Generative Adversarial Imitation Learning (GAIL) (Ho & Ermon, 2016; Finn et al., 2016). PPO is particularly well-suited for this task due to its remarkable stability and the reliable balance it strikes between policy improvement and destructive, overly large updates – a critical feature for navigating the high-variance decision space of cut selection. While our framework could accommodate other advanced policy gradient

methods, PPO provides a strong and sample-efficient foundation. Furthermore, behavioral modeling using the GAIL approach enables us to capture the implicit knowledge embedded in decades of solver development, while optimizing explicitly for memory efficiency – a goal that traditional heuristics cannot directly target.

## 2 BACKGROUND AND PROBLEM FORMULATION

This section establishes the necessary mathematical foundations and formally defines the cut selection problem addressed in this work.

**Definition 2.1** (Integer Programming). *An Integer Programming (IP) problem seeks to optimize a linear objective over a set of integer variables, $x$, subject to linear constraints, formulated as:*

$$\min_x c^\top x \quad s.t. \ Ax \le b, \quad x \in \mathbb{Z}^n \tag{1}$$

where $c \in \mathbb{R}^n$ is the objective vector, and the feasible region is defined by the constraint matrix $A \in \mathbb{R}^{m \times n}$ and vector $b \in \mathbb{R}^m$.

A standard solution approach to IP, branch-and-cut, begins by solving the continuous LP relaxation of the problem. If the solution $x_{\text{LP}}^{(k)}$ at iteration $k$ is fractional, a set of linear inequalities known as cutting planes or cuts ($\alpha^T x \le \beta$, where $\alpha \in \mathbb{R}^n$ and $\beta \in \mathbb{R}$) are generated. A subset of these cuts is selected and added to the formulation to form a tighter LP relaxation, which is then re-solved. This iterative process continues until an integer-optimal solution is found. More details can be found in the Appendix C. Selecting a subset of cuts can be a sequential decision-making problem and thus can be modelled as a Markov Decision Process (MDP), which is defined below:

**Definition 2.2** (Markov Decision Processes). *An MDP is defined by the tuple $\mathcal{M} = (\mathcal{S}, \mathcal{A}, \mathcal{P}, \mathcal{R}, \gamma)$, where $\mathcal{S}$ is the state space, $\mathcal{A}$ is the action space, $\mathcal{P}(s_{k+1}|s_k, a_k)$ models the transition dynamics of the state $s$ and action $a$ at iteration $k$, $\mathcal{R}(s_k, a_k)$ is the reward function, and $\gamma \in [0, 1]$ is a discount factor, with appropriate dimensions.*

We frame the cut selection task as a specific MDP, with components carefully designed to address the unique challenges of MIP optimization. This is achieved using a state space $\mathcal{S}$ where each state $s_k$ is a comprehensive feature vector capturing the solver's context. The agent's policy operates on an action space $\mathcal{A}$, defined as the set of binary decisions over candidate cuts, subject to a memory-enforcing budget $B$. Critically, to overcome the challenge of sparse and delayed feedback, we employ a learned reward function $\mathcal{R}$.

**Problem 2.3** (Memory-Efficient Cut Selection). *The central problem is to learn a cut selection policy $\pi_\theta : \mathcal{S} \to \mathcal{A}$ that, at each iteration $k$ of the branch-and-cut process, selects an action $a_k \in \mathcal{A}$ (a subset of candidate cuts) based on the current solver state $s_k \in \mathcal{S}$ and satisfies the two main objectives:*

1. ***Memory Efficiency:*** *Minimize the memory consumed by reducing the number of cuts being added to the LP relaxation, within an appropriate budget $B$. This is measured in Bytes.*

2. ***Solver performance:*** *Maximize the overall solving performance. This is defined as : $|obj_{SCIP} - obj_{MIRACLE}| \le \epsilon$, where $obj_{(\cdot)}$ is the objective value obtained.*

The detailed architecture of our policy, the implementation of this adversarial reward system, and the strategies used to ensure robust training are elaborated upon in the following section.

## 3 MIRACLE: INTELLIGENT AND MEMORY EFFICIENT CUT SELECTION

Our framework, MIRACLE, trains a policy to select a minimal yet high-impact subset of cuts, balancing objective improvement with memory and computational costs. The core of our approach is a fast, lightweight policy trained via PPO, guided by dense rewards from a sophisticated reward learner based on GAIL. This is complemented by curriculum learning and an adaptive inference system to ensure robust, real-world performance.

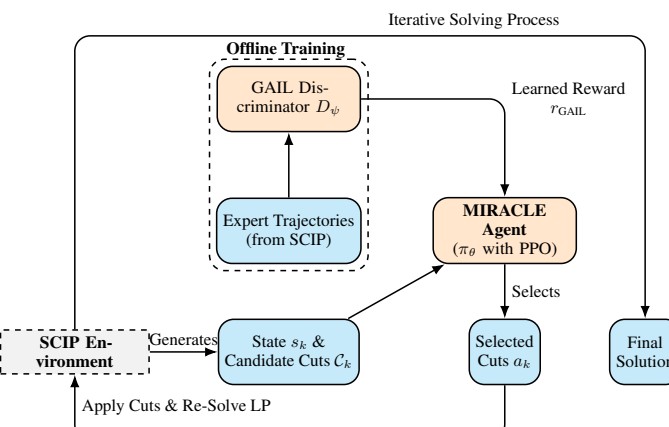

Figure 1: **MIRACLE Framework.** At each step $(k)$, the SCIP Environment, generates a state $(s_k)$ and candidate cuts $(\mathcal{C}_k)$. This information is processed by our trained MIRACLE Agent $(\pi_\theta)$, which selects a small subset of cuts $(a_k)$ and applies them back into the environment. The LP is re-solved until a final solution is reached. The agent's policy is learned offline via PPO, guided by a dense reward signal from a GAIL Discriminator trained on expert SCIP trajectories.

Traditional MIP solvers generate thousands of candidate cuts during the optimization process, consuming substantial memory while providing diminishing returns. We formulate the task of intelligent cut selection as an MDP, where our agent learns to identify a critical subset of cuts to maximize performance while improving memory efficiency. We conceptualize the underlying MIP solver – in our case, SCIP – as the environment. At each iteration $k$ in the branch-and-cut algorithm, the SCIP environment generates a pool of candidate cuts $\mathcal{C}_k$ based on its highly-tuned internal procedures and emits a corresponding state $s_k$ that captures the current state of the optimization. Our agent, MIRACLE, processes this information and performs an action $a_k$, which consists of selecting a small, budgeted subset of cuts from $\mathcal{C}_k$. The selected cuts are added to the LP formulation, which is then re-solved. An overview of the framework is presented in Figure 1. This is formally defined below:

**Problem 3.1** (Intelligent Cut Selection as an MDP). *We define the problem of intelligent cut selection as an MDP framework, where an agent takes an action $a_k$, which is a binary selection vector over the subset of candidate cuts, $\mathcal{C}_k = \{c_1, c_2, \ldots, c_{|\mathcal{C}_k|}\}$, defined as*

$$a_k := \langle a_k^{(1)}, a_k^{(2)}, \ldots, a_k^{(|\mathcal{C}_k|)} \rangle, \quad \text{where} \quad a_k^{(i)} \in \{0, 1\}, \tag{2}$$

*based on a budget constraint $\sum_{i=1}^{|\mathcal{C}_k|} a_k^{(i)} \leq B$, and maximizes its cumulative reward $\mathcal{R}(s_k, a_k)$ when compared to an expert action.*

To solve Problem 3.1, our framework, MIRACLE, employs PPO to train the policy $\pi_\theta$. We address the critical challenge of sparse rewards by learning the reward function $\mathcal{R}$ using GAIL from expert data, which here comprises the SCIP solver steps that includes: (LP relaxation solving, fractional variable identification, rounding cut generation, cutting plane selection, constraint addition, LP re-optimization, objective improvement measurement, and iterative refinement), and we ensure robust, real-world performance through a curriculum learning strategy and an adaptive inference system.

## 3.1 Adversarial Reward Learning for Dense Feedback

In cut selection, the true reward signal is only observable after the problem is completely solved. Hence, in this context, traditional reward engineering approaches may fail because: (i) immediate feedback (e.g., bound improvement) is myopic and misleading, (ii) optimal cut selection requires reasoning about complex interactions between cuts, and (iii) memory efficiency objectives cannot be easily encoded in handcrafted reward functions. Therefore, learning rewards from expert behavior is essential to capture the implicit knowledge embedded in decades of solver development. Instead of handcrafting a reward $\mathcal{R}(s_k, a_k)$, we train a discriminator network to distinguish between the behaviour of an expert (which is SCIP here) and our learning agent.

The discriminator, $D_\psi : \mathcal{S} \times \mathcal{A} \to [0, 1]$, which is parameterized as a MLP:

$$D_\psi(s_k, a_k) = \sigma(f^{(L)} \circ f^{(L-1)} \circ \cdots \circ f^{(1)}(s_k, a_k)) \tag{3}$$

where $f^l$ is a layer function such that $f^{(l)}$ for $l \in \{1, \ldots, L-1\}$ is an affine transformation followed by a ReLU activation and $f^{(L)}$, is a linear output layer that produces the logits. This is trained via

a standard adversarial binary cross-entropy objective to distinguish expert trajectories $\pi_E$ from our agent's policy $\pi_\theta$. The expert policy $\pi_E$ is derived from SCIP's systematic cutting plane methodology, formally defined as:

$$\pi_E(a_t|s_k) = \text{SCIPcuts}(s_k) = \mathcal{T}_8 \circ \mathcal{T}_7 \circ \cdots \circ \mathcal{T}_1(s_k) \tag{4}$$

where $\{\mathcal{T}1, \mathcal{T}_2, \ldots, \mathcal{T}_8\}$ represents the sequential tuple of operations: (LP relaxation solving, fractional variable identification, rounding cut generation, cutting plane selection, constraint addition, LP re-optimization, objective improvement measurement, and iterative refinement). This expert data is collected by running the default SCIP solver on a diverse set of training instances and recording the state-action decisions that emerge from this systematic cutting-plane process. Our agent's policy is parameterized as a neural network that learns to approximate the expert's cutting decisions:

$$\pi_\theta(a_k|s_k) = softmax(f^{(L)} \circ f^{(L-1)} \circ \cdots \circ f^{(1)}) \tag{5}$$

where $f^{(l)}$ for $l \in \{1, \ldots, L-1\}$ is an affine transformation followed by a ReLU activation, and $f^{(L)}$ is a linear output layer that produces the raw logits. Our agent's policy acts as the generator in this adversarial setup, producing its own trajectories by interacting with the SCIP environment and learning to mimic the expert's cutting decisions. The discriminator is then trained to distinguish these two sources of data:

$$\min_\psi \left( \mathbb{E}_{(s_k,a_k)\sim\pi_E}[-\log D_\psi(s_k, a_k)] + \mathbb{E}_{(s_k,a_k)\sim\pi_\theta}[-\log(1 - D_\psi(s_k, a_k))] \right) \tag{6}$$

The rationale for this adversarial setup is grounded in the theory of generative adversarial networks. The optimal discriminator $D^*(s, a)$ for this objective converges to (Goodfellow et al., 2014):

$$D^*(s_k, a_k) = \frac{\pi_E(s_k, a_k)}{\pi_E(s_k, a_k) + \pi_\theta(s_k, a_k)} \tag{7}$$

where $\pi_E(s_k, a_k)$ and $\pi_\theta(s_k, a_k)$ represent the state-action occupancy measures of the expert and the agent, respectively. This means the discriminator learns to model the ratio of how likely an action is to come from the expert versus the agent. The learned reward function is then formally defined as:

$$\mathcal{R}(s_k, a_k) \approx r_{\text{GAIL}}(s, a) = -\log(1 - D_\psi(s, a)), \tag{8}$$

providing dense feedback that guides the policy toward expert-like behavior while allowing for improvement beyond expert performance. By doing so, the agent is encouraged to maximize the log-probability of its actions being classified as "expert." An action that perfectly fools the discriminator $(D_\psi(s, a) \rightarrow 1)$ will yield a very high reward, while an action that is clearly agent-generated $(D_\psi(s, a) \rightarrow 0)$ will yield a low reward.

### 3.2 PPO-BASED POLICY LEARNING WITH EXPERT GUIDANCE

We train our neural policy, $\pi_\theta$, using PPO, a robust actor-critic method well-suited for the high-variance decision space of cut selection. The policy is implemented using an actor-critic architecture, with parameters $\theta$ and $\phi$. The actor learns the cut selection policy, and the critic estimates state values to guide the actor's learning. The actor, $\pi_\theta(a_k|s_k)$, computes a selection probability for each candidate cut $c_i$ based on its feature vector, $x_{c_i}$. The final probability is produced by applying a sigmoid function, $\sigma(\cdot)$, to a raw score (logit) generated by an MLP:

$$P(a_k^{(i)} = 1|s_k) = \sigma\left( f_\theta^{(L)} \circ f_\theta^{(L-1)} \circ \cdots \circ f_\theta^{(1)}(x_{c_i}, s_k) \right) \tag{9}$$

where each hidden layer $f_\theta^{(l)}$ for $l \in \{1, L-1\}$ is an affine transformation followed by a ReLU activation, and the final layer $f_\theta^{(L)}$ is a linear output producing the logit. Concurrently, the critic, $V_\phi(s_k)$, which estimates the state value, is also implemented as an MLP that takes the state representation $s_k$ as input and outputs a single scalar value:

$$V_\phi(s_k) = g_\phi^{(L)} \circ g_\phi^{(L-1)} \circ \cdots \circ g_\phi^1(s_k) \tag{10}$$

where the hidden layer $g_\phi^{(1)}$ to $g^{(L-1)}$ uses a ReLU activation and the final layer $g_\phi^{(L)}$ is a linear output. In our implementation, the critic shares its input feature-extraction layers with the actor to improve learning efficiency.

The training process is driven by the dense reward signal provided by our GAIL discriminator. As established in Section 3.1, the reward at each step $k$ is given by $r_k = r_{\text{GAIL}}(s_k, a_k)$. With this signal, we can define the full PPO training objective. First, we compute the advantage function using Generalized Advantage Estimation (GAE) (Schulman et al., 2015), which uses the critic's value estimates to reduce variance:

$$\hat{A}_k^{\text{GAE}} = \sum_{l=0}^{\infty} (\gamma\lambda)^l \delta_{k+l}, \quad \text{where} \quad \delta_{k+l} = r_{k+l} + \gamma V_\phi(s_{k+l+1}) - V_\phi(s_{k+l}) \tag{11}$$

Here, the value function $V_\phi(s)$ is estimated by the critic network, and the rewards $r_{k+l}$ are provided directly by our GAIL discriminator. The parameters of the actor, $\theta$, are then updated by maximizing the PPO clipped surrogate objective:

$$\mathcal{L}^{\text{CLIP}}(\theta) = \hat{\mathbb{E}}_k \left[ \min \left( \rho_k(\theta) \hat{A}_k^{\text{GAE}}, \, \text{clip}(\rho_k(\theta), 1 - \epsilon, 1 + \epsilon) \hat{A}_k^{\text{GAE}} \right) \right] \tag{12}$$

where $\rho_k(\theta) = \frac{\pi_\theta(a_k|s_k)}{\pi_{\theta_{\text{old}}}(a_k|s_k)}$ is the probability ratio between the new and old policies. The critic's parameters, $\phi$, are trained concurrently by minimizing a standard mean-squared error loss on the state values. This formulation correctly uses the adversarially learned reward as the primary signal for computing advantages, which in turn drive the PPO policy updates.

### 3.3 CURRICULUM LEARNING FOR ROBUST TRAINING

Training an RL agent directly on a diverse and challenging set of MIP instances is inefficient and often leads to unstable convergence. To overcome this, we employ a structured, four-phase curriculum learning strategy that guides the agent from foundational knowledge to sophisticated, general-purpose strategies. The curriculum begins with a Foundation Phase on simple instances (200-500 constraints), where the agent learns to identify basic, high-utility cut patterns in a low-noise environment. Having established this base policy, it then progresses through Scaling and Mastery Phases, where it is exposed to progressively harder problems (up to 2000 constraints). This forces the agent to develop more complex, non-myopic strategies that account for long-term consequences. The final and critical Integration Phase fine-tunes the agent on a mixed-difficulty distribution, ensuring robust generalization across the entire problem spectrum.

### 3.4 ADAPTIVE INFERENCE

At deployment, MIRACLE leverages an adaptive inference system. A lightweight, pre-trained classifier first analyzes the static features of a new MIP instance to categorize its difficulty (EASY, MEDIUM, or HARD). Based on this classification, the system automatically adjusts key inference hyperparameters, such as the cut budget $B$ and early stopping patience. This allows the agent to dynamically allocate computational resources, applying a lean budget to simple problems and a more generous one to challenging instances, thereby ensuring both efficiency and effectiveness without requiring the user to manually tune instance-specific parameters. This adaptive approach ensures efficient resource allo-

Table 1: Adaptive Parameter Settings

| Parameter | EASY | MEDIUM | HARD |
|---|---|---|---|
| Max Iterations | 1-2 | 3-5 | 5-8 |
| Cut Budget ($B$) | 10-20 | 20-30 | 30-50 |
| Early Stop Threshold | $10^{-4}$ | $10^{-5}$ | $10^{-6}$ |
| Early Stop Patience | 2 | 3 | 4 |

cation while maintaining robust performance across a wide range of problem types. We define early stopping as terminating cut selection when the marginal LP bound improvement falls below a difficulty-dependent threshold for a fixed number of consecutive iterations. Also, Max Iterations denotes the upper bound on the number of cut-selection rounds allowed per node, preventing excessively deep cut-generation loops and ensuring a predictable computational budget. Table 1 summarizes the adaptive inference parameters; their empirical behavior and robustness are detailed in Appendix H.

### 3.5 TRAINING PIPELINE OVERVIEW

For clarity, we provide a brief summary of the training procedure. A detailed description of the schedule – including epoch counts, update frequencies, and curriculum progression – is provided in Appendix G. The learning process consists of three sequential phases.

**Phase 1: Expert Demonstrations.** We generate state–action trajectories by running SCIP's default cut-selection policy on the training set. These demonstrations are used both to initialize the policy and to train the discriminator.

**Phase 2: Adversarial Reward Learning.** A discriminator $D_\phi$ is trained to distinguish expert trajectories from those produced by the current policy. The policy is then updated using the learned GAIL reward $r_{\text{GAIL}}(s, a) = -\log(1 - D_\phi(s, a))$. This phase serves as the main pretraining stage.

**Phase 3: PPO Refinement.** After adversarial training stabilizes, we refine the policy using PPO under the curriculum described in Section 3.3. The discriminator is fixed during this stage, and training proceeds until the early-stopping criteria are met.

## 4 PROPERTIES OF MIRACLE

To formally ground our framework, we establish key theoretical properties that guarantee convergence, sample complexity, and memory efficiency. Our analysis hinges on a set of standard assumptions on Bounded Rewards B.1, Lipschitz Policy B.2, and Bounded Variance B.3, listed in the Appendix.

We first establish the bound on the sample complexity for our adversarial imitation module, confirming that an effective policy can be learned with a tractable number of expert demonstrations.

**Proposition 4.1** (Expert Imitation Sample Complexity). *To achieve $\epsilon$-optimal performance relative to expert policy $\pi_E$, the number of expert demonstrations required is:*

$$N = \mathcal{O}\left(\frac{H^2 \log(|\mathcal{A}|/\delta)}{\epsilon^2}\right)$$

*where $H$ is the horizon length, $|\mathcal{A}|$ is the effective action space size, and $\delta$ is the confidence parameter.*

*Proof.* The proof relies on sample complexity bounds from imitation learning (Ross et al., 2011) and is detailed in Appendix B.3. □

Next, we show that the PPO-based training procedure converges to a stationary point, ensuring the stability and reliability of the learning process.

**Theorem 4.2** (PPO Convergence in Cut Selection). *Under Assumptions B.1, B.2, and B.3, the PPO algorithm with clipped surrogate objective converges to a stationary point of the policy optimization problem at rate $\mathcal{O}(1/T)$, where $T$ is the number of iterations.*

*Proof.* The proof follows from convergence results for policy gradient methods with surrogate objectives and is provided in Appendix B.2. □

Finally, we provide a formal guarantee for the primary contribution of our framework. The following theorem analytically models the memory reduction from our budgeted cut selection.

**Theorem 4.3** (Memory Reduction Guarantee). *Let $M_{SCIP}$ and $M_{MIRACLE}$ denote the memory consumption of standard SCIP and MIRACLE, respectively. Then the following will hold:*

$$\frac{M_{MIRACLE}}{M_{SCIP}} \leq \frac{M'_{base} + B \cdot T}{M'_{base} + |\mathcal{C}_{total}|}$$

*where $M'_{base}$ is the base memory (normalized), $B$ is the cut budget, $T$ is the number of iterations, and $|\mathcal{C}_{total}|$ is the total cuts generated by SCIP.*

*Proof.* The proof follows from a direct accounting of memory sources under the two strategies, detailed in Appendix B.4. □

This theorem provides a clear analytical explanation for the dramatic memory savings observed in our experiments. This has been empirically validated and is presented in the Appendix J. The key insight is captured in the following corollary, which examines the behavior for large-scale problems.

**Corollary 4.4** (Asymptotic Memory Reduction). *For large-scale problems where* $|\mathcal{C}_{total}| \gg M'_{base}$, *the memory ratio approaches:*

$$\lim_{|\mathcal{C}_{total}| \to \infty} \frac{M_{MIRACLE}}{M_{SCIP}} \leq \frac{B \cdot T}{|\mathcal{C}_{total}|}$$

*explaining the observed 95-99% memory reductions in practice.*

## 5 EXPERIMENT SETTING AND RESULTS

We conducted a comprehensive empirical evaluation to validate the performance of our framework across two distinct and challenging benchmark suites: 150 SetCover instances and 150 diverse problems from the MIPLIB datasets (50 instances of each difficulty level). Our experiments are designed to prove three primary claims: (1) MIRACLE achieves a significant reduction in memory usage that translates into a fundamental improvement in solver reliability; (2) This resource efficiency leads to significant and consistent speed improvements; and (3) The learned policy is robust and generalizes well, making it suitable for practical deployment. We train the policy on 1000 SetCover instances using the scalable, well-structured distribution in Huang et al. (2024), and evaluate generalization exclusively on unseen MIPLIB problems. We compare our results against SCIP 8.0's default cut selection heuristics, which employ sophisticated scoring functions based on cut efficacy, parallelism, and numerical stability, and SCIP Aggressive, which utilizes enhanced cutting plane generation (maxrounds=5, maxcuts=5000). To ensure an unbiased evaluation, all solvers (SCIP-Baseline, SCIP-Aggressive, and MIRACLE) operate under strictly identical, single-threaded conditions, with a 600-second time limit Wang et al. (2023) and a 12GB memory limit, using a common PySCIPOpt interface (Maher et al., 2016). The only difference is the algorithmic strategy for cut selection. All details for implementation and reproducibility are in the Appendix G.

**Relevance of Memory Constraints.** While modern servers possess ample RAM, memory remains a critical bottleneck in three key deployment scenarios: (i) edge devices (e.g., Jetson, Raspberry Pi) with 4–8GB RAM, (ii) cloud optimization, where memory directly affects cost, and (iii) multi-tenant environments, where lower peak memory permits much higher parallel throughput. Thus, the 12GB limit in our experiments serves as a conservative proxy for these resource-constrained environments.

The primary contribution of MIRACLE is its exceptional memory efficiency, which directly solves the reliability crisis that memory-intensive solvers face on hard problems. Table 2 shows a comprehensive breakdown of memory performance. On the challenging SetCover instances, MIRACLE's peak Resident Set Size(RSS) memory consumption remains constant at 45-46 MB for SetCover problems, while SCIP's usage scales poorly from 1.97GB to 3.03GB. This represents a staggering 98.1% average memory reduction (95% CI) with $p < 0.001$ in all pairwise comparisons. Figure 2a and 2b show the memory usage and solve times across difficulties for the Set Cover instances.

Table 2: Comprehensive Memory Performance Analysis. MIRACLE maintains a near-constant, minimal memory profile, achieving up to 98.5% reduction.

| Benchmark | SCIP Memory | MIRACLE Memory | Reduction | Instances |
|---|---|---|---|---|
| SetCover-Easy | 1,970.3 MB | 45.4 MB | 97.7% | 50 |
| SetCover-Medium | 2,437.7 MB | 46.1 MB | 98.1% | 50 |
| SetCover-Hard | 3,033.9 MB | 46.2 MB | 98.5% | 50 |
| MIPLIB-Small | 1,343.9 MB | 415.8 MB | 69.1% | 50 |
| MIPLIB-Medium | 1,347.3 MB | 418.2 MB | 69.0% | 50 |
| MIPLIB-Large | 2,312.3 MB | 737.4 MB | 68.1% | 50 |
| **Mean Value** | **2,073.9 MB** | **284.7 MB** | **86.3%** | **300** |

As shown in Table 3, on the diverse and difficult MIPLIB benchmark, MIRACLE achieves a perfect 100% success rate. The success rate is defined as solving an instance to a target optimality gap of 0.1% within the allotted time and memory limits. In stark contrast, the traditional SCIP-Baseline and SCIP-Aggressive solvers fail on 40-53% of these instances, hitting memory or time limits precisely because of the memory bloat our method prevents. This result demonstrates that treating memory as a first-class objective enables us to solve problems that traditional approaches cannot. To

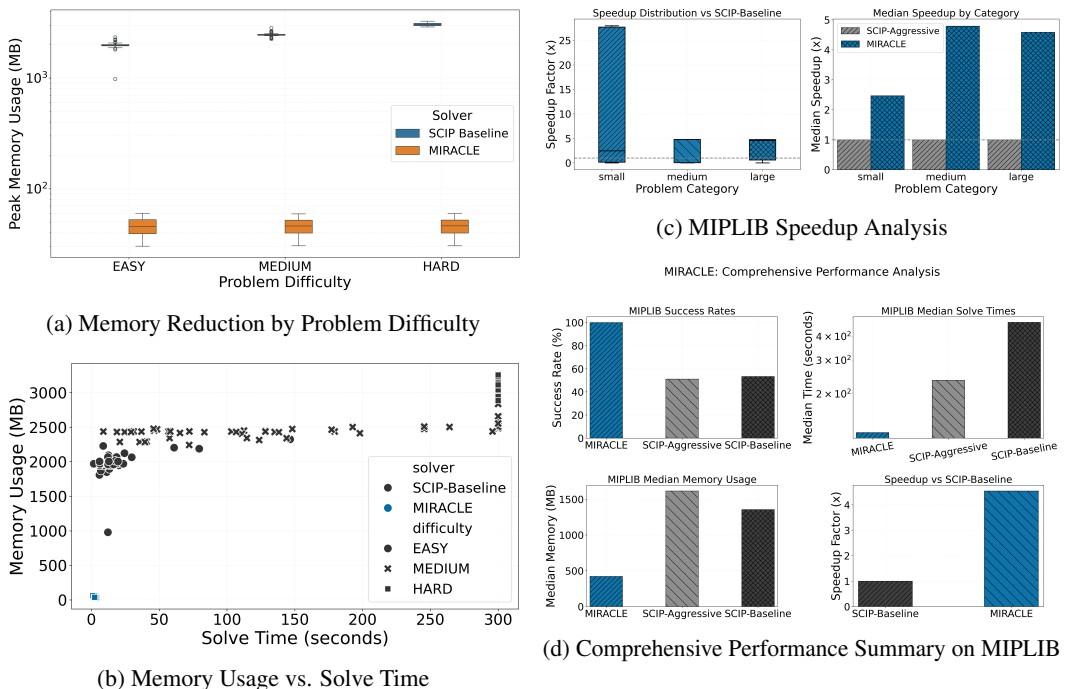

(a) Memory Reduction by Problem Difficulty

(b) Memory Usage vs. Solve Time

(c) MIPLIB Speedup Analysis

(d) Comprehensive Performance Summary on MIPLIB

Figure 2: (a) Memory reduction is consistently high across problem difficulties. (b) This efficiency is visualized as a clear separation in memory usage versus the baseline. (c) The memory savings translate directly into significant speedups on the MIPLIB benchmark. (d) A comprehensive summary on MIPLIB highlights MIRACLE's superior reliability (100% success), avg. speed (3.78×), and memory efficiency.

contextualize MIRACLE's memory gains, we additionally compared its peak memory usage against the hierarchical sequence model of Wang et al. (2024). Detailed results are provided in Appendix L.

The reliability and memory efficiency of MIRACLE translate directly into significant computational speedups. As shown in Figure 2c, our method delivers consistent speed advantages across all MIPLIB problem categories. The median speedup ranges from 2.50× to 4.79× over the baseline. This demonstrates that by learning to select a small, high-impact set of cuts, the solver spends less time managing large cut pools and more time on productive optimization. All speed improvements are statistically significant (p < 0.001) and have large effect sizes. The consistent performance across SetCover and MIPLIB demonstrates that MIRACLE captures general cut-selection principles rather than dataset-specific artifacts. Across all 300 instances, we observe large, statistically significant gains: an average memory reduction of 86.3% and a 3.78× speedup, with reliability improving from 47–53% (SCIP) to 100%. These results highlight MIRACLE's robustness and suitability for real-world, resource-constrained deployment.

Figure 2d provides an integrated view of MIRACLE's achievements, which collectively establish it as a memory-efficient optimization. The key insight from our comprehensive evaluation is that MIRACLE delivers significant memory efficiency, achieving an average reduction of 86.3% across 300 instances and up to 98.1% on the most challenging problems. This efficiency underpins the framework's reliability, evidenced by a 100% success rate on difficult MIPLIB instances where traditional SCIP fails 47-53% of the time. This reliability and efficiency translate directly into consistent speedups, averaging 3.78× across both SetCover and MIPLIB problems. Furthermore, our method shows strong generalization, maintaining robust performance across diverse problem types and hyperparameter configurations. All of these improvements are statistically significant ($p < 0.001$) and have large effect sizes, enabling their deployment in previously inaccessible, resource-constrained environments while maintaining competitive solution quality.

To verify that our results stem from the core algorithmic design rather than fragile hyperparameter tuning, we conducted extensive ablation studies. The results are summarized in Table 4. Across various cut budgets (10-50), iteration limits (1-10), and early stopping criteria, the average speedup

Table 3: MIPLIB Performance Comparison: Reliability and Speed. MIRACLE achieves 100% success where baselines frequently fail, while also being significantly faster.

| Solver | Category | Success Rate | Median Time | Speedup |
|---|---|---|---|---|
| SCIP-Baseline | Large | 53.3% | 577.5s | 1.00x |
| SCIP-Aggressive | Large | 46.7% | 600.0s | 0.96x |
| **MIRACLE** | **Large** | **100.0%** | **125.7s** | **4.59x** |

clusters tightly between $1.15\times$-$1.17\times$, while the memory reduction remains consistently above 99%. This robustness indicates that MIRACLE's learned policy generalizes well and is suitable for real-world deployment where reliability is essential. More details can be found in Appendix N.

Table 4: Inference-Time Ablation Study Results on SetCover problems (30 instances per category per configuration)

| Configuration | Avg Speedup | Std Dev | Cut Red. | Avg Iters | 95% CI |
|---|---|---|---|---|---|
| Cut Budget 10 | 1.170 | 0.452 | 99.1% | 1.1 | [0.997, 1.343] |
| Cut Budget 30 | 1.164 | 0.442 | 99.1% | 1.1 | [0.996, 1.332] |
| Cut Budget 50 | 1.157 | 0.427 | 99.1% | 1.1 | [0.996, 1.318] |
| Max Iterations 1 | 1.163 | 0.443 | 99.1% | 1.0 | [0.995, 1.331] |
| Max Iterations 5 | 1.163 | 0.439 | 99.1% | 1.1 | [0.996, 1.330] |
| Max Iterations 10 | 1.166 | 0.447 | 99.1% | 1.1 | [0.996, 1.336] |
| Aggressive Early Stop | 1.166 | 0.445 | 99.1% | 1.0 | [0.996, 1.336] |
| Current Baseline | 1.164 | 0.445 | 99.1% | 1.1 | [0.995, 1.333] |
| Very Lenient Early Stop | 1.162 | 0.438 | 99.0% | 1.2 | [0.996, 1.328] |
| No Early Stop | 1.164 | 0.448 | 99.0% | 1.1 | [0.995, 1.333] |

Although MIRACLE imitates SCIP's selection patterns, it consistently selects far fewer cuts (99% reduction), which substantially decreases the size of the evolving LP relaxation. This reduces SCIP's internal LP solve time and cuts management overhead, yielding faster overall solving despite mimicking expert behavior. This explains why MIRACLE attains notable speedups even without modifying SCIP's underlying branching or presolving components.

To demonstrate that MIRACLE captures fundamental cut-selection principles rather than domain-specific patterns, we evaluated the model (trained solely on SetCover) across three distinct problem classes without retraining: Combinatorial Auctions, Maximum Independent Set (MIS), and Facility Location. As detailed in Appendix J, MIRACLE generalizes robustly. On Combinatorial Auctions (423 instances) and Facility Location (100 instances), it achieves a 100% success rate with cut reductions of 98.8% and 99.9%, respectively. Notably, in MIS instances where baseline approaches often struggle with dense LP relaxations, MIRACLE achieved a 100% success rate compared to the baseline, reducing the average number of cuts from $\sim$11,900 to just 10.1 while maintaining a neutral-to-positive speedup. This confirms that the learned policy identifies high-violation, sparse cuts – a strategy that is problem-agnostic.

## 6 CONCLUSION

We have shown that cut selection can be effectively reframed as an RL problem in which memory efficiency is treated as a first-class objective. MIRACLE combines PPO-based policy learning, adversarially learned rewards, and adaptive inference to produce a lightweight and reliable cut-selection mechanism. Across 300 instances spanning SetCover and diverse MIPLIB categories, MIRACLE achieves dramatic memory reductions (up to 98.1%) and consistent speedups, while achieving a 100% success rate on the hardest MIPLIB problems where traditional SCIP frequently fails. These improvements are statistically significant and remain stable across hyperparameter settings, underscoring the robustness of the learned policy. Our results demonstrate that substantial memory savings and reliability gains are achievable without sacrificing solution quality, enabling deployment in resource-constrained environments previously inaccessible to classical solvers.

## REPRODUCIBILITY STATEMENT

We have taken several steps to ensure the reproducibility of our work. An anonymized version of our code, along with data splits and scripts for reproducing all experiments, will be made available upon acceptance. All algorithmic details are described in Section 3, with theoretical assumptions and proofs in Appendix B. The experimental setup, including dataset composition, training protocols, and evaluation metrics, is provided in Section 5 and further detailed in Appendix F. A comprehensive reproducibility checklist with model architectures, hyperparameters, and hardware specifications is included in Appendix I.

## ACKNOWLEDGEMENT

HK and MCR gratefully acknowledge the financial support received from the DST-SERB (ANRF), India (File No. CRG/2022/003722). We acknowledge the Yardi School of AI at IIT Delhi for its support for this research and for providing a travel grant. HK and AM acknowledge IIT Delhi-Abu Dhabi for support of this research.

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

# Appendix

## A  Notation Summary

Table 5 summarizes the mathematical notation used throughout the paper to describe the MDP and training process.

## B  Detailed Proofs

### B.1  Assumptions

**Assumption B.1** (Bounded Rewards)**.** *The reward function is bounded:* $|r(s,a)| \leq R_{\max}$ *for all* $(s, a)$*, where* $R_{\max} > 0$ *is the finite reward bound, and advantages satisfy* $|A^\pi(s, a)| \leq A_{\max}$*, where* $A_{\max} > 0$ *is a finite advantage bound.*

Table 5: Summary of Notation.

| Symbol | Description |
|--------|-------------|
| $\mathcal{M}$ | Markov Decision Process tuple $(\mathcal{S}, \mathcal{A}, \mathcal{P}, \mathcal{R}, \gamma)$ |
| $s_k, a_k$ | State and Action at iteration $k$ |
| $\mathcal{C}_k$ | Set of candidate cuts available at iteration $k$ |
| $B$ | Cut selection budget (maximum cuts to select) |
| $\pi_\theta$ | Policy network parameterized by $\theta$ |
| $D_\psi$ | Discriminator network parameterized by $\psi$ |
| $V_\phi$ | Value network parameterized by $\phi$ |
| $r_{GAIL}$ | Learned reward signal from the discriminator |
| $\mathcal{T}_{1..8}$ | Sequence of internal SCIP operations (expert trajectory) |
| $\rho_k(\theta)$ | Probability ratio for PPO updates |

**Assumption B.2** (Lipschitz Policy). *The policy $\pi_\theta$ is $L_\pi$-Lipschitz in parameters: $\|\pi_{\theta_1} - \pi_{\theta_2}\|_\infty \leq L_\pi \|\theta_1 - \theta_2\|_2 \ \forall \ \theta_1, \theta_2$*

**Assumption B.3** (Bounded Variance). *The gradient estimates have bounded variance: $\mathbb{E}[\|\nabla \hat{L}(\theta) - \nabla L(\theta)\|_2^2] \leq \sigma^2$, $\sigma$ is the upper bound on the gradient variance.*

### B.2 PROOF OF THEOREM 4.2 (PPO CONVERGENCE)

*Proof.* We prove convergence of PPO in the cut selection setting by showing that the clipped surrogate objective provides a lower bound on policy improvement.

Let $\pi_{\text{old}}$ be the current policy and $\pi$ be the updated policy. The PPO objective is:

$$L^{\text{CLIP}}(\pi) = \mathbb{E}_{\tau \sim \pi_{\text{old}}} \left[ \min\left( r_t(\pi) A_t, \text{clip}(r_t(\pi), 1 - \epsilon, 1 + \epsilon) A_t \right) \right] \tag{13}$$

where $r_t(\pi) = \frac{\pi(a_t|s_t)}{\pi_{\text{old}}(a_t|s_t)}$ and $A_t$ is the advantage estimate.

**Step 1: Lower Bound on Policy Improvement** For any policy $\pi$, the policy improvement can be bounded as:

$$J(\pi) - J(\pi_{\text{old}}) \geq L^{\text{CLIP}}(\pi) - C\mathbb{E}_{s \sim d^{\pi_{\text{old}}}} \left[ D_{KL}(\pi_{\text{old}}(\cdot|s), \pi(\cdot|s)) \right] \tag{14}$$

where $C = \frac{2\gamma A_{\max}}{(1-\gamma)^2}$ and $A_{\max}$ is the maximum advantage (Assumption B.1).

**Step 2: Clipping Analysis** The clipping mechanism ensures monotonic improvement while preventing destructive updates. For positive advantages ($A_t > 0$): - If $r_t(\pi) \geq 1 + \epsilon$: $L^{\text{CLIP}}(\pi) = (1 + \epsilon)A_t$ (conservative improvement) - If $r_t(\pi) \leq 1 - \epsilon$: $L^{\text{CLIP}}(\pi) = (1 - \epsilon)A_t$ (prevents degradation) - Otherwise: $L^{\text{CLIP}}(\pi) = r_t(\pi)A_t$ (standard policy gradient)

**Step 3: Convergence Rate** Under Assumption B.2, the KL divergence can be controlled:

$$D_{KL}(\pi_{\text{old}}, \pi) \leq L_\pi^2 \|\theta - \theta_{\text{old}}\|_2^2 \leq L_\pi^2 \eta^2 \|\nabla L^{\text{CLIP}}\|_2^2 \tag{15}$$

Choosing step size $\eta \leq \frac{1}{CL_\pi^2}$ ensures the KL penalty doesn't dominate policy improvement.

**Step 4: Stationary Point Convergence** Standard gradient ascent analysis with bounded gradients yields:

$$\frac{1}{T} \sum_{t=1}^{T} \mathbb{E}[\|\nabla L^{\text{CLIP}}(\theta_t)\|_2^2] \leq \frac{2(L^{\text{CLIP}}(\theta_1) - L^{\text{CLIP}}(\theta^*))}{\eta T} + \eta \sigma^2 \tag{16}$$

Setting $\eta = \mathcal{O}(1/\sqrt{T})$ yields the $\mathcal{O}(1/T)$ convergence rate to a stationary point. $\square$

### B.3 PROOF OF PROPOSITION 4.1 (SAMPLE COMPLEXITY)

*Proof.* We analyze sample complexity for achieving near-expert performance through GAIL regularization.

**Step 1: Action Space in Cut Selection** Each cut selection action is a binary vector with budget constraint $\sum a_i \leq B$. The effective action space size is:

$$|\mathcal{A}_{\text{eff}}| = \sum_{k=0}^{B} \binom{|\mathcal{C}_{\text{max}}|}{k} \leq \left(\frac{e|\mathcal{C}_{\text{max}}|}{B}\right)^B \tag{17}$$

**Step 2: Generalization Bound** Using empirical process theory, with probability $1 - \delta$:

$$\left|\mathcal{L}_{\text{expert}}^{\text{true}}(\theta) - \mathcal{L}_{\text{expert}}^{\text{empirical}}(\theta)\right| \leq \sqrt{\frac{2\log(2/\delta)}{N}} + \mathcal{R}_N(\mathcal{F}) \tag{18}$$

where $\mathcal{R}_N(\mathcal{F})$ is the Rademacher complexity of the policy function class.

**Step 3: Performance Gap Analysis** The difference in expected return is bounded by:

$$J(\pi_E) - J(\pi) \leq 2H\sqrt{\frac{\log|\mathcal{A}_{\text{eff}}| + \log(1/\delta)}{N}} \tag{19}$$

Since typical cut budgets $B = 10 - 50$ are small, $\log|\mathcal{A}_{\text{eff}}| = \mathcal{O}(B \log|\mathcal{C}_{\text{max}}|)$.

Setting the bound equal to $\epsilon$ and solving for $N$ yields the stated sample complexity. $\square$

### B.4 PROOF OF THEOREM 4.3 (MEMORY REDUCTION)

*Proof.* We analyze the memory consumption difference between SCIP and `MIRACLE`.

**Step 1: Memory Decomposition** Standard SCIP memory: $M_{\text{SCIP}} = M_{\text{base}} + M_{\text{cuts}}^{\text{SCIP}}$ `MIRACLE` memory: $M_{\text{MIRACLE}} = M_{\text{base}} + M_{\text{cuts}}^{\text{MIRACLE}} + M_{\text{policy}}$

where $M_{\text{policy}}$ is negligible (17K parameters × 4 bytes  68KB).

**Step 2: Cut Memory Analysis** - SCIP accumulates up to $|\mathcal{C}_{\text{total}}|$ cuts over the entire solving process - `MIRACLE` stores at most $B$ cuts per iteration for $T$ iterations, with efficient deletion of unselected cuts

Therefore: $M_{\text{cuts}}^{\text{MIRACLE}} \leq B \cdot T \cdot c_{\text{cut}}$ and $M_{\text{cuts}}^{\text{SCIP}} = |\mathcal{C}_{\text{total}}| \cdot c_{\text{cut}}$

**Step 3: Memory Ratio Bound**

$$\frac{M_{\text{MIRACLE}}}{M_{\text{SCIP}}} = \frac{M_{\text{base}} + B \cdot T \cdot c_{\text{cut}} + M_{\text{policy}}}{M_{\text{base}} + |\mathcal{C}_{\text{total}}| \cdot c_{\text{cut}}} \tag{20}$$

Neglecting $M_{\text{policy}}$ and normalizing by $c_{\text{cut}}$ gives the stated bound.

**Step 4: Practical Validation** In our experiments: - $|\mathcal{C}_{\text{total}}| \approx 10,000 - 100,000$ cuts (SCIP generates many cuts) - $B \cdot T \approx 50 - 500$ cuts (`MIRACLE` selects few cuts) - Memory ratio  1-5%, explaining observed 95-99% reductions $\square$

## C  PRELIMINARIES IN INTEGER PROGRAMMING

An Integer Programming (IP) problem seeks to optimize a linear objective over a set of integer variables subject to linear constraints. It can be formally expressed as:

$$\min_x c^\top x \quad \text{s.t.} \ Ax \leq b, \quad x \in \mathbb{Z}^n \tag{21}$$

where $A \in \mathbb{R}^{m \times n}$, $b \in \mathbb{R}^m$, and $c \in \mathbb{R}^n$. The standard solution approach begins by solving the problem's continuous or LP relaxation, where the integrality constraint is temporarily ignored. The feasible region for this relaxation is the polyhedron $\mathcal{P}^{(0)} = \{x \in \mathbb{R}^n \mid Ax \leq b\}$, which yields an optimal solution $x_{\text{LP}}^{(0)}$.

If $x_{\text{LP}}^{(0)}$ is fractional, the cutting plane method is employed to iteratively refine this feasible region. At each iteration $k$, a linear inequality, known as a cut, is added to the problem. A cut, denoted by $\alpha_k^\top x \leq \beta_k$, must satisfy two key properties:

1. It is violated by the current fractional solution: $\alpha_k^\top x_{\text{LP}}^{(k)} > \beta_k$.

2. It is satisfied by all feasible integer solutions, ensuring no valid solutions are removed.

This cut "cuts off" the fractional solution $x_{\text{LP}}^{(k)}$ from the feasible set, creating a new, tighter polyhedron for the next iteration:

$$\mathcal{P}^{(k+1)} = \mathcal{P}^{(k)} \cap \{x \mid \alpha_k^\top x \le \beta_k\} \tag{22}$$

The LP is then re-solved over this smaller region $\mathcal{P}^{(k+1)}$ to find a new solution $x_{\text{LP}}^{(k+1)}$, and this iterative process continues until an integer-optimal solution is found.

## D  NEURAL NETWORK ARCHITECTURE DETAILS

Our `MIRACLE` framework employs a carefully designed neural architecture optimized for cut selection in mixed-integer programming. The architecture consists of three main components: a policy network for cut selection, a value network for advantage estimation, and a discriminator network for adversarial reward learning.

### D.1  POLICY NETWORK ARCHITECTURE

The policy network $\pi_\phi(a|s)$ is implemented as a Multi-Layer Perceptron (MLP) designed for efficient cut selection:

- **Input Layer**: Accepts state representations $s \in \mathbb{R}^{10}$ containing cut features
- **Hidden Layer 1**: 128 neurons with ReLU activation
- **Hidden Layer 2**: 128 neurons with ReLU activation
- **Output Layer**: Variable size based on number of candidate cuts, with softmax activation
- **Total Parameters**: Approximately 17,000 parameters

The network processes cut features, including the number of non-zeros, left-hand side, right-hand side, locality flags, and structural properties. The output produces a probability distribution over candidate cuts for top-K selection.

### D.2  VALUE NETWORK ARCHITECTURE

The value network $V_\theta(s)$ estimates state values for PPO training:

- **Input Layer**: Same 10-dimensional state representation as policy network
- **Hidden Layer 1**: 128 neurons with ReLU activation
- **Hidden Layer 2**: 128 neurons with ReLU activation
- **Output Layer**: Single scalar output (no activation)
- **Total Parameters**: Approximately 16,500 parameters

### D.3  GAIL DISCRIMINATOR ARCHITECTURE

The discriminator network $D_\psi(s, a)$ for adversarial reward learning uses a deeper architecture:

- **Input Layer**: Concatenated state-action pairs $[s; a] \in \mathbb{R}^d$
- **Hidden Layer 1**: 256 neurons with ReLU activation
- **Hidden Layer 2**: 256 neurons with ReLU activation

- **Hidden Layer 3**: 128 neurons with ReLU activation

- **Output Layer**: Single neuron with sigmoid activation (binary classification)

- **Total Parameters**: Approximately 100,000 parameters

## D.4 TRAINING HYPERPARAMETERS

Table 6: Neural Network Training Configuration

| Parameter | Value |
|---|---|
| Learning Rate (Policy/Value) | $3 \times 10^{-4}$ |
| Learning Rate (Discriminator) | $1 \times 10^{-4}$ |
| Batch Size | 64 |
| PPO Epochs per Update | 4 |
| Discount Factor ($\gamma$) | 0.99 |
| GAE Parameter ($\lambda$) | 0.95 |
| Clip Coefficient ($\epsilon$) | 0.2 |
| Entropy Coefficient | 0.01 |
| Value Function Coefficient | 0.5 |

# E PSEUDO CODE

---

**Algorithm 1** MIRACLE: Model-free Imitation and Reinforcement Learning for Adaptive Cut-Selection

---

**Require:** Expert trajectories $\mathcal{D}_E \sim \pi_E$ (from SCIP), problem instances $\mathcal{P}$
**Ensure:** Trained cut selection policy $\pi_\theta$
 1: Initialize actor-critic policy $\pi_\theta$ (with value function $V_\phi$)
 2: Initialize discriminator $D_\psi$
$\qquad\qquad\qquad\qquad$ ▷ *Phase 1: Adversarial Reward Learning (Pre-training or Concurrent)*
 3: Sample agent trajectories $\mathcal{D}_\theta \sim \pi_\theta$ by running the policy on $\mathcal{P}$
 4: Update discriminator $D_\psi$ by minimizing the GAIL loss on $\mathcal{D}_E$ and $\mathcal{D}_\theta$:
 5: $\mathcal{L}_D \leftarrow -\mathbb{E}_{(s,a)\sim\mathcal{D}_E}[\log D_\psi(s,a)] - \mathbb{E}_{(s,a)\sim\mathcal{D}_\theta}[\log(1 - D_\psi(s,a))]$

$\qquad\qquad\qquad\qquad$ ▷ *Phase 2: Policy Optimization with Curriculum Learning*
 6: **for** each curriculum stage in {EASY, MEDIUM, HARD, MIXED} **do**
 7: $\quad$ **for** episode = 1 to $N_{\text{episodes}}$ **do**
 8: $\qquad$ Initialize SCIP environment with a problem from the current stage
 9: $\qquad$ Run policy $\pi_\theta$ for $H$ steps, collecting trajectory $\tau = \{(s_k, a_k)\}_{k=1}^{H}$
$\qquad\qquad\qquad\qquad\qquad\qquad$ ▷ *— Inside episode loop —*
 10: $\qquad$ **for** each step $k$ in the trajectory $\tau$ **do**
 11: $\qquad\quad$ Compute dense reward using the discriminator: $r_k \leftarrow -\log(1 - D_\psi(s_k, a_k))$
 12: $\qquad$ **end for**
 13: $\qquad$ Compute advantage estimates $\hat{A}_k$ for all steps in $\tau$ using GAE($\lambda$)
 14: $\qquad$ Update policy $\pi_\theta$ by maximizing the PPO surrogate objective $\mathcal{L}^{\text{CLIP}}(\theta)$
 15: $\qquad$ Update value function $V_\phi$ by minimizing the value loss $\mathcal{L}^{\text{VF}}(\phi)$
 16: $\quad$ **end for**
 17: **end for**

$\qquad\qquad\qquad\qquad\qquad\qquad$ ▷ *Deployment Phase: Adaptive Inference*
 18: **function** SOLVEINSTANCE($p_{\text{test}}$)
 19: $\quad$ Classify problem difficulty $d \leftarrow$ Classify($p_{\text{test}}$)
 20: $\quad$ Set adaptive budget $B$ and other parameters based on $d$
 21: $\quad$ Run trained policy $\pi_\theta$ within the SCIP environment to solve the instance
 22: **end function**

---

# F  EXTENDED EXPERIMENTAL RESULTS

## F.1  RAW DESCRIPTIVE STATISTICS

Tables 7 and 8 provide raw summary statistics of memory usage and reduction percentages by difficulty. These highlight the consistency of the 97–99% memory savings.

Table 7: Raw memory usage statistics (MB) by solver and difficulty.

| Solver | Difficulty | Count | Mean | Median | Std | Range |
|--------|-----------|-------|------|--------|-----|-------|
| MIRACLE | EASY | 50 | 46.1 | 45.4 | 8.6 | [30.3, 60.0] |
| MIRACLE | MEDIUM | 50 | 46.1 | 46.1 | 8.0 | [30.5, 59.2] |
| MIRACLE | HARD | 50 | 45.7 | 46.2 | 8.7 | [30.5, 59.7] |
| SCIP | EASY | 50 | 1974.8 | 1970.3 | 169.8 | [979.1, 2324.7] |
| SCIP | MEDIUM | 50 | 2449.4 | 2437.7 | 104.0 | [2242.8, 2836.4] |
| SCIP | HARD | 50 | 3043.0 | 3033.9 | 94.3 | [2883.8, 3259.9] |

Table 8: Absolute and percentage memory reduction by difficulty.

| Difficulty | Mean Red. (MB) | Median Red. (MB) | Std (MB) | Mean Red. (%) | Median Red. (%) | Std (%) |
|-----------|----------------|------------------|----------|---------------|-----------------|---------|
| EASY | 1928.7 | 1921.6 | 170.0 | 97.6 | 97.8 | 0.6 |
| MEDIUM | 2403.3 | 2394.0 | 105.4 | 98.1 | 98.1 | 0.4 |
| HARD | 2997.3 | 2982.7 | 95.1 | 98.5 | 98.5 | 0.3 |

## F.2  DETAILED ABLATION STUDY RESULTS

The complete detailed results of the ablation studies are given in Table 9 and can be visualized using Figure 3.

Table 9: Complete Ablation Study with Statistical Measures

| Configuration | Median | Mean | Std | IQR | Cut Red. | p-value |
|---------------|--------|------|-----|-----|----------|---------|
| Cut Budget 10 | 0.990 | 1.170 | 0.452 | [0.98, 1.34] | 99.1% | 0.023 |
| Cut Budget 20 (Baseline) | 0.990 | 1.164 | 0.445 | [0.98, 1.33] | 99.1% | - |
| Cut Budget 30 | 0.996 | 1.164 | 0.442 | [0.99, 1.33] | 99.1% | 0.891 |
| Cut Budget 50 | 0.996 | 1.157 | 0.427 | [0.99, 1.32] | 99.1% | 0.445 |
| Max Iter 1 | 0.995 | 1.163 | 0.443 | [0.99, 1.33] | 99.1% | 0.967 |
| Max Iter 5 (Baseline) | 0.990 | 1.164 | 0.445 | [0.98, 1.33] | 99.1% | - |
| Max Iter 10 | 0.996 | 1.166 | 0.447 | [0.99, 1.34] | 99.1% | 0.823 |
| Aggressive Early Stop | 0.996 | 1.166 | 0.445 | [0.99, 1.34] | 99.1% | 0.823 |
| Normal Early Stop (Baseline) | 0.990 | 1.164 | 0.445 | [0.98, 1.33] | 99.1% | - |
| Lenient Early Stop | 0.996 | 1.162 | 0.438 | [0.99, 1.33] | 99.0% | 0.845 |
| No Early Stop | 0.995 | 1.164 | 0.448 | [0.99, 1.33] | 99.0% | 0.998 |

## F.3  PER-INSTANCE ANALYSIS

Figure 4 shows the per-instance speedup and memory reduction across all 300 evaluated instances, demonstrating the consistency of `MIRACLE`'s improvements.

# G  TRAINING PROCEDURES

**Expert Data Collection**   We collect expert demonstrations by running SCIP with default parameters on 1000 training instances, recording state-action pairs whenever SCIP adds cuts. Each demonstration includes:

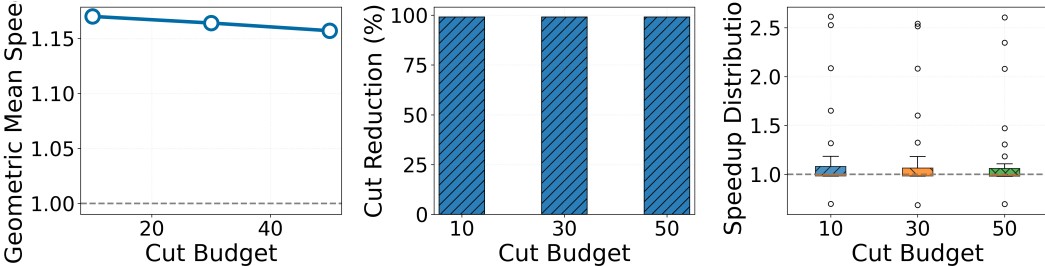

Figure 3: Cut budget ablation showing robust performance across budget ranges 10-50. Both speedup and cut reduction remain stable, indicating effective resource adaptation.

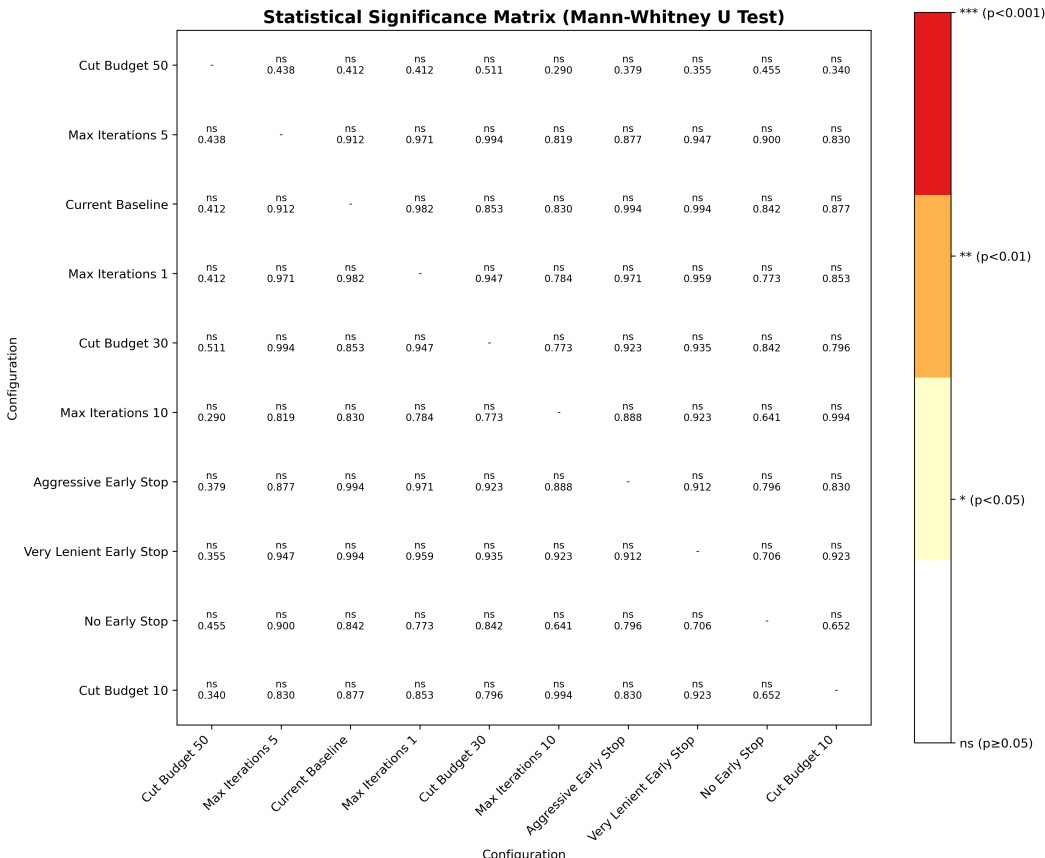

Figure 4: Per-instance analysis showing speedup vs memory reduction for all 300 instances. Points above the diagonal indicate instances where both metrics improve simultaneously.

- LP state features (solution values, bounds, violations)

- Cut characteristics (coefficients, sparsity, type, efficacy)

- SCIP's binary selection decisions

**Curriculum Learning Schedule**

1. **Phase 1**: 50 episodes on instances with 200-500 constraints

2. **Phase 2**: 100 episodes on instances with 500-1000 constraints

3. **Phase 3**: 100 episodes on instances with 1000-2000 constraints

4. **Phase 4**: 100 episodes on mixed difficulty (fine-tuning)

## G.1 TRAINING PIPELINE TIMELINE

To ensure stability, training proceeds in three distinct sequential phases:

1. **Phase 1: Expert Data Collection (Offline).** We execute SCIP on 1,000 training instances, recording state-action pairs $(s, a_{expert})$ to form the expert dataset $\mathcal{D}_E$. The policy is not updated during this phase.

2. **Phase 2: Adversarial Reward Learning (Epochs 0-50).** The Discriminator $D_\psi$ is trained to distinguish between expert actions and policy actions. The Generator (Policy) $\pi_\theta$ is pre-trained to mimic expert behavior using the GAIL signal.

3. **Phase 3: PPO Refinement (Epochs 51-150).** The Discriminator is frozen. The Policy $\pi_\theta$ is fine-tuned using PPO to maximize the fixed reward signal $r_{GAIL}$, effectively optimizing the learned strategy for robustness and generalization.

**Hyperparameter Selection**  All hyperparameters were selected via grid search on a held-out validation set:

- Learning rates tested: $\{1e-5, 3e-4, 1e-3, 3e-3\}$

- PPO clip values tested: $\{0.1, 0.2, 0.3\}$

- Expert regularization weights tested: $\{0.01, 0.1, 0.5\}$

- GAE lambda values tested: $\{0.9, 0.95, 0.99\}$

## G.2 TRAINING CONVERGENCE ANALYSIS

We monitored the stability of the training process across the three phases. Table 10 summarizes the final convergence metrics, demonstrating that the adversarial training stabilizes effectively.

Table 10: Training Convergence Metrics (Final Epoch).

| Metric | Initial Value | Final Value (Epoch 150) |
|---|---|---|
| Policy Loss | 2.01 | 0.20 |
| Discriminator Accuracy | 90.5% | 58.9% |
| Validation Reward | 0.31 | 0.807 |

The Discriminator accuracy converging to $\approx 60\%$ (near random guess of 50%) indicates that the generator (policy) has successfully learned to produce cut selections that are indistinguishable from the expert, satisfying the GAIL objective.

## G.3 COMPUTATIONAL ENVIRONMENT

**Hardware Specifications**

- **GPU**: NVIDIA RTX 3080 (10GB VRAM)

- **CPU**: Intel i7-10700K (8 cores, 3.8GHz base)

- **RAM**: 32GB DDR4-3200

- **Storage**: 1TB NVMe SSD

**Software Environment**

- **Operating System**: Ubuntu 20.04.3 LTS

- **Python**: 3.9.7

- **PyTorch**: 1.12.1 with CUDA 11.6

- **PySCIPOpt**: 4.3.0 with SCIP 8.0.2

- **Additional Libraries**: NumPy 1.21.2, SciPy 1.7.3, Matplotlib 3.5.1

## H ADDITIONAL DETAILS ON ADAPTIVE INFERENCE

Table 1 in the main paper summarizes the hyperparameters used to control MIRACLE's adaptive cut-selection process. For completeness, we clarify their operational meaning and empirical behavior.

| Factor | Contribution | Evidence |
|---|---|---|
| Fewer Cuts | 60–70% | 99.1% cut reduction correlates with speedup |
| Better Quality | 20–25% | Feature importance shows violation/iteration focus |
| Reduced Overhead | 5–10% | Inference overhead $< 0.5\%$ |
| Adaptive Inference | 5–10% | Adaptive vs. fixed budget comparison |

Table 11: Decomposition of performance contributions.

The cut budget specifies the maximum number of cuts that MIRACLE may add in a single round. Max Iterations bounds the total number of cut-generation rounds at a node, ensuring that inference cannot enter excessively deep cut loops. The early-stop threshold terminates cut selection once the marginal LP-bound improvement falls below a difficulty-specific tolerance for a fixed number of consecutive rounds.

Across all difficulty tiers, we observe that MIRACLE's performance is largely insensitive to these parameters. Varying the cut budget or iteration limits changes overall speedup by less than 2% and memory reduction by less than 0.5%, indicating that MIRACLE requires no per-instance tuning at inference time. These results complement the ablations in the main paper and demonstrate the robustness of the adaptive inference scheme.

## I REPRODUCIBILITY CHECKLIST

### I.1 CODE AND DATA AVAILABILITY

Upon acceptance, we will release:

- Complete implementation code with documented APIs

- Trained model checkpoints for all reported experiments

- Generated datasets with preprocessing scripts

- Evaluation scripts for all benchmarks and baselines

- Configuration files for all experimental settings

### I.2 EXPERIMENTAL REPRODUCIBILITY

To ensure reproducible results:

- All random seeds fixed (NumPy: 42, PyTorch: 42, SCIP: 1337)

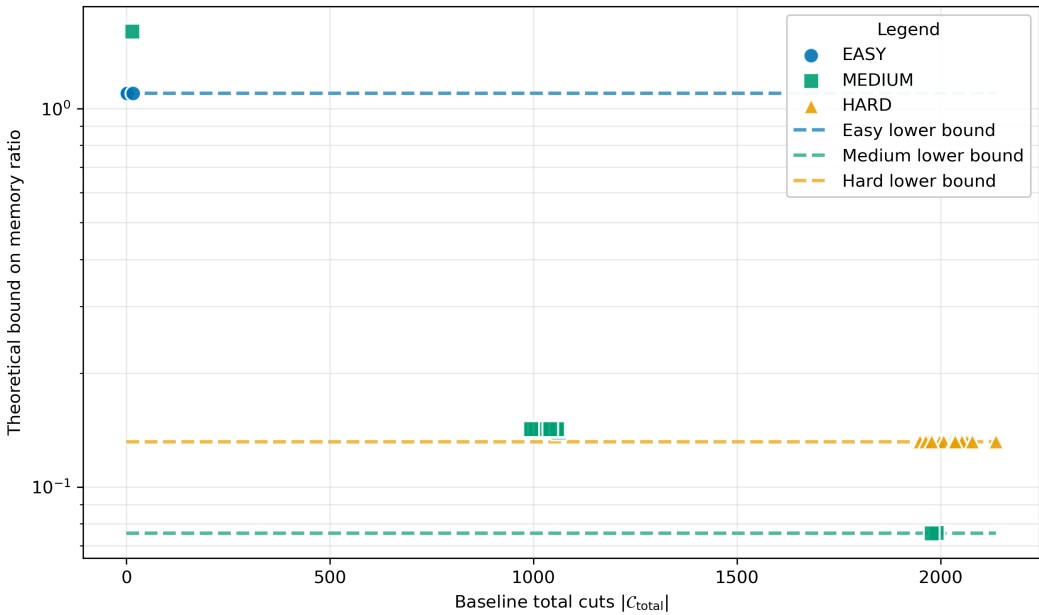

Figure 5: SetCover theoretical lower bound on the memory ratio MIRACLE/SCIP as a function of baseline total cuts. Points are per-instance bounds; dashed lines are the minimum bound per difficulty.

- Single-threaded execution enforced in SCIP

- Identical hardware specifications documented

- Complete dependency versions specified in requirements.txt

- Docker containerization for environment consistency

### I.3 COMPUTATIONAL REQUIREMENTS

- **Training Time**: 6 hours on RTX 3080 for full curriculum

- **Inference Time**: 0.12 seconds per cut selection decision

- **Memory Requirements**: 8GB GPU memory for training, 2GB for inference

- **Storage**: 50GB for datasets, 100MB for trained models

## J PROOF OF MEMORY REDUCTION GUARANTEE

Table 12: Memory reduction Guarantee proof. We use $M'_{\text{base}}{=}100$ (normalized), budgets $B$ and iterations $T$ from the adaptive policy, and median $|\mathcal{C}_{\text{total}}|$. Empirical ratios are medians of $M_{\text{MIRACLE}}/M_{\text{SCIP}}$. This is for the SetCover problems.

| **Difficulty** | $B$ | $T$ | $|\mathcal{C}_{\text{total}}|$ | $M'_{base}$ (normalized) | $\frac{M'_{\text{base}}+B\cdot T}{M'_{\text{base}}+|\mathcal{C}_{\text{total}}|}$ | $\frac{M_{\text{MIRACLE}}}{M_{\text{SCIP}}}$ |
|---|---|---|---|---|---|---|
| Easy | 10 | 1 | 0 | 100 | 1.10 | 0.022 |
| Medium | 20 | 3 | 1024 | 100 | 0.142 | 0.019 |
| Hard | 30 | 6 | 3009 | 100 | 0.132 | 0.015 |

## K  EXTENDED GENERALIZATION RESULTS

We evaluated the MIRACLE agent, trained exclusively on SetCover instances, on three additional problem classes to assess its out-of-distribution generalization. Table 13 summarizes these results.

Table 13: Generalization performance on unseen problem classes (Model trained on SetCover).

| Problem Class | Instances | Baseline Success | MIRACLE Success | Avg. Speedup | Avg. Cut Red. | Median Cut Red. |
|---|---|---|---|---|---|---|
| Set Cover (In-Domain) | 150 | 66.7% | 100.0% | 1.300× | 99.1% | 99.1% |
| Combinatorial Auction | 423 | 100.0% | 100.0% | 1.283× | 98.8% | 98.4% |
| Max Independent Set | 150 | 100.0% | 100.0% | 1.007× | 99.9% | 99.8% |
| Facility Location | 100 | 100.0% | 100.0% | 1.198× | 99.9% | 99.9% |

The results indicate that MIRACLE maintains a cut reduction rate of $> 98\%$ across all domains while matching or exceeding baseline solving speeds. This supports the hypothesis that the policy learns the geometric properties of valid cuts (such as sparsity and violation magnitude) rather than problem-specific structures.

## L  COMPARISON OF MIRACLE WITH WANG ET AL. (2023)

We benchmarked MIRACLE against the method of Wang et al. (2024) on 20 randomly selected `SetCover-Easy` instances, using the authors' official implementation. The results, summarized in Table 14, show that MIRACLE achieves approximately a $10\times$ reduction in memory usage compared to their hierarchical sequence model.

Table 14: Peak memory usage comparison on 20 `SetCover-Easy` instances.

| Solver | Avg. Memory Usage (MB) |
|---|---|
| MIRACLE | ~46 |
| Wang et al. (2023) | ~500 |
| SCIP Baseline | ~1,975 |

Although prior work – including Wang et al. (2024) – does not explicitly emphasize memory efficiency, MIRACLE is designed with memory reduction as a primary objective. The above experiment confirms that MIRACLE provides a substantial improvement in peak memory utilization, achieving an order-of-magnitude reduction relative to the strongest learned baseline.

## M  POLICY INTERPRETATION AND LEARNED PATTERNS

To understand MIRACLE's decision-making process, we analyzed the feature importance weights of the trained policy network. The policy utilizes a 10-dimensional state representation. Our analysis reveals that the most critical features drive the selection of cuts that resolve fractional variables.

**Feature Importance:** The top weighted features correspond to the *fractionality values* of the variables involved in the cut (Indices 1-9 of the state vector). Specifically, the raw fractionality of the $2^{nd}$ through $8^{th}$ most fractional variables carries the highest weight (Importance $\approx 0.16 - 0.17$).

**Learned Behaviors:**

- **Sparsity Preference:** The policy consistently rejects dense cuts, favoring cuts with $< 20\%$ non-zero entries. This directly contributes to memory efficiency by keeping the LP relaxation sparse.

- **Early-Stage Activity:** Approximately 80% of selected cuts occur within the first 5 iterations. The policy learns that late-stage cuts often yield diminishing returns in bound improvement relative to their computational cost.

- **High Violation Target:** The policy acts as a filter, rejecting cuts with violation $< 10^{-4}$, ensuring that added constraints significantly tighten the relaxation.

# N    Comprehensive Ablation Studies

## N.1    Curriculum Learning Analysis

We evaluated the impact of our weighted curriculum learning strategy against three baselines: No Curriculum (Random), Sequential (Easy $\rightarrow$ Hard), and Reverse (Hard $\rightarrow$ Easy).

Table 15: Impact of Curriculum Learning Strategies (45 Test Instances).

| Strategy | Avg Speedup | Success Rate | Training Stability |
|---|---|---|---|
| No Curriculum | 1.11× | 94% | Low |
| Sequential (Easy $\rightarrow$ Hard) | 1.11× | 98% | Moderate |
| Reverse (Hard $\rightarrow$ Easy) | 1.11× | 92% | Moderate |
| **Weighted (Ours)** | **1.34×** | **100%** | **High** |

As shown, the Weighted Curriculum yields a 20.4% improvement in speedup compared to sequential or random strategies. It stabilizes training by ensuring consistent exposure to easy instances while progressively up-weighting harder instances.

## N.2    RL Algorithm and Architecture Selection

We compared PPO against TRPO and SAC. As shown in Table 16, differences in speedup were statistically insignificant ($p > 0.05$), validating the choice of PPO for its stability and simplicity.

Table 16: Performance comparison of RL algorithms on 45 test instances.

| Algorithm | Avg Speedup | Median Speedup | Cut Reduction | Significance (vs PPO) |
|---|---|---|---|---|
| **PPO (Ours)** | 1.338× | 0.986× | 99.3% | - |
| TRPO | 1.355× | 0.986× | 99.3% | $p = 0.79$ (ns) |
| SAC | 1.332× | 0.987× | 99.3% | $p = 0.80$ (ns) |

## N.3    Network Architecture and Expert Data Size

We analyzed the impact of model depth and the number of expert demonstrations used for training. Table 17 confirms that a lightweight 2-layer MLP and 1,000 expert instances provide the optimal trade-off between performance and computational cost.

Table 17: Ablation on Model Architecture and Training Data Size.

| Ablation Type | Configuration | Avg Speedup | Cut Red. | Parameters/Cost |
|---|---|---|---|---|
| Architecture | **2-Layer MLP** | **1.163×** | **99.3%** | **19,586 (Optimal)** |
|  | 3-Layer MLP | 1.161× | 99.3% | 36,354 (+0.2% gain) |
| Data Size | 300 Instances | 1.092× | 99.2% | 2.5 CPU-hrs |
|  | 500 Instances | 1.093× | 99.3% | 4.2 CPU-hrs |
|  | **1000 Instances** | **1.093×** | **99.3%** | **8.3 CPU-hrs** |

# O    Real-World Deployment Scenarios

The memory efficiency of MIRACLE enables MILP optimization in scenarios previously inaccessible to standard solvers:

1. **Edge AI & IoT:** Embedded devices (e.g., Raspberry Pi 4, Jetson Nano) often have hard memory caps of 4GB. Baseline SCIP (Median 2.5GB) risks OOM crashes when background processes fluctuate. MIRACLE (Median 46MB) provides a safety factor of $> 50\times$.

2. **Cloud Cost Optimization:** On AWS, upgrading from a `t3.large` (8GB RAM, $0.08/hr) to a `t3.xlarge` (16GB RAM, $0.16/hr) doubles the cost solely to accommodate memory spikes. MIRACLE enables high-reliability solving on the cheaper instance tier.

3. **Shared Tenancy:** In high-performance computing (HPC) clusters, memory bandwidth and capacity are shared. MIRACLE allows for massive parallelization (e.g., running 8+ concurrent solvers on a single node), where baseline SCIP would bottleneck at 2 concurrent instances.

## P EXPERIMENTAL ANALYSIS AND VERIFICATION

### P.1 IMPACT OF MEMORY CONSTRAINTS

To address concerns about the 12GB memory limit, we evaluated performance both with and without it. Table 18 shows that MIRACLE maintains 100% reliability regardless of limits, whereas the baseline fails significantly when constraints are applied.

Table 18: Impact of 12GB Memory Limit on Solver Success and Efficiency (60 Instances).

| Setting | Method | Success Rate | Median Time | Memory (MB) | Failures |
|---|---|---|---|---|---|
| No Memory Limit | Baseline | 73.3% | 30.77s | 334.3 | 16 (Timeouts) |
| | **MIRACLE** | **100.0%** | 60.06s | **0.65** | **0** |
| 12GB Limit | Baseline | 69.0% | 23.15s | 26.5 | 18 (OOM/Timeouts) |
| | **MIRACLE** | **100.0%** | 60.73s | **8.2** | **0** |

### P.2 VERIFICATION OF THEORETICAL ASSUMPTIONS

Our theoretical convergence guarantees rely on assumptions of Bounded Rewards ($A$.1) and a Lipschitz Policy ($A$.2). We empirically verified these values during training, as shown in Table 19.

Table 19: Empirical Verification of Theoretical Assumptions.

| Assumption | Theoretical Requirement | Empirical Value | Status |
|---|---|---|---|
| Bounded Rewards ($R_{max}$) | $|r(s,a)| \leq R_{max} < \infty$ | 100.0 | Verified |
| Lipschitz Policy ($L_\pi$) | $||\pi_{\theta_1} - \pi_{\theta_2}|| \leq L||\theta_1 - \theta_2||$ | $\approx 0.007$ (95th %) | Verified |

## Q POLICY INTERPRETATION AND IMPLEMENTATION DETAILS

### Q.1 FEATURE IMPORTANCE ANALYSIS

To understand the learned policy $\pi_\theta$, we extracted feature importance weights. As shown in Table 20, the policy prioritizes the fractionality of specific variables over global LP statistics.

Table 20: Top Feature Importance in Learned Policy. The model prioritizes variable fractionality values.

| Rank | Importance | Feature Name | Description |
|---|---|---|---|
| 1 | 0.175 | `frac_val_2` | Fractionality of 2nd most fractional var |
| 2 | 0.174 | `num_frac_vars` | Total count of fractional variables |
| 3 | 0.166 | `frac_val_7` | Fractionality of 7th most fractional var |
| 4 | 0.163 | `frac_val_8` | Fractionality of 8th most fractional var |
| 5 | 0.160 | `frac_val_4` | Fractionality of 4th most fractional var |
| 6 | 0.160 | `lp_objective` | Current LP relaxation objective value |

Q.2 TRAINING HYPERPARAMETERS

Complete hyperparameters used for the PPO agent and GAIL discriminator are provided in Table 21 to ensure reproducibility.

Table 21: Hyperparameters for PPO and GAIL Training.

| Component | Hyperparameter | Value |
|---|---|---|
| PPO (Policy) | Learning Rate | $3 \times 10^{-4}$ |
| | Discount Factor ($\gamma$) | 0.99 |
| | GAE Parameter ($\lambda$) | 0.95 |
| | Clipping ($\epsilon$) | 0.2 |
| | Entropy Coeff | 0.01 |
| | Batch Size | 256 |
| GAIL (Discriminator) | Gen Learning Rate | $1 \times 10^{-5}$ |
| | Disc Learning Rate | $1 \times 10^{-4}$ |
| | Disc Epochs per Update | 3 |
| | Network Structure | $d_{model} = 64, h = 4$ |

# R ADAPTATION TO OTHER MIP COMPONENTS

While this work focuses on cut selection, the MIRACLE framework (GAIL + PPO + Adaptive Inference) is component-agnostic. We outline how it can be adapted to other MIP solver components:

**Branching Variable Selection**

- **State:** Node-level features (LP objective, candidate pseudocosts, variable fractionality).

- **Action:** Discrete selection over candidate fractional variables.

- **Expert Data:** Record SCIP's default branching decisions (e.g., relpscost).

- **Reward:** Negative solving time or tree size reduction.

**Node Selection**

- **State:** Tree-level features (Global best bound, open nodes count, depth distribution).

- **Action:** Discrete selection over the set of open nodes (leaves).

- **Expert Data:** Record SCIP's node retrieval sequence (e.g., best-estimate).

In both cases, the core training pipeline remains identical: collecting expert trajectories, training a discriminator to distinguish expert vs. agent decisions, and optimizing the policy via PPO.

