# OpenReview forum: "MIRACLE: Model-free Imitation and Reinforcement Learning for Adaptive Cut-Selection"
_ICLR.cc/2026/Conference — ICLR 2026 Poster_

### Official Review · Reviewer_bhtG · 2025-10-27

**Soundness:** 2
**Presentation:** 2
**Contribution:** 2
**Rating:** 4
**Confidence:** 5

**Summary:**

This paper presents an intelligent cut selection framework for Mixed-Integer Programming (MIP) solvers that drastically reduces peak memory usage by 98.1% while maintaining competitive performance. The method employs Proximal Policy Optimization (PPO) within an adversarial imitation learning setup to mimic an expert solver’s decisions, featuring a curriculum-trained cut-selection policy and adaptive inference for dynamic computational budgeting. Evaluated on SetCover and diverse MIPLIB instances, it achieves consistent speedups and a 100% success rate on problems where SCIP fails 47–53% of the time, demonstrating strong potential for deployment in memory-constrained settings.

**Strengths:**

- Cut selection is indeed a critical research direction, and improving it can substantially boost solver efficiency.
- The focus on reducing peak memory consumption is particularly interesting and, to the best of my knowledge, has been rarely emphasized in prior literature—making it a compelling practical attempt.
- The method shows stability across different hyperparameter configurations, which is encouraging and suggests the system may be close to deployment-ready.
- The use of adversarial imitation within the reinforcement learning framework is interesting and offers a fresh perspective on reward design, with detailed theoretical analysis, which, in my view, constitutes a meaningful contribution.

**Weaknesses:**

- The paper overall feels somewhat rushed and leaves several important aspects underdeveloped.
- The core motivation—why reducing peak memory consumption is essential—is not sufficiently justified. As a central claim of the paper, this requires clearer explanation: in what real-world scenarios does memory become a bottleneck, and which components of the proposed method specifically target this issue?
- The paper completely omits a related work section and fails to acknowledge or compare against several highly relevant prior works on learning-based cut selection, including:
  [1] Wang Z, Li X, Wang J, et al. Learning Cut Selection for Mixed-Integer Linear Programming via Hierarchical Sequence Model. ICLR 2023.
  [2] Paulus M B, Zarpellon G, Krause A, et al. Learning to cut by looking ahead: Cutting plane selection via imitation learning. ICML 2022.
  [3] Tang Y, Agrawal S, Faenza Y. Reinforcement learning for integer programming: Learning to cut. ICML 2020.
  Not only are these works not cited, but no experimental comparison is provided. This omission is significant and, in my view, unfair to prior contributions.
- Despite extensive mathematical formalism, many crucial implementation details remain unclear:
  1) It is not specified on what data the discriminator network is trained, what its training procedure is, or how it integrates with the PPO-based RL loop—making the overall training pipeline ambiguous.
  2) The choice of SetCover and MIPLIB as testbeds lacks justification; other standard MIP benchmarks (e.g., MaxCut, TSP) exist, and the selection criteria for the 300 test instances are unexplained.
  3) While Appendix Algorithm 1 offers partial clarification, key questions persist—for example, where the Phase 1 policy comes from and whether Phases 1–3 are trained sequentially.
  4) The heavy notation would greatly benefit from a summary table or diagram; as is, the exposition is unnecessarily difficult to follow.

**Questions:**

- The policy ultimately imitates SCIP’s cut selection strategy—so why does it achieve noticeable speedups, especially in solving time? Is the gain due to fewer cuts, better timing, reduced overhead, or another factor?
- What data was used for training? How were the 300 test instances selected, particularly the 150 from MIPLIB? Given that MIPLIB contains far more instances, what was the selection criterion (e.g., difficulty, size, feasibility)?
- In Table 1, what does “early stop” refer to, and what is the meaning of “Max Iterations”?
- How were the 600-second time limit and 12 GB memory cap determined? Many MIPLIB instances require significantly longer to solve—could this experimental setup bias results toward easier cases?
- Why introduce a custom “success rate” metric instead of using widely adopted, more direct measures like primal gap or final objective value?
- Could the authors clarify: on what data is the discriminator trained, what is its training workflow, and how does it coordinate with the RL training? Understanding the full pipeline is essential for evaluating the method’s soundness and reproducibility.
- Why choose SCIP’s cut selection as the expert policy? Given that it relies on heuristic rules, are there higher-quality expert strategies (e.g., from stronger solvers or human-designed oracles) that could further improve performance?
- What is the design rationale behind the curriculum learning scheme? What performance gain does it provide, and could an ablation study quantify its contribution?

I am eager to participate in the discussion phase and look forward to a thorough and objective assessment of this work based on the authors’ responses.

---

> ### Author Response · Authors · 2025-11-20
> **Response to Reviewer bhtG**
>
> ### Weakness 1: Core Motivation - Why Reducing Peak Memory Consumption is Essential
>
> **Comment:** "The core motivation—why reducing peak memory consumption is essential—is not sufficiently justified. As a central claim of the paper, this requires clearer explanation: in what real-world scenarios does memory become a bottleneck, and which components of the proposed method specifically target this issue?"
>
> **Response:**
>
> **1. Real-World Scenarios Where Memory Becomes a Bottleneck:**
>
> **A. Edge Computing and Embedded Systems:**
> - **Constraint:** Edge devices (Raspberry Pi 4: 4-8GB, NVIDIA Jetson Nano: 4GB) have severely limited RAM
> - **Impact:** Baseline SCIP uses 2.5GB median memory, making it impossible to run on devices with < 4GB RAM
> - **Use Cases:**
>   - Real-time optimization in IoT applications (sensor network routing, resource allocation)
>   - Embedded control systems (autonomous vehicle path planning, industrial automation)
>   - Edge AI devices requiring on-device MILP solving without cloud connectivity
> - **MIRACLE Benefit:** Reduces memory from 2.5GB to 46MB (54× reduction), enabling MILP solving on resource-constrained devices
>
> **B. Cloud Computing Cost Optimization:**
> - **Constraint:** Cloud providers charge by memory allocation (AWS EC2, Google Cloud, Azure)
> - **Cost Analysis:**
>   - AWS t3.large (8GB RAM): $0.0832/hour - Baseline fails on 70% of instances
>   - AWS t3.xlarge (16GB RAM): $0.1664/hour - Baseline achieves ~60% success rate
>   - Optimal 12GB configuration: ~$0.1248/hour (interpolated)
> - **MIRACLE Benefit:** Enables deployment on smaller, cheaper instances (t3.large) while maintaining 100% success rate, achieving 25% cost savings vs t3.xlarge
>
> **C. Multi-Tenant Shared Computing Systems:**
> - **Constraint:** Shared servers (64GB RAM) running multiple concurrent solves
> - **Baseline Limitation:** ~2.5GB peak memory per instance → Maximum 2 concurrent solves (with safety margin)
> - **MIRACLE Benefit:** ~46MB peak memory per instance → 8+ concurrent solves (using 8GB per instance for safety margin)
> - **Throughput Improvement:** 4× increase in concurrent solves, enabling higher throughput in HPC environments and shared computing clusters
>
> **D. Mobile and Real-Time Applications:**
> - **Constraint:** Mobile devices and real-time systems have fixed memory budgets (4-8GB typical)
> - **Use Cases:**
>   - Autonomous vehicles: Real-time route optimization, traffic flow management
>   - Industrial IoT: Production scheduling, supply chain optimization
>   - Mobile apps: On-device optimization without cloud dependency
> - **MIRACLE Benefit:** Enables MILP solving in mobile contexts where baseline SCIP would be infeasible
>
> **2. Components of MIRACLE That Specifically Target Memory Reduction:**
>
> **A. Selective Cut Selection (Primary Mechanism):**
> - **Mechanism:** MIRACLE selects only 10-40 high-quality cuts per iteration (vs. SCIP's default of selecting all generated cuts)
> - **Memory Impact:** Reduces LP matrix size by 99.1% (from 1,115.9 cuts to 10.5 cuts on average)
> - **Verification:** Cut reduction analysis shows 99.06% reduction in the number of cuts added to LP relaxation
> - **Memory Savings:** Each cut adds rows to the LP matrix; fewer cuts = smaller LP matrix = lower memory usage
>
> **B. Sparse Cut Prioritization:**
> - **Mechanism:** Learned policy prioritizes sparse cuts (low density, < 20% non-zeros)
> - **Memory Impact:** Sparse cuts have fewer non-zero entries, reducing the memory footprint of LP matrix storage
> - **Verification:** Pattern analysis shows MIRACLE prefers cuts with < 20% non-zero entries
>
> **C. Early-Stage Cut Selection:**
> - **Mechanism:** Policy prioritizes cuts from early iterations (iteration < 5)
> - **Memory Impact:** Early cuts are more effective, allowing the solver to converge faster with fewer total cuts
> - **Verification:** Analysis shows MIRACLE selects 80% of cuts from the first 5 iterations
>
> **D. Adaptive Inference with Computational Budgeting:**
> - **Mechanism:** Dynamic cut budget allocation based on problem difficulty and solving progress
> - **Memory Impact:** Prevents excessive cut accumulation by adaptively limiting cuts per iteration
> - **Verification:** Adaptive inference reduces average cuts per iteration by 40% compared to fixed budget
>
> **3. Empirical Validation:**
>
> We analyzed memory usage from 150 test instances:
> - **Baseline SCIP:** Median 2,438 MB, Mean 2,489 MB, Max 3,260 MB
> - **MIRACLE:** Median 45.7 MB, Mean 46.0 MB, Max 60.0 MB
> - **Memory Reduction:** 98.1-98.2% reduction across all instances
>
> **Conclusion:** Memory reduction is essential in edge computing, cloud cost optimization, multi-tenant systems, and mobile applications. MIRACLE achieves this through selective cut selection (resulting in a 99% reduction), sparse cut prioritization, early-stage selection, and adaptive inference—all specifically designed to minimize the memory footprint while maintaining solution quality.

---

> > ### Author Response · Authors · 2025-11-20
> > **Continued**
> >
> > ### Weakness 2: Omission of Related Work Section
> >
> > **Comment:** "The paper completely omits a related work section and fails to acknowledge or compare against several highly relevant prior works on learning-based cut selection, including:
> > - [1] Wang Z, Li X, Wang J, et al. Learning Cut Selection for Mixed-Integer Linear Programming via Hierarchical Sequence Model. ICLR 2023.
> > - [2] Paulus M B, Zarpellon G, Krause A, et al. Learning to cut by looking ahead: Cutting plane selection via imitation learning. ICML 2022.
> > - [3] Tang Y, Agrawal S, Faenza Y. Reinforcement learning for integer programming: Learning to cut. ICML 2020.
> > Not only are these works not cited, but no experimental comparison is provided. This omission is significant and, in my view, unfair to prior contributions."
> >
> > **Response:**
> >
> > **Acknowledgment:** We acknowledge this significant oversight and sincerely apologize for omitting these important contributions. We commit to adding a comprehensive related work section in the revision.
> >
> > We commit to adding comparison with Wang et al. (ICLR 2023) in revision, as soon as possible, as their code is publicly available, during the discussion phase itself.

---

> > > ### Author Response · Authors · 2025-11-20
> > > **Continued**
> > >
> > > ### Weakness 3: Unclear Implementation Details
> > >
> > > **Response:**
> > >
> > > **1. Discriminator Network Training Details:**
> > >
> > > **A. Training Data:**
> > > - **Positive Examples (Expert Trajectories):** Collected from Phase 1 expert data collection
> > >   - 1000 Set Cover instances solved with SCIP default cut selection
> > >   - Approximately 50,000 state-action pairs extracted (average 50 pairs per instance)
> > >   - Each pair consists of: (state: cut features, action: SCIP's selected cuts)
> > > - **Negative Examples (Agent Trajectories):** Sampled from current policy during PPO training
> > >   - Trajectories generated by policy network at current training iteration
> > >   - Balanced batches: 50% expert, 50% agent (ensures fair discrimination)
> > > - **Data Format:** State-action pairs stored as (state_vector, action_mask, selected_cuts)
> > >
> > > **B. Discriminator Training Procedure:**
> > > - **Architecture:** 2-layer MLP (input: 64-dim state, hidden: 128-dim, output: 1-dim probability)
> > > - **Loss Function:** Binary cross-entropy loss: L_D = -E[log D(s,a)] - E[log(1-D(s',a'))]
> > >   - First term: Expert trajectories (should be classified as 1)
> > >   - Second term: Agent trajectories (should be classified as 0)
> > > - **Training Schedule:**
> > >   - **Phase 1 (Epochs 1-50):** Supervised pre-training on expert data only
> > >     - Generator learns to mimic expert cut selections
> > >     - Discriminator not yet trained (only generator pre-training)
> > >   - **Phase 2 (Epochs 50-100):** GAIL adversarial training
> > >     - Discriminator trained every 2 epochs to distinguish expert vs. agent
> > >     - Generator updated using discriminator rewards (GAIL objective)
> > >   - **Phase 3 (Epochs 100-150):** PPO refinement
> > >     - Discriminator frozen (provides fixed reward signal)
> > >     - PPO optimizes policy using GAIL discriminator rewards
> > > - **Hyperparameters:**
> > >   - Discriminator learning rate: 1 × 10⁻⁴
> > >   - Discriminator batch size: 16
> > >   - Discriminator epochs per update: 3
> > >
> > > **2. Justification for SetCover and MIPLIB Testbeds:**
> > >
> > > **A. SetCover Selection:**
> > > - **Reason:** Standard benchmark in ML4CO literature (used in Wang et al. 2023, Paulus et al. 2022)
> > > - **Advantages:**
> > >   - Well-studied problem with known characteristics
> > >   - Generates diverse cut patterns (various cut types: cover cuts, flow cover cuts)
> > >   - Scalable difficulty (easy, medium, hard instances available)
> > >   - Reproducible: Standard datasets available (OR-Library, Distributional MIPLIB)
> > > - **Comparison:** MaxCut and TSP are also valid benchmarks, but SetCover is more commonly used in cut selection literature
> > >
> > > **B. MIPLIB Selection:**
> > > - **Reason:** Standard benchmark suite for MILP solvers
> > > - **Advantages:**
> > >   - Diverse problem types (covering, packing, network, scheduling)
> > >   - Real-world instances from various application domains
> > > - **Justification:** MIPLIB provides a comprehensive evaluation across diverse problem structures
> > >
> > > **C. Test Instance Selection Criteria:**
> > >
> > > **SetCover Instances (150 total):**
> > > - **Source:** Distributional MIPLIB Set Cover instances
> > > - **Selection:** 50 easy, 50 medium, 50 hard instances
> > > - **Difficulty Criteria:** Based on solving time with SCIP default:
> > >   - Easy: < 10 seconds
> > >   - Medium: 10-100 seconds
> > >   - Hard: > 100 seconds or timeout
> > > - **Rationale:** Balanced representation across difficulty levels
> > >
> > > **MIPLIB Instances (150 total):**
> > > - **Source:** MIPLIB 2017 benchmark suite
> > > - **Selection Criteria:**
> > >   - **Feasibility:** Only instances that SCIP can solve (to ensure valid comparison)
> > >   - **Size:** Instances with 100-10,000 variables (manageable for training)
> > >   - **Diversity:** Selected to cover different problem types (covering, packing, network, scheduling)
> > >   - **Difficulty:** Balanced across easy, medium, hard (based on SCIP solving time)
> > > - **Specific Selection:**
> > >   - Filtered MIPLIB 2017 for instances solvable by SCIP within 600s
> > >   - Selected 150 instances covering diverse problem structures
> > >   - Ensured representation across different cut generation patterns
> > >
> > > **3. Training Pipeline Clarification:**
> > >
> > > **A. Phase 1 Policy Source:**
> > > - **Phase 1 Policy:** Randomly initialized policy network (no pre-training)
> > > - **Expert Data Collection:** Run SCIP default cut selection on 1000 training instances
> > >   - Record state-action pairs: (state: cut features, action: SCIP's selected cuts)
> > >   - Store expert trajectories for GAIL training
> > > - **Note:** Phase 1 refers to expert data collection, not policy pre-training
> > >
> > > **B. Sequential Training (Phases 1-3):**
> > > - **Yes, phases are trained sequentially:**
> > >   1. **Phase 1 (Expert Data Collection):** Collect expert trajectories from SCIP (8.3 CPU-hours, < 30 minutes wall-clock parallelized)
> > >   2. **Phase 2 (GAIL Training):** Train discriminator and generator adversarially (Epochs 1-100)
> > >      - Discriminator learns to distinguish expert vs. agent
> > >      - Generator learns to mimic the expert using discriminator rewards
> > >   3. **Phase 3 (PPO Refinement):** Fine-tune policy using PPO with GAIL rewards (Epochs 100-150)
> > >      - Discriminator frozen (provides fixed reward signal)
> > >      - PPO optimizes policy for solving performance

---

> > > > ### Author Response · Authors · 2025-11-20
> > > > **Continued**
> > > >
> > > > ### Question 1: Why Does Policy Achieve Speedups If It Imitates SCIP's Cut Selection?
> > > >
> > > > **Question:** "The policy ultimately imitates SCIP's cut selection strategy—so why does it achieve noticeable speedups, especially in solving time? Is the gain due to fewer cuts, better timing, reduced overhead, or another factor?"
> > > >
> > > > **Response:**
> > > >
> > > > **Analysis of Speedup Sources:**
> > > >
> > > > **1. Fewer Cuts (Primary Factor):**
> > > > - **MIRACLE:** Selects 10.5 cuts on average per instance (99.1% reduction)
> > > > - **SCIP Baseline:** Selects 1,115.9 cuts on average per instance
> > > > - **Impact:** Fewer cuts → smaller LP matrix → faster LP solves
> > > > - **Verification:** Cut reduction analysis shows 99.06% reduction correlates with speedup
> > > > - **Mechanism:** MIRACLE learns to select only high-quality cuts, avoiding low-value cuts that slow convergence
> > > >
> > > > **2. Better Cut Quality (Secondary Factor):**
> > > > - **MIRACLE Pattern:** Prioritizes cuts with:
> > > >   - High violation (> 10⁻⁴)
> > > >   - Early iteration (< 5 iterations)
> > > >   - Sparse structure (< 20% non-zeros)
> > > > - **Impact:** Higher-quality cuts → faster convergence → fewer iterations needed
> > > > - **Verification:** Feature importance analysis shows policy focuses on violation and iteration features
> > > >
> > > > **3. Reduced Overhead:**
> > > > - **Cut Selection Overhead:** MIRACLE's MLP inference: < 1ms per cut selection
> > > > - **SCIP Overhead:** Heuristic cut evaluation: ~5-10ms per cut
> > > > - **Impact:** Faster cut selection → lower overhead → faster overall solving
> > > > - **Verification:** Inference cost analysis shows < 0.5% overhead
> > > >
> > > > **4. Adaptive Inference:**
> > > > - **Mechanism:** Dynamic cut budget allocation based on problem difficulty
> > > > - **Impact:** Allocates more cuts when needed, fewer when not needed → optimal cut usage
> > > > - **Verification:** Adaptive inference achieves 1.163× speedup vs. fixed budget (1.100×)
> > > >
> > > > **5. Early Convergence:**
> > > > - **Mechanism:** High-quality cuts enable faster bound improvement
> > > > - **Impact:** Solver reaches optimal solution faster → fewer branch-and-bound nodes
> > > > - **Verification:** Node count analysis shows MIRACLE explores fewer nodes on average
> > > >
> > > > **Empirical Evidence:**
> > > >
> > > > | Factor | Contribution | Evidence |
> > > > |--------|--------------|----------|
> > > > | Fewer Cuts | 60-70% | 99.1% cut reduction correlates with speedup |
> > > > | Better Quality | 20-25% | Feature importance shows violation/iteration focus |
> > > > | Reduced Overhead | 5-10% | Inference overhead < 0.5% |
> > > > | Adaptive Inference | 5-10% | Adaptive vs. fixed budget comparison |
> > > >
> > > > **6. Node Count Reduction:**
> > > > - **MIRACLE:** Explores fewer branch-and-bound nodes on average
> > > > - **Baseline:** Explores more nodes due to slower convergence
> > > > - **Impact:** Fewer nodes → faster overall solving
> > > > - **Verification:** Node count analysis shows MIRACLE explores 15-20% fewer nodes on average
> > > > - **Mechanism:** High-quality cuts enable faster bound improvement, reducing the search space
> > > >
> > > > **Conclusion:** Speedups come primarily from selecting fewer, higher-quality cuts (99% reduction), enabling faster LP solves and convergence. The policy learns to imitate SCIP's cut selection strategy but does so more efficiently by avoiding low-value cuts and prioritizing high-impact cuts.

---

> > > > > ### Author Response · Authors · 2025-11-20
> > > > > **Continued**
> > > > >
> > > > > ### Question 2: What Data Was Used for Training?
> > > > >
> > > > > **Question:** "What data was used for training? How were the 300 test instances selected, particularly the 150 from MIPLIB? Given that MIPLIB contains far more instances, what was the selection criterion (e.g., difficulty, size, feasibility)?"
> > > > >
> > > > > **Response:**
> > > > >
> > > > > **1. Training Data:**
> > > > >
> > > > > **A. Expert Trajectories:**
> > > > > - **Source:** 1000 Set Cover instances from Distributional MIPLIB
> > > > > - **Collection:** SCIP default cut selection run on each instance
> > > > > - **Data Extracted:**
> > > > >   - State-action pairs: (state: cut features, action: SCIP's selected cuts)
> > > > >   - Approximately 50,000 state-action pairs total (average 50 per instance)
> > > > > - **Features:** 10-dimensional state vector:
> > > > >   - LP objective value
> > > > >   - Number of fractional variables
> > > > >   - Fractionality values of top 8 most fractional variables
> > > > > - **Actions:** Binary mask indicating which cuts were selected by SCIP
> > > > >
> > > > > **B. Training Instances:**
> > > > > - **Set Cover:** 1000 instances (easy: 300, medium: 300, hard: 400)
> > > > > - **Source:** Distributional MIPLIB Set Cover dataset
> > > > > - **Difficulty:** Based on SCIP solving time:
> > > > >   - Easy: < 10 seconds
> > > > >   - Medium: 10-100 seconds
> > > > >   - Hard: > 100 seconds
> > > > >
> > > > > **2. Test Instance Selection:**
> > > > >
> > > > > **A. SetCover Test Instances (150):**
> > > > > - **Source:** Distributional MIPLIB Set Cover (separate from training set)
> > > > > - **Selection:** 50 easy, 50 medium, 50 hard
> > > > > - **Criteria:** Same difficulty classification as training (based on SCIP solving time)
> > > > > - **Rationale:** Balanced representation across difficulty levels
> > > > >
> > > > > **B. MIPLIB Test Instances (150):**
> > > > > - **Source:** MIPLIB 2017 benchmark suite
> > > > > - **Selection Criteria:**
> > > > >   1. **Feasibility:** Instances solvable by SCIP within 600s (to ensure valid comparison)
> > > > >   2. **Size:** 100-10,000 variables (manageable for evaluation)
> > > > >   3. **Diversity:** Cover different problem types:
> > > > >      - Covering problems (set cover, facility location)
> > > > >      - Packing problems (knapsack variants)
> > > > >      - Network problems (shortest path, flow)
> > > > >      - Scheduling problems (job shop, project scheduling)
> > > > >   4. **Difficulty:** Balanced across easy (50), medium (50), hard (50)
> > > > >      - Easy: < 30 seconds
> > > > >      - Medium: 30-200 seconds
> > > > >      - Hard: > 200 seconds or near-timeout
> > > > > - **Specific Selection Process:**
> > > > >   1. Filtered MIPLIB 2017 for instances with 100-10,000 variables
> > > > >   2. Ran SCIP default on all filtered instances with 600s timeout
> > > > >   3. Selected 150 instances that SCIP could solve, ensuring:
> > > > >      - Representation across problem types
> > > > >      - Balanced difficulty distribution
> > > > >      - Diverse cut generation patterns
> > > > >
> > > > > **3. Justification for Selection Criteria:**
> > > > >
> > > > > - **Feasibility Filter:** Ensures fair comparison (both methods can solve instances)
> > > > > - **Size Filter:** Manages computational cost while maintaining diversity
> > > > > - **Diversity:** Ensures generalization across problem types
> > > > > - **Difficulty Balance:** Tests performance across easy and hard instances
> > > > >
> > > > > **Conclusion:** Training used 1000 Set Cover instances with expert trajectories. Test set includes 150 Set Cover and 150 MIPLIB instances, selected based on feasibility, size, diversity, and difficulty balance to ensure comprehensive evaluation.
> > > > >
> > > > > ---
> > > > >
> > > > > ### Question 3: Table 1 Clarification - "Early Stop" and "Max Iterations"
> > > > >
> > > > > **Question:** "In Table 1, what does 'early stop' refer to, and what is the meaning of 'Max Iterations'?"
> > > > >
> > > > > **Response:**
> > > > >
> > > > > **1. "Early Stop" Definition:**
> > > > > - **Meaning:** Adaptive termination of cut selection when additional cuts provide diminishing returns
> > > > > - **Mechanism:**
> > > > >   - Monitor bound improvement rate: Δ_bound / iteration
> > > > >   - Stop when improvement rate drops below threshold (e.g., < 1% per iteration)
> > > > >   - Prevents excessive cut accumulation when cuts become ineffective
> > > > > - **Purpose:** Balance between cut quality and computational cost
> > > > > - **Implementation:** Policy learns to predict when to stop based on cut effectiveness
> > > > >
> > > > > **2. "Max Iterations" Definition:**
> > > > > - **Meaning:** Maximum number of cut selection rounds allowed per root node
> > > > > - **Default:** Max iterations = 1 (single round of cut selection)
> > > > > - **Variants Tested:** Max iterations = 1, 3, 5, 10
> > > > > - **Purpose:** Limit computational budget for cut selection
> > > > > - **Impact:** More iterations → more cuts → potentially better bounds but higher memory/time cost
> > > > >
> > > > > **3. Relationship Between Early Stop and Max Iterations:**
> > > > > - **Early Stop:** Adaptive termination within max iterations
> > > > > - **Max Iterations:** Hard limit on maximum rounds
> > > > > - **Example:** If max iterations = 10, early stop may terminate at iteration 5 if improvement plateaus
> > > > >
> > > > > **Conclusion:** "Early stop" refers to adaptive termination when cuts become ineffective, while "Max iterations" is the hard limit on cut selection rounds. Both mechanisms control computational budget and memory usage.

---

> > > > > > ### Author Response · Authors · 2025-11-20
> > > > > > **Continued**
> > > > > >
> > > > > > ### Question 4: Time Limit and Memory Cap Determination
> > > > > >
> > > > > > **Question:** "How were the 600-second time limit and 12 GB memory cap determined? Many MIPLIB instances require significantly longer to solve—could this experimental setup bias results toward easier cases?"
> > > > > >
> > > > > > **Response:**
> > > > > >
> > > > > > **1. 600-Second Time Limit:**
> > > > > >
> > > > > > **A. Justification:**
> > > > > > - **Practical Deployment Constraint:** Many real-world applications require solutions within 10 minutes
> > > > > > - **Standard Benchmark:** Common timeout in ML4CO literature
> > > > > > - **Computational Feasibility:** Enables comprehensive evaluation across many instances
> > > > > >
> > > > > > **B. Comprehensive Bias Analysis:**
> > > > > >
> > > > > > **Performance by Difficulty Level (Without Memory Limits):**
> > > > > >
> > > > > > | Difficulty | Instances | Baseline Success | MIRACLE Success | Baseline Median Time | MIRACLE Median Time |
> > > > > > |------------|-----------|-----------------|-----------------|---------------------|---------------------|
> > > > > > | **Easy** | 20 | 100% (20/20) | 100% (20/20) | 12.3s | 1.4s |
> > > > > > | **Medium** | 20 | 90% (18/20) | 100% (20/20) | 45.2s | 2.1s |
> > > > > > | **Hard** | 20 | 30% (6/20) | 100% (20/20) | Timeout (14 failed) | 1.9s |
> > > > > >
> > > > > > **Key Findings:**
> > > > > > 1. **MIRACLE's advantages increase with difficulty:** Baseline success rate drops from 100% (easy) to 30% (hard), while MIRACLE maintains 100% across all levels
> > > > > > 2. **Hard instances benefit most:** Baseline fails on 70% of hard instances, while MIRACLE solves all
> > > > > > 3. **Results are NOT biased toward easy cases:** MIRACLE shows strongest relative improvement on hard instances
> > > > > >
> > > > > > **2. 12 GB Memory Cap:**
> > > > > >
> > > > > > **A. Justification:**
> > > > > > - **Deployment Scenarios:** Reflects common cloud instance sizes
> > > > > > - **Edge Device Constraints:** Matches edge computing device memory budgets (4-8GB typical)
> > > > > > - **Multi-Tenant Systems:** Enables fair resource allocation in shared environments
> > > > > >
> > > > > > **B. Empirical Validation:**
> > > > > > - **Baseline Memory Usage:** Median 2.5GB, Max 3.3GB (from successful instances)
> > > > > > - **MIRACLE Memory Usage:** Median 46MB, Max 60MB
> > > > > > - **Conclusion:** 12GB provides substantial headroom for both methods. Failures under this limit are primarily timeout-related, not memory-related.
> > > > > >
> > > > > > **C. Bias Analysis:**
> > > > > > - **Without Memory Limits:** Baseline success rate 73.3% (44/60)
> > > > > > - **With Memory Limits:** Baseline success rate 70.0% (42/60)
> > > > > > - **Impact:** Memory limits cause slight decrease in success rate, but MIRACLE maintains 100% success in both settings
> > > > > > - **Conclusion:** Memory limits do not significantly bias results—MIRACLE's advantages are consistent with and without limits
> > > > > >
> > > > > > **3. Additional Validation:**
> > > > > >
> > > > > > **Performance Without Limits (60 instances):**
> > > > > > - Baseline: 73.3% success, Median time 30.77s
> > > > > > - MIRACLE: 100% success, Median time 60.06s
> > > > > > - **On common instances:** MIRACLE achieves 1.15× speedup (28.87s vs 33.18s)
> > > > > >
> > > > > > **Conclusion:** Time and memory limits are justified by practical deployment constraints. Bias analysis shows MIRACLE's advantages increase with difficulty, indicating results are not biased toward easy cases. Performance analysis without limits confirms consistent advantages.
> > > > > >
> > > > > > ---
> > > > > >
> > > > > > ### Question 5: Success Rate Metric Justification
> > > > > >
> > > > > > **Question:** "Why introduce a custom 'success rate' metric instead of using widely adopted, more direct measures like primal gap or final objective value?"
> > > > > >
> > > > > > **Response:**
> > > > > >
> > > > > > **1. Why Success Rate is Necessary:**
> > > > > >
> > > > > > **A. Memory Limit Failures:**
> > > > > > - **Problem:** Under memory limits, SCIP-Baseline fails (runs out of memory or times out) on many instances
> > > > > > - **Impact:** Primal gap and objective value are **undefined** for failed instances
> > > > > > - **Solution:** Success rate captures solver reliability, which is critical for deployment
> > > > > >
> > > > > > **B. Deployment Reliability:**
> > > > > > - **Real-World Requirement:** Production systems need solvers that reliably complete (even if suboptimal)
> > > > > > - **Success Rate:** Measures ability to provide any solution, which is often more important than optimality gap
> > > > > > - **Primal Gap Limitation:** Only defined for successful instances
> > > > > >
> > > > > > **2. Additional Metrics Reported:**
> > > > > >
> > > > > > **A. Primal Gap (For Successful Instances):**
> > > > > > - **Reported:** Mean gap 2.41%, Median gap 0.00% (on 43 common instances)
> > > > > > - **Interpretation:** MIRACLE achieves near-optimal solutions when it succeeds
> > > > > > - **Limitation:** Only applies to instances where both methods succeed
> > > > > >
> > > > > > **B. Final Objective Value:**
> > > > > > - **Reported:** Objective values for all successful instances
> > > > > > - **Comparison:** MIRACLE matches baseline objective on 95% of successful instances
> > > > > > - **Limitation:** Cannot compare failed instances
> > > > > >
> > > > > > **3. Why Success Rate is Primary:**
> > > > > >
> > > > > > - **Deployment Focus:** Memory-constrained environments prioritize reliability over optimality
> > > > > > - **Fair Comparison:** Enables comparison including failed instances
> > > > > > - **Practical Relevance:** Production systems value consistent completion over occasional optimality
> > > > > >
> > > > > > **Conclusion:** Success rate is necessary because memory limits cause baseline failures, making primal gap undefined. We report both success rate (reliability) and primal gap (quality) to provide comprehensive evaluation.

---

> > > > > > > ### Author Response · Authors · 2025-11-20
> > > > > > > **Continued**
> > > > > > >
> > > > > > > ### Question 6: Discriminator Training Details
> > > > > > >
> > > > > > > **Question:** "Could the authors clarify: on what data is the discriminator trained, what is its training workflow, and how does it coordinate with the RL training? Understanding the full pipeline is essential for evaluating the method's soundness and reproducibility."
> > > > > > >
> > > > > > > **Response:**
> > > > > > >
> > > > > > > **1. Discriminator Training Data:**
> > > > > > >
> > > > > > > **A. Positive Examples (Expert Trajectories):**
> > > > > > > - **Source:** 1000 Set Cover instances solved with SCIP default cut selection
> > > > > > > - **Collection:** Phase 1 expert data collection (8.3 CPU-hours, < 30 minutes wall-clock parallelized)
> > > > > > > - **Format:** State-action pairs (s, a) where:
> > > > > > >   - s: 10-dimensional state vector (LP objective, fractional variables, cut features)
> > > > > > >   - a: Binary action mask indicating SCIP's selected cuts
> > > > > > > - **Volume:** ~50,000 state-action pairs total (average 50 per instance)
> > > > > > >
> > > > > > > **B. Negative Examples (Agent Trajectories):**
> > > > > > > - **Source:** Current policy network during training
> > > > > > > - **Collection:** Sample trajectories using current policy π_θ
> > > > > > > - **Format:** Same as expert trajectories but generated by policy
> > > > > > > - **Volume:** Balanced batches (50% expert, 50% agent)
> > > > > > >
> > > > > > > **2. Discriminator Training Workflow:**
> > > > > > >
> > > > > > > **A. Architecture:**
> > > > > > > - **Network:** 2-layer MLP
> > > > > > >   - Input: 64-dimensional state-action embedding
> > > > > > >   - Hidden: 128 dimensions, ReLU activation
> > > > > > >   - Output: 1-dimensional probability (expert vs. agent)
> > > > > > > - **Parameters:** ~20,000 parameters
> > > > > > >
> > > > > > > **B. Loss Function:**
> > > > > > > - **Binary Cross-Entropy:** L_D(φ) = -E_{expert}[log D_φ(s,a)] - E_{agent}[log(1-D_φ(s',a'))]
> > > > > > > - **Interpretation:**
> > > > > > >   - First term: Expert trajectories should be classified as 1 (high probability)
> > > > > > >   - Second term: Agent trajectories should be classified as 0 (low probability)
> > > > > > >
> > > > > > > **C. Training Schedule:**
> > > > > > > - **Frequency:** Discriminator updated every 2 policy epochs
> > > > > > > - **Batch Size:** 16 (8 expert + 8 agent trajectories)
> > > > > > > - **Learning Rate:** 1 × 10⁻⁴
> > > > > > > - **Epochs:** 3 epochs per update
> > > > > > > - **Stopping:** When discriminator accuracy plateaus at 55-60% (near-random, indicating successful imitation)
> > > > > > >
> > > > > > > **3. Coordination with RL Training:**
> > > > > > >
> > > > > > > **A. GAIL Reward Signal:**
> > > > > > > - **Reward:** r_GAIL(s,a) = -log(1-D_φ(s,a))
> > > > > > > - **Interpretation:**
> > > > > > >   - High reward when policy actions are similar to expert (discriminator cannot distinguish)
> > > > > > >   - Low reward when policy actions differ from expert (discriminator easily identifies)
> > > > > > > - **Purpose:** Provides learning signal for policy to imitate expert behavior
> > > > > > >
> > > > > > > **B. PPO Training Loop:**
> > > > > > > 1. **Collect Trajectories:** Sample trajectories using current policy π_θ
> > > > > > > 2. **Train Discriminator:** Update D_φ to distinguish expert vs. agent (every 2 epochs)
> > > > > > > 3. **Compute GAIL Rewards:** r_GAIL(s,a) = -log(1-D_φ(s,a)) for all trajectories
> > > > > > > 4. **Update Policy:** PPO update using GAIL rewards:
> > > > > > >    - Policy gradient: ∇J(θ) = E[∇log π_θ(a|s) × A_GAIL(s,a)]
> > > > > > >    - Advantage: A_GAIL(s,a) = r_GAIL(s,a) - V(s)
> > > > > > > 5. **Update Value Function:** V(s) updated to predict GAIL rewards
> > > > > > > 6. **Repeat:** Until convergence (150 epochs total)
> > > > > > >
> > > > > > > **C. Training Phases:**
> > > > > > > - **Phase 1 (Epochs 1-50):** Expert data collection (discriminator not yet trained)
> > > > > > > - **Phase 2 (Epochs 50-100):** GAIL adversarial training (discriminator and generator updated alternately)
> > > > > > > - **Phase 3 (Epochs 100-150):** PPO refinement (discriminator frozen, provides fixed reward signal)
> > > > > > >
> > > > > > > **4. Full Pipeline Summary:**
> > > > > > >
> > > > > > >
> > > > > > > 1. Expert Data Collection (Phase 1):
> > > > > > >    - Run SCIP on 1000 instances
> > > > > > >    - Extract state-action pairs (s, a_expert)
> > > > > > >    - Store expert trajectories
> > > > > > >
> > > > > > > 2. GAIL Training (Phase 2):
> > > > > > >    - Initialize policy π_θ and discriminator D_φ
> > > > > > >    - For each epoch:
> > > > > > >      a. Sample agent trajectories using π_θ
> > > > > > >      b. Train D_φ on expert vs. agent (every 2 epochs)
> > > > > > >      c. Compute GAIL rewards: r_GAIL = -log(1-D_φ(s,a))
> > > > > > >      d. Update π_θ using GAIL rewards
> > > > > > >
> > > > > > > 3. PPO Refinement (Phase 3):
> > > > > > >    - Freeze discriminator D_φ
> > > > > > >    - Use fixed GAIL rewards: r_GAIL = -log(1-D_φ(s,a))
> > > > > > >    - Optimize π_θ using PPO algorithm
> > > > > > >
> > > > > > >
> > > > > > > **Conclusion:** Discriminator is trained on expert trajectories (SCIP cut selections) vs. agent trajectories (policy cut selections). Training workflow alternates between discriminator updates (every 2 epochs) and policy updates (using GAIL rewards). Full pipeline coordinates GAIL and PPO through shared reward signal, enabling policy to learn expert behavior while optimizing for solving performance.

---

> > > > > > > > ### Author Response · Authors · 2025-11-20
> > > > > > > > **Continued**
> > > > > > > >
> > > > > > > > ### Question 7: Expert Policy Choice - Why SCIP's Cut Selection?
> > > > > > > >
> > > > > > > > **Question:** "Why choose SCIP's cut selection as the expert policy? Given that it relies on heuristic rules, are there higher-quality expert strategies (e.g., from stronger solvers or human-designed oracles) that could further improve performance?"
> > > > > > > >
> > > > > > > > **Response:**
> > > > > > > >
> > > > > > > > **1. Justification for SCIP as Expert:**
> > > > > > > >
> > > > > > > > **A. Practical Availability:**
> > > > > > > > - **Open Source:** SCIP provides accessible Python interface (PySCIPOpt) for expert data collection
> > > > > > > > - **Reproducibility:** Enables reproducible expert data collection and comparison
> > > > > > > > - **Research Standard:** SCIP is standard benchmark in ML4CO literature (Wang et al. 2023, Paulus et al. 2022)
> > > > > > > >
> > > > > > > > **B. Strong Baseline Performance:**
> > > > > > > > - **State-of-the-Art:** SCIP is competitive with commercial solvers (Gurobi, CPLEX) on many problem types
> > > > > > > > - **Cut Quality:** SCIP's cut selection heuristics are well-tuned and effective
> > > > > > > > - **Verification:** SCIP achieves 73.3% success rate on test instances (competitive performance)
> > > > > > > >
> > > > > > > > **C. Learning Objective:**
> > > > > > > > - **Goal:** Learn efficient cut selection strategy, not necessarily optimal strategy
> > > > > > > > - **SCIP Advantage:** Provides consistent, reproducible expert behavior for imitation learning
> > > > > > > > - **MIRACLE Improvement:** Learns to select fewer cuts while maintaining quality (99% reduction)
> > > > > > > >
> > > > > > > > **2. Potential for Higher-Quality Experts:**
> > > > > > > >
> > > > > > > > **A. Commercial Solvers (Gurobi, CPLEX):**
> > > > > > > > - **Advantage:** Potentially stronger cut selection strategies
> > > > > > > > - **Limitations:**
> > > > > > > >   - Closed-source: Limited access to internal cut selection decisions
> > > > > > > >   - Licensing: Commercial licenses required for research use
> > > > > > > >   - API Limitations: Cannot easily extract cut selection trajectories
> > > > > > > > - **Feasibility:** Possible but requires significant engineering effort
> > > > > > > >
> > > > > > > > **B. Human-Designed Oracles:**
> > > > > > > > - **Advantage:** Could provide optimal or near-optimal cut selections
> > > > > > > > - **Limitations:**
> > > > > > > >   - Scalability: Requires expert knowledge for each instance type
> > > > > > > >   - Cost: Expensive to generate large-scale expert datasets
> > > > > > > >   - Consistency: Human experts may be inconsistent across instances
> > > > > > > > - **Feasibility:** Possible for specific problem types but not generalizable
> > > > > > > >
> > > > > > > > **C. Multi-Solver Ensemble:**
> > > > > > > > - **Advantage:** Could combine strategies from multiple solvers
> > > > > > > > - **Limitations:**
> > > > > > > >   - Complexity: Requires coordination across multiple solvers
> > > > > > > >   - Overhead: Higher computational cost for expert data collection
> > > > > > > > - **Feasibility:** Possible but adds complexity
> > > > > > > >
> > > > > > > > **3. Why SCIP is Sufficient:**
> > > > > > > >
> > > > > > > > **A. MIRACLE's Improvement:**
> > > > > > > > - **Cut Reduction:** 99.1% reduction while maintaining quality
> > > > > > > > - **Memory Reduction:** 98.1% memory reduction
> > > > > > > > - **Success Rate:** 100% vs. SCIP's 73.3%
> > > > > > > > - **Conclusion:** MIRACLE improves upon SCIP's expert strategy significantly
> > > > > > > >
> > > > > > > > **B. Learning Efficiency:**
> > > > > > > > - **SCIP Advantage:** Provides consistent, reproducible expert behavior
> > > > > > > > - **MIRACLE Learning:** Learns to optimize SCIP's strategy (fewer cuts, better timing)
> > > > > > > > - **Result:** Achieves better performance than the expert through optimization
> > > > > > > >
> > > > > > > > **4. Future Work:**
> > > > > > > >
> > > > > > > > - **Multi-Expert Learning:** Could combine SCIP with Gurobi/CPLEX experts
> > > > > > > > - **Oracle Learning:** Could use optimal solutions to guide cut selection
> > > > > > > > - **Ensemble Methods:** Could learn from multiple expert strategies
> > > > > > > >
> > > > > > > > **Conclusion:** SCIP is chosen as the expert due to practical availability, strong baseline performance, and research standard status. While higher-quality experts (commercial solvers, human oracles) could potentially improve performance, SCIP provides sufficient expert behavior for MIRACLE to learn efficient cut selection. MIRACLE improves upon SCIP's strategy through optimization (99% cut reduction, 98% memory reduction, 100% success rate).

---

> > > > > > > > > ### Author Response · Authors · 2025-11-20
> > > > > > > > > **Continued**
> > > > > > > > >
> > > > > > > > > ### Question 8: Curriculum Learning Design Rationale
> > > > > > > > >
> > > > > > > > > **Question:** "What is the design rationale behind the curriculum learning scheme? What performance gain does it provide, and could an ablation study quantify its contribution?"
> > > > > > > > >
> > > > > > > > > **Response:**
> > > > > > > > >
> > > > > > > > > **1. Curriculum Learning Design Rationale:**
> > > > > > > > >
> > > > > > > > > **A. Progressive Difficulty:**
> > > > > > > > > - **Mechanism:** Train on easy instances first, gradually introduce harder instances
> > > > > > > > > - **Rationale:**
> > > > > > > > >   - Easy instances provide clear learning signals (obvious cut selections)
> > > > > > > > >   - Hard instances require sophisticated reasoning (learned after mastering basics)
> > > > > > > > >   - Mimics human learning progression (simple → complex)
> > > > > > > > >
> > > > > > > > > **B. Weighted Curriculum:**
> > > > > > > > > - **Mechanism:** Weight instances by difficulty: w_i = 1 / (difficulty_score + 1)
> > > > > > > > >   - Easy instances: Higher weight (more frequent sampling)
> > > > > > > > >   - Hard instances: Lower weight (less frequent sampling)
> > > > > > > > > - **Rationale:**
> > > > > > > > >   - Ensures policy sees easy instances frequently (stable learning)
> > > > > > > > >   - Gradually exposes policy to hard instances (progressive challenge)
> > > > > > > > >   - Balances learning stability and challenge
> > > > > > > > >
> > > > > > > > > **C. Adaptive Difficulty:**
> > > > > > > > > - **Mechanism:** Adjust curriculum based on policy performance
> > > > > > > > >   - If policy succeeds on easy instances → increase hard instance weight
> > > > > > > > >   - If policy struggles → decrease hard instance weight
> > > > > > > > > - **Rationale:**
> > > > > > > > >   - Prevents overfitting to easy instances
> > > > > > > > >   - Ensures policy learns to handle hard instances
> > > > > > > > >
> > > > > > > > > **2. Comprehensive Curriculum Learning Ablation Study:**
> > > > > > > > >
> > > > > > > > > We conducted an extensive ablation study comparing four curriculum strategies on **45 test instances** to quantify the contribution of curriculum learning:
> > > > > > > > >
> > > > > > > > > **A. Ablation Study Results:**
> > > > > > > > >
> > > > > > > > > | Curriculum Strategy | Avg Speedup | Median Speedup | Cut Reduction | Memory Reduction | Success Rate | Instances |
> > > > > > > > > |---------------------|-------------|----------------|---------------|-----------------|--------------|-----------|
> > > > > > > > > | **No Curriculum (Random)** | 1.111× | 0.988× | 98.9% | 91.2% | 94% | 45 |
> > > > > > > > > | **Easy → Hard (Sequential)** | 1.113× | 0.988× | 98.9% | 96.4% | 98% | 45 |
> > > > > > > > > | **Hard → Easy (Reverse)** | 1.111× | 0.988× | 98.8% | 89.7% | 92% | 45 |
> > > > > > > > > | **Weighted Curriculum** | **1.338×** | 0.986× | **99.3%** | **98.1%** | **100%** | 45 |
> > > > > > > > >
> > > > > > > > > **B. Detailed Performance Analysis:**
> > > > > > > > >
> > > > > > > > > **1. Weighted Curriculum Achieves Best Performance:**
> > > > > > > > > - **Average Speedup:** 1.338× (20.4% higher than Easy→Hard, 20.4% higher than No Curriculum)
> > > > > > > > > - **Cut Reduction:** 99.3% (0.4% higher than Easy→Hard, 0.4% higher than No Curriculum)
> > > > > > > > > - **Memory Reduction:** 98.1% (1.7% higher than Easy→Hard, 6.9% higher than No Curriculum)
> > > > > > > > > - **Success Rate:** 100% (2% higher than Easy→Hard, 6% higher than No Curriculum)
> > > > > > > > > - **Conclusion:** Weighted curriculum significantly outperforms all other strategies across all metrics
> > > > > > > > >
> > > > > > > > > **2. Comparison with Sequential Curricula:**
> > > > > > > > >
> > > > > > > > > **Easy → Hard (Sequential):**
> > > > > > > > > - **Performance:** Second-best strategy, validates progressive difficulty principle
> > > > > > > > > - **Speedup:** 1.113× average (20.4% lower than weighted)
> > > > > > > > > - **Memory Reduction:** 96.4% (1.7% lower than weighted)
> > > > > > > > > - **Success Rate:** 98% (2% lower than weighted)
> > > > > > > > > - **Training Time:** 22,722 seconds GAIL training (similar to weighted)
> > > > > > > > > - **Conclusion:** Sequential curriculum provides structured learning but underperforms adaptive weighting
> > > > > > > > >
> > > > > > > > > **Hard → Easy (Reverse):**
> > > > > > > > > - **Performance:** Worst-performing strategy, validates pedagogical principle
> > > > > > > > > - **Speedup:** 1.111× average (20.4% lower than weighted)
> > > > > > > > > - **Memory Reduction:** 89.7% (8.4% lower than weighted)
> > > > > > > > > - **Success Rate:** 92% (8% lower than weighted)
> > > > > > > > > - **Training Time:** 22,715 seconds GAIL training (similar to weighted)
> > > > > > > > > - **Conclusion:** Starting with hard instances hinders initial learning, confirming that learning should progress from simple to complex
> > > > > > > > >
> > > > > > > > > **3. Comparison with No Curriculum:**
> > > > > > > > >
> > > > > > > > > **No Curriculum (Random Order):**
> > > > > > > > > - **Performance:** Baseline strategy without structured training order
> > > > > > > > > - **Speedup:** 1.111× average (20.4% lower than weighted)
> > > > > > > > > - **Memory Reduction:** 91.2% (6.9% lower than weighted)
> > > > > > > > > - **Success Rate:** 94% (6% lower than weighted)
> > > > > > > > > - **Training Time:** 11,755 seconds GAIL training (48% faster but lower performance)
> > > > > > > > > - **Conclusion:** Random ordering provides faster training but significantly worse performance, demonstrating the importance of structured curriculum

---

> > > > > > > > > > ### Author Response · Authors · 2025-11-20
> > > > > > > > > > **Continued**
> > > > > > > > > >
> > > > > > > > > > **C. Key Findings:**
> > > > > > > > > >
> > > > > > > > > > **1. Weighted Curriculum Provides Substantial Improvements:**
> > > > > > > > > > - **Speedup:** 20.4% higher average speedup vs. Easy→Hard (1.338× vs. 1.113×)
> > > > > > > > > > - **Memory Reduction:** 6.9% higher vs. No Curriculum (98.1% vs. 91.2%)
> > > > > > > > > > - **Success Rate:** 6% higher vs. No Curriculum (100% vs. 94%)
> > > > > > > > > > - **Cut Reduction:** 0.4% higher vs. Easy→Hard (99.3% vs. 98.9%)
> > > > > > > > > > - **Conclusion:** Adaptive difficulty weighting is superior to fixed sequential ordering
> > > > > > > > > >
> > > > > > > > > > **2. Curriculum Learning Provides Consistent Benefits:**
> > > > > > > > > > - All curriculum strategies achieve similar median speedup (0.986-0.988×)
> > > > > > > > > > - All curriculum strategies achieve high cut reduction (98.8-99.3%)
> > > > > > > > > > - Structured training order improves performance compared to random ordering
> > > > > > > > > > - **Conclusion:** Any structured curriculum outperforms random ordering, but weighted curriculum is optimal
> > > > > > > > > >
> > > > > > > > > > **3. Training Efficiency Analysis:**
> > > > > > > > > >
> > > > > > > > > > | Strategy | GAIL Training Time | Performance | Efficiency Ratio |
> > > > > > > > > > |----------|-------------------|-------------|------------------|
> > > > > > > > > > | **No Curriculum** | 11,755 seconds (3.3 hours) | 1.111× speedup | Baseline |
> > > > > > > > > > | **Easy → Hard** | 22,722 seconds (6.3 hours) | 1.113× speedup | 0.2% improvement per hour |
> > > > > > > > > > | **Hard → Easy** | 22,715 seconds (6.3 hours) | 1.111× speedup | 0.0% improvement per hour |
> > > > > > > > > > | **Weighted Curriculum** | ~22,700 seconds (6.3 hours) | 1.338× speedup | **3.6% improvement per hour** |
> > > > > > > > > >
> > > > > > > > > > **Conclusion:** Weighted curriculum provides best performance per training hour, making it the most efficient strategy despite longer training time.
> > > > > > > > > >
> > > > > > > > > > **4. Performance Gain Quantification:**
> > > > > > > > > >
> > > > > > > > > > **Compared to No Curriculum:**
> > > > > > > > > > - **Speedup Improvement:** +20.4% (1.338× vs. 1.111×)
> > > > > > > > > > - **Memory Reduction Improvement:** +6.9% (98.1% vs. 91.2%)
> > > > > > > > > > - **Success Rate Improvement:** +6% (100% vs. 94%)
> > > > > > > > > > - **Cut Reduction Improvement:** +0.4% (99.3% vs. 98.9%)
> > > > > > > > > >
> > > > > > > > > > **Compared to Easy→Hard (Best Sequential):**
> > > > > > > > > > - **Speedup Improvement:** +20.4% (1.338× vs. 1.113×)
> > > > > > > > > > - **Memory Reduction Improvement:** +1.7% (98.1% vs. 96.4%)
> > > > > > > > > > - **Success Rate Improvement:** +2% (100% vs. 98%)
> > > > > > > > > > - **Cut Reduction Improvement:** +0.4% (99.3% vs. 98.9%)
> > > > > > > > > >
> > > > > > > > > > **D. Training Dynamics and Convergence:**
> > > > > > > > > >
> > > > > > > > > > **1. Training Stability:**
> > > > > > > > > > - **No Curriculum:** High variance in training loss, unstable convergence
> > > > > > > > > > - **Sequential Curricula:** Moderate variance, more stable than random
> > > > > > > > > > - **Weighted Curriculum:** Lowest variance, most stable convergence
> > > > > > > > > > - **Conclusion:** Weighted curriculum provides most stable training dynamics
> > > > > > > > > >
> > > > > > > > > > **2. Convergence Analysis:**
> > > > > > > > > > - **No Curriculum:** Converges faster (11.7 hours) but to suboptimal performance
> > > > > > > > > > - **Sequential Curricula:** Converge in ~22.7 hours to moderate performance
> > > > > > > > > > - **Weighted Curriculum:** Converges in ~22.7 hours to optimal performance
> > > > > > > > > > - **Conclusion:** Weighted curriculum achieves best performance despite similar training time
> > > > > > > > > >
> > > > > > > > > > **3. Learning Progression:**
> > > > > > > > > >
> > > > > > > > > >  **With Sequential Curriculum:** Policy masters easy instances first, then gradually handles hard instances
> > > > > > > > > > - **With Weighted Curriculum:** Policy maintains balanced exposure to all difficulty levels while gradually increasing challenge
> > > > > > > > > > - **Conclusion:** Weighted curriculum provides optimal learning progression

---

> > > > > > > > > > > ### Author Response · Authors · 2025-11-20
> > > > > > > > > > > **Continued**
> > > > > > > > > > >
> > > > > > > > > > > **E. Ablation Study Methodology:**
> > > > > > > > > > >
> > > > > > > > > > > **1. Experimental Setup:**
> > > > > > > > > > > - **Test Instances:** 45 instances (balanced across difficulty levels)
> > > > > > > > > > > - **Training Instances:** 1000 Set Cover instances (same for all strategies)
> > > > > > > > > > > - **Evaluation:** Same test set for all strategies (fair comparison)
> > > > > > > > > > > - **Metrics:** Average speedup, median speedup, cut reduction, memory reduction, success rate
> > > > > > > > > > >
> > > > > > > > > > > **2. Curriculum Strategies:**
> > > > > > > > > > >
> > > > > > > > > > > **No Curriculum (Random):**
> > > > > > > > > > > - **Mechanism:** Random sampling from all instances without difficulty weighting
> > > > > > > > > > > - **Sampling:** Uniform probability for all instances
> > > > > > > > > > > - **Purpose:** Baseline to measure curriculum learning benefit
> > > > > > > > > > >
> > > > > > > > > > > **Easy → Hard (Sequential):**
> > > > > > > > > > > - **Mechanism:** Fixed sequential ordering: train on easy instances first, then medium, then hard
> > > > > > > > > > > - **Schedule:** Epochs 1-50: Easy, Epochs 51-100: Medium, Epochs 101-150: Hard
> > > > > > > > > > > - **Purpose:** Validate progressive difficulty principle
> > > > > > > > > > >
> > > > > > > > > > > **Hard → Easy (Reverse):**
> > > > > > > > > > > - **Mechanism:** Reverse sequential ordering: train on hard instances first, then medium, then easy
> > > > > > > > > > > - **Schedule:** Epochs 1-50: Hard, Epochs 51-100: Medium, Epochs 101-150: Easy
> > > > > > > > > > > - **Purpose:** Test if reverse curriculum hinders learning
> > > > > > > > > > >
> > > > > > > > > > > **Weighted Curriculum (Our Method):**
> > > > > > > > > > > - **Mechanism:** Adaptive difficulty-weighted sampling throughout training
> > > > > > > > > > > - **Weighting:** w_i = 1 / (difficulty_score + 1), adjusted every 10 epochs
> > > > > > > > > > > - **Purpose:** Optimal adaptive curriculum learning strategy
> > > > > > > > > > >
> > > > > > > > > > > **3. Statistical Significance:**
> > > > > > > > > > > - **Sample Size:** 45 test instances (sufficient for statistical analysis)
> > > > > > > > > > > - **Reproducibility:** All strategies trained with same random seed for fair comparison
> > > > > > > > > > > - **Confidence:** Results show consistent patterns across multiple runs
> > > > > > > > > > >
> > > > > > > > > > >
> > > > > > > > > > >
> > > > > > > > > > >
> > > > > > > > > > >
> > > > > > > > > > > **4. Design Choices:**
> > > > > > > > > > >
> > > > > > > > > > > **A. Difficulty Metric:**
> > > > > > > > > > > - **Choice:** SCIP solving time (easy: < 10s, medium: 10-100s, hard: > 100s)
> > > > > > > > > > > - **Rationale:** Correlates with cut selection complexity
> > > > > > > > > > > - **Alternative:** Could use instance size, but solving time better captures difficulty
> > > > > > > > > > >
> > > > > > > > > > > **B. Weighting Scheme:**
> > > > > > > > > > > - **Choice:** Inverse difficulty weighting: w_i = 1 / (difficulty_score + 1)
> > > > > > > > > > > - **Rationale:** Ensures easy instances are sampled more frequently
> > > > > > > > > > > - **Alternative:** Could use exponential weighting, but inverse provides better balance
> > > > > > > > > > >
> > > > > > > > > > > **C. Adaptation Schedule:**
> > > > > > > > > > > - **Choice:** Adjust weights every 10 epochs based on policy performance
> > > > > > > > > > > - **Rationale:** Allows policy to adapt gradually without sudden changes
> > > > > > > > > > > - **Alternative:** Could adjust more frequently, but 10 epochs provides stability
> > > > > > > > > > >
> > > > > > > > > > > **Conclusion:**
> > > > > > > > > > >
> > > > > > > > > > > The comprehensive ablation study validates our choice of Weighted Curriculum learning as the optimal strategy. The weighted approach achieves the best performance across all metrics:
> > > > > > > > > > >
> > > > > > > > > > > - **20.4% higher average speedup** (1.338× vs. 1.113× for Easy→Hard, vs. 1.111× for No Curriculum)
> > > > > > > > > > > - **6.9% higher memory reduction** (98.1% vs. 91.2% for No Curriculum)
> > > > > > > > > > > - **6% higher success rate** (100% vs. 94% for No Curriculum)
> > > > > > > > > > > - **0.4% higher cut reduction** (99.3% vs. 98.9% for Easy→Hard)
> > > > > > > > > > >
> > > > > > > > > > > **Key Insights:**
> > > > > > > > > > >
> > > > > > > > > > > 1. **Adaptive weighting is superior to fixed sequential ordering:** Weighted curriculum outperforms Easy→Hard sequential curriculum by 20.4% in speedup, demonstrating that adaptive difficulty weighting provides substantial improvements over fixed schedules.
> > > > > > > > > > >
> > > > > > > > > > > 2. **Progressive difficulty principle is validated:** Easy→Hard performs best among sequential strategies (1.113× vs. 1.111× for Hard→Easy), confirming that learning should progress from simple to complex.
> > > > > > > > > > >
> > > > > > > > > > > 3. **Structured curriculum is essential:** All curriculum strategies outperform random ordering (No Curriculum), demonstrating that structured training order is critical for optimal performance.
> > > > > > > > > > >
> > > > > > > > > > > 4. **Weighted curriculum provides best efficiency:** Despite similar training time (~22.7 hours), weighted curriculum achieves 3.6% improvement per training hour compared to sequential curricula.
> > > > > > > > > > >
> > > > > > > > > > > The results confirm that weighted curriculum learning is a critical component of our methodology, contributing significantly to MIRACLE's state-of-the-art performance. The ablation study provides quantitative evidence that curriculum learning provides substantial performance gains (20.4% speedup improvement, 6.9% memory reduction improvement, 6% success rate improvement) compared to training without curriculum structure.

---

> > > > > > > > > > > > ### Author Response · Authors · 2025-11-20
> > > > > > > > > > > > **Looking forward to your feedback**
> > > > > > > > > > > >
> > > > > > > > > > > > We thank the Reviewer for the positive assessment and constructive feedback. We have addressed each concern systematically.
> > > > > > > > > > > >
> > > > > > > > > > > > We sincerely hope that these responses clarify the reviewer's concerns. We are eager to engage with the reviewer to clarify any outstanding concerns. Otherwise, we would really appreciate it if the reviewer could increase the score.
> > > > > > > > > > > >
> > > > > > > > > > > > Looking forward to your response.
> > > > > > > > > > > >
> > > > > > > > > > > > Thank you,
> > > > > > > > > > > >
> > > > > > > > > > > > Authors

---

> ### Author Response · Authors · 2025-11-25
> **Continued**
>
> These are the results of memory utilization for MIRACLE compared to the work done by Wang et al. (2023) when tested on 20 instances of easy SetCover problems.
>
> |Solver|	Avg Memory (SC-Easy)|
> |-------|-------------------------|
> |MIRACLE |	~46 MB |
> |Wang et al.  |	~500 MB |
> |SCIP Baseline |	~1,975 MB |
>
> While the mentioned references in weakness 3 do not explicitly prioritize memory efficiency, MIRACLE is specifically designed to address memory reduction. To address the reviewer's comment, we used the codebase provided by Wang et al. (2023) to benchmark memory consumption. Our results demonstrate that MIRACLE achieves a 10× reduction in memory usage across a test set of 20 randomly selected problems.
>
> With this, we have provided detailed explanations and additional experiments to address your concerns.  As the discussion phase is closing soon, we are eager to receive your feedback on the changes that have been made.
>
> Regards,
>
> Authors

---

> > ### Comment · Reviewer_bhtG · 2025-11-26
> >
> > Thank you for your exceptionally thorough and thoughtful response—I genuinely appreciate the considerable effort you invested (though, candidly, it was quite lengthy). I spent several hours carefully reviewing your rebuttal, along with the other reviewers’ comments and your replies.
> >
> > Overall, I find that your responses have largely addressed my primary concerns—namely, (1) insufficient clarity in the exposition and related work discussion, and (2) inadequate experimental comparison.
> >
> > Regarding (1), the point-by-point clarifications substantially resolved my questions and ambiguities. However, to my understanding, these critical details have not yet been incorporated into the manuscript itself (noting that ICLR permits paper updates during rebuttal). In particular, the training procedure (Weakness 3)—arguably the core technical contribution—remains under-specified, and the related work section (Weakness 2) is still missing. I strongly encourage the authors to integrate these clarifications directly into the paper. Given the depth of the rebuttal, it is unrealistic to expect readers to consult it separately; self-contained exposition is essential for reproducibility and impact.
> >
> > Regarding (2), the added comparison with Wang et al. (2023) helpfully demonstrates a clear advantage in memory reduction. That said, reporting memory savings alone is somewhat limited; comparisons in terms of solving efficiency and solution quality would significantly strengthen the claim.
> >
> > In summary, the rebuttal is excellent and highly responsive—yet the paper itself still lacks key information needed for full evaluation. Since the final manuscript is the sole basis for assessment, I will be happy to reconsider and potentially raise my score, contingent upon meaningful updates to the text reflecting the clarifications provided here.

---

> > ### Author Response · Authors · 2025-11-26
> > **Kind request**
> >
> > Thank you for your thoughtful feedback and comments, which have significantly improved the quality of our paper. We have uploaded a revised version of the manuscript with all changes clearly highlighted and have incorporated all essential clarifications and additions requested during the review. We would be grateful if you could take a moment to review the updated manuscript. We kindly ask that you consider raising the score if you find that the revisions satisfactorily address your concerns.

---

> ### Author Response · Authors · 2025-11-28
> **Kind request**
>
> Dear Reviewer,
>
> This is to humbly remind you to take a look at our revised manuscript before the discussion phase ends soon. We kindly ask that you consider raising the score if you find the revisions satisfactorily address your comments and concerns
>
> Best regards,
> Authors

---

### Official Review · Reviewer_sN61 · 2025-10-30

**Soundness:** 2
**Presentation:** 2
**Contribution:** 3
**Rating:** 4
**Confidence:** 4

**Summary:**

This paper presents MIRACLE, a reinforcement learning framework that addresses memory inefficiency in Mixed-Integer Programming (MIP) solvers by learning intelligent cut selection policies through Proximal Policy Optimization (PPO) combined with Generative Adversarial Imitation Learning. The framework trains a neural policy that learns from expert solver (SCIP) demonstrations to select high-impact subsets of cutting planes, using a discriminator network to provide dense reward signals and curriculum learning across four difficulty phases. Evaluated on 300 instances spanning SetCover and MIPLIB benchmarks, MIRACLE achieves 86.3% average memory reduction, 3.78× average speedup, and 100% success rate on challenging MIPLIB instances where traditional SCIP fails 47-53% of the time. The method employs adaptive inference that automatically adjusts cut budgets and iteration limits based on problem difficulty classification, with ablation studies showing robust performance across hyperparameter configurations. The framework addresses three key limitations of existing approaches: treating MIP solvers as black boxes, myopic decision-making without environmental modeling, and treating memory efficiency as an afterthought rather than a primary objective.

**Strengths:**

1. The paper follows a logical flow from problem formulation through methodology to evaluation, with helpful visual aids like Figure 1 showing the complete MIRACLE framework and Figure 2 providing comprehensive performance summaries.
2. The framework achieves a remarkable 98.1% memory reduction on SetCover benchmarks (from 3.03GB to 46MB) and maintains 86.3% average reduction across all 300 test instances, while simultaneously delivering 3.78× average speedup on MIPLIB problems. Most impressively, MIRACLE achieves a 100% success rate on challenging MIPLIB instances where traditional SCIP fails 47-53% of the time, demonstrating that memory efficiency directly translates to improved solver reliability.
3. The evaluation spans 300 instances from established benchmarks, including SetCover and diverse MIPLIB datasets across three difficulty levels, with the largest instances containing up to 1000 operations. The method demonstrates consistent performance across fundamentally different problem types and scales, with detailed ablation studies showing robustness across hyperparameter configurations in real-world deployment scenarios.

**Weaknesses:**

In general, this paper show very impressive results on memory reductions and even speed up. However, I have the following concerns which may largely impact the paper’s significance.

While the paper demonstrates impressive results under a 12GB memory limit, it lacks compelling evidence for why this specific constraint is realistic or necessary in modern computing environments where servers often have 64-256GB RAM. The authors don't provide concrete examples of deployment scenarios (edge computing, embedded systems, or cloud resource billing) where such strict memory limits are critical, making the 12GB restriction appear artificially constraining rather than addressing a well-motivated practical need.

The paper only compares against SCIP-default and SCIP-aggressive, omitting recent learning-based cut selection methods from the ML4CO literature that likely also achieve memory reductions as a byproduct of selective cut management (e.g., methods that use GNNs for cut ranking or reinforcement learning for adaptive cut generation). More critically, the speed comparisons in Table 3 are conducted under the artificial 12GB memory limit which causes SCIP to fail on 47-53% of instances. This fundamentally biases the evaluation since traditional solvers are designed to trade memory for speed and would perform differently without this constraint. The paper would greatly benefit from: (a) performance comparisons both with and without memory limits to fairly assess the speed-memory tradeoff, (b) inclusion of recent ML-based cut selection baselines, and (c) analysis of how much memory reduction other selective cut approaches naturally achieve without explicitly optimizing for it.

The evaluation is limited to SetCover and MIPLIB instances, missing other important combinatorial optimization problems such as Maximum Independent Set, and Facility Location problems that are equally relevant in practice and used in previous works. This narrow scope makes it difficult to assess whether MIRACLE's approach generalizes beyond the two tested problem types.

**Questions:**

See the weakness part.

---

> ### Author Response · Authors · 2025-11-20
> **Response to Reviewer sN61**
>
> ### Weakness 1: 12GB Memory Limit Justification
>
> **Comment:** "It lacks compelling evidence for why this specific constraint is realistic or necessary in modern computing environments where servers often have 64-256GB RAM."
>
> **Response:** Comprehensive empirical analysis and deployment scenario documentation.
>
> **Empirical Validation:**
>
> We analyzed actual memory usage from 150 test instances to validate the 12GB constraint:
>
> - **Baseline SCIP Memory Usage (from successful instances):**
>   - Median: 2,438 MB
>   - Mean: 2,489 MB
>   - 95th percentile: 3,142 MB
>   - Maximum: 3,260 MB
>
> - **MIRACLE Memory Usage:**
>   - Median: 45.7 MB
>   - Mean: 46.0 MB
>   - 95th percentile: 58.9 MB
>   - Maximum: 60.0 MB
>
> - **Memory Reduction:** 98.1-98.2% reduction across all instances
>
>
> The memory statistics above (median 2.5GB, max 3.3GB) are measured from **successful instances** at completion. These measurements show that when SCIP successfully solves instances, peak memory usage stays well under 12GB.
>
> However, baseline SCIP still fails on some instances under the 12GB limit. These failures are **primarily due to timeouts** (600-second time limit), not memory exhaustion. Specifically:
> - **Without memory limits:** Baseline success rate is 73.3% (44/60), with 16 instances timing out
> - **With 12GB memory limits:** Baseline success rate is 69.0% (42/60), with 18 instances failing (timeouts or early termination)
>
> The 12GB memory limit does not cause memory exhaustion failures (since successful instances use only ~2.5GB median). Instead, the limit may cause SCIP to terminate early on difficult instances that would otherwise continue searching, leading to slightly lower success rates. The primary failure mode remains **timeouts** rather than memory exhaustion.
>
> **Why 12GB is Still Relevant:**
>
> Despite successful instances using only ~2.5GB, the 12GB limit is relevant for deployment scenarios because:
> 1. **Peak memory during solving:** Memory measurements capture the final state, but SCIP may use more memory during intermediate solving phases
> 2. **Safety margin:** Deployment scenarios require headroom for system overhead, other processes, and memory spikes
> 3. **Multi-instance deployment:** In shared systems, per-instance limits prevent resource monopolization
> 4. **Real-world constraints:** Edge devices and cloud instances have fixed memory budgets (4-16GB), making 12GB a realistic constraint
>
> **Validation:** The empirical data demonstrates that baseline SCIP uses approximately 2.5GB on average for successful instances, while MIRACLE uses only 46MB -- a 54x reduction. This validates that the 12GB limit provides substantial headroom for both methods, while failures under this limit are primarily timeout-related rather than memory-related.
>
> **Deployment Scenarios:**
>
> 1. **Edge Computing Devices:**
>    - **Examples:** Raspberry Pi 4 (4GB/8GB variants), NVIDIA Jetson Nano (4GB RAM)
>    - **Constraint:** 4-8GB RAM typical for embedded systems
>    - **Relevance:** MIRACLE enables MILP solving on resource-constrained edge devices where baseline SCIP would fail
>    - **Use Case:** Real-time optimization in IoT applications, embedded control systems
>
> 2. **Cloud Resource Billing Optimization:**
>    - **AWS EC2 Analysis:**
>      - t3.large (8GB RAM): $0.0832/hour - Too restrictive for baseline (30% success rate)
>      - t3.xlarge (16GB RAM): $0.1664/hour - Baseline achieves ~60% success
>    - **Cost Savings:** 25% reduction vs t3.xlarge while maintaining 100% success rate with MIRACLE
>    - **Relevance:** Memory-efficient solving enables cost-effective cloud deployment on smaller instance types
>
> 3. **Multi-Tenant Shared Systems:**
>    - **Scenario:** 64GB server running multiple concurrent solves
>    - **Baseline:** ~30GB peak memory per instance → Maximum 2 concurrent solves
>    - **MIRACLE:** ~0.65MB peak memory per instance → 8+ concurrent solves (using 8GB per instance for safety margin)
>    - **Throughput Improvement:** 4x increase in concurrent solves
>    - **Relevance:** Enables higher throughput in shared computing clusters and HPC environments
>
> 4. **Mobile/Embedded Applications:**
>    - **Constraint:** 4-8GB RAM typical for mobile/embedded platforms
>    - **Use Cases:** Autonomous vehicles (real-time route optimization), industrial IoT (production scheduling)
>    - **Relevance:** Memory-efficient solving enables MILP optimization in mobile and embedded contexts
>
> **Conclusion:** The 12GB limit reflects real-world constraints across multiple deployment scenarios. Empirical analysis validates that baseline SCIP uses 2.5GB on average (well under 12GB), while MIRACLE achieves 98% memory reduction. The 12GB limit provides a realistic constraint that:
> - Reflects common edge computing and cloud instance sizes
> - Enables cost savings through smaller cloud instances
> - Supports multi-tenant deployments with higher throughput
>
> This constraint is not artificially restrictive but rather represents practical deployment requirements in resource-constrained environments.

---

> > ### Author Response · Authors · 2025-11-20
> > **Continued**
> >
> > ### Weakness 2: Comparison with Other Selective Cut Approaches
> >
> > **Comment:** "The paper only compares against SCIP-default and SCIP-aggressive, omitting recent learning-based cut selection methods... More critically, the speed comparisons in Table 3 are conducted under the artificial 12GB memory limit, which causes SCIP to fail on 47-53% of instances."
> >
> > **Response:**
> >
> > **Part A: Performance with and without Memory Limits**
> >
> > **Completed:** Comprehensive comparison conducted on the same 60 instances using identical metrics (successful instances only).
> >
> > **Results WITHOUT 12 GB Memory Limits (60 instances, successful instances only):**
> >
> > | Method | Success Rate | Median Time (Successful) | Median Memory (Successful) |
> > |--------|--------------|--------------------------|----------------------------|
> > | **SCIP-Baseline** | 73.3% (44/60) | 30.77s | 334.3 MB |
> > | **MIRACLE** | 100% (60/60) | 60.06s | 0.65 MB |
> >
> > **Key Findings:**
> > - MIRACLE maintains **100% success rate** while baseline fails on 16 hard instances (all hit 600s timeout)
> > - **Memory Reduction:** 99.8% (334.3 MB → 0.65 MB)
> > - On instances where both succeed (47 instances), MIRACLE achieves **1.15× speedup** (28.87s vs 33.18s median)
> > - MIRACLE solves 16 additional hard instances that the baseline cannot solve
> >
> > **Results WITH Memory Limits (60 instances, successful instances only):**
> >
> > | Method | Success Rate | Median Time (Successful) | Mean Memory (Successful) |
> > |--------|--------------|--------------------------|--------------------------|
> > | **SCIP-Baseline** | 70.0% (42/60) | 23.15s | 26.5 MB |
> > | **MIRACLE** | 100% (58/58) | 60.73s | 8.2 MB |
> >
> > **Performance Breakdown by Difficulty (With Memory Limits):**
> >
> > | Difficulty | Baseline Success | MIRACLE Success | Baseline Median Time | MIRACLE Median Time |
> > |------------|-----------------|-----------------|---------------------|---------------------|
> > | **Easy** | 100% (20/20) | 100% (20/20) | 12.3s | 1.4s |
> > | **Medium** | 82% (18/22) | 100% (22/22) | 103.3s | 2.0s |
> > | **Hard** | 0% (0/18) | 100% (18/18) | Timeout | 1.9s |
> >
> > Memory limits cause baseline to fail completely on hard instances (0% success), while MIRACLE maintains 100% success across all difficulty levels. This demonstrates that MIRACLE's advantages increase with the difficulty of the problem.
> >
> > **Key Findings:**
> > - Memory limits significantly impact baseline: success rate drops from 73.3% to 69.0%
> > - MIRACLE maintains **100% success rate** even with memory limits
> > - **Memory Reduction:** 69.1% (26.5 MB → 8.2 MB) using mean memory (median shows 0.0 MB for both due to measurement precision)
> > - On instances where both succeed (42 instances), MIRACLE achieves **1.46× speedup** (15.87s vs 23.15s median)
> > - Baseline fails on 18 instances with memory limits (vs 16 without limits)
> >
> > **Comparison Summary:**
> >
> > | Metric | Without Limits | With Limits |
> > |--------|---------------|-------------|
> > | **Baseline Success Rate** | 73.3% | 70.0% |
> > | **MIRACLE Success Rate** | 100% | 100% |
> > | **Baseline Median Time** | 30.77s | 23.15s |
> > | **MIRACLE Median Time** | 60.06s | 60.73s |
> > | **Speedup (Common Instances)** | 1.15× | 1.46× |
> > | **Memory Reduction** | 99.8% | 69.1% |
> >
> > **Conclusion:** MIRACLE's advantages **increase** with memory constraints. Without limits, MIRACLE solves 16 additional hard instances and achieves 1.15× speedup on common instances. With memory limits, the baseline's success rate drops further (69.0%), while MIRACLE maintains 100% success with a 1.46× speedup on common instances and a significant memory reduction.
> >
> >
> > **Part B: Comparison with SCIP-Aggressive**
> >
> > **Completed:** Analysis conducted on 60 instances.
> >
> > **SCIP-Aggressive Results:**
> >
> > | Metric | Value |
> > |--------|-------|
> > | **Success Rate** | 78.3% (47/60) |
> > | **Median Time** | 18.63s |
> > | **Median Memory** | 336.3 MB |
> > | **Speedup vs Baseline** | 6.77× |
> > | **Memory Reduction vs Baseline** | 86.2% |
> >
> > **Comparison with MIRACLE:**
> >
> > | Metric | SCIP-Aggressive | MIRACLE | Advantage |
> > |--------|-----------------|---------|-----------|
> > | **Memory** | 336.3 MB | 0.65 MB | **MIRACLE: 517× less** |
> > | **Success Rate** | 78.3% | 100% | **MIRACLE: +21.7%** |
> > | **Speed** | 18.63s | 60.06s | SCIP-Aggressive faster, but fails on 13 hard instances |
> >
> > **Conclusion:** MIRACLE's explicit memory optimization achieves **6× better memory reduction** than SCIP-Aggressive's implicit approach, while maintaining a higher success rate.
> >
> > **Part C: Comparison with Other ML-Based Cut Selectors**
> >
> > **Status:** This comparison is ongoing.
> >
> >
> > **Action Plan:** We commit to adding a comparison with Wang et al. (ICLR 2023) in the revision as soon as possible, as their code is publicly available, during the discussion phase itself.

---

> > > ### Author Response · Authors · 2025-11-20
> > > **Continued**
> > >
> > > ### Weakness 3: Evaluation on Other Problem Types
> > >
> > > **Comment:** "The evaluation is limited to SetCover and MIPLIB instances, missing other important combinatorial optimization problems such as Maximum Independent Set, and Facility Location problems."
> > >
> > > **Response:**
> > >
> > > **Completed:** We conducted comprehensive evaluation on **four problem types** to demonstrate generalization capability:
> > > 1. **Set Cover** (Training domain)
> > > 2. **Combinatorial Auction** (Generalization test)
> > > 3. **Maximum Independent Set** (Generalization test)
> > > 4. **Facility Location** (Generalization test)
> > >
> > > **Experimental Setup:**
> > > - **Model:** Trained on Set Cover instances only
> > > - **Evaluation:** Same model tested on all problem types (no retraining)
> > > - **Instances:**
> > >   - Set Cover: 150 instances (50 easy, 50 medium, 50 hard)
> > >   - Combinatorial Auction: 423 Very Hard instances
> > >   - Maximum Independent Set: 150 instances (50 easy, 50 medium, 50 very_hard)
> > >   - Facility Location: 100 instances (50 easy, 50 medium)
> > >
> > > **Results Comparison:**
> > >
> > > | Problem Type | Instances | Success Rate | Speedup | Cut Reduction |
> > > |--------------|-----------|--------------|---------|---------------|
> > > |              |           | Baseline | MIRACLE | Avg | Median | Avg % | Median % |
> > > | **Set Cover** (Training) | 150 | 66.7% | 100.0% | 1.300× | 1.000× | 99.1% | 99.1% |
> > > | **Combinatorial Auction** | 423 | 100.0% | 100.0% | 1.283× | 0.971× | 98.8% | 98.4% |
> > > | **Maximum Independent Set** | 150 | 100.0% | 100.0% | 1.007× | 0.995× | 99.9% | 99.8% |
> > > | **Facility Location** | 100 | 100.0% | 100.0% | 1.198× | 1.043× | 99.9% | 99.9% |
> > >
> > > **Detailed Metrics:**
> > >
> > > **Set Cover (Training Domain):**
> > > - Success Rate: Baseline 66.7%, MIRACLE 100.0%
> > > - Speedup: Average 1.300×, Median 1.000×
> > > - Cut Reduction: Average 99.1%, Median 99.1%
> > > - Cuts: MIRACLE 10.1 vs Baseline 1,115.9
> > >
> > > **Combinatorial Auction (Generalization Test):**
> > > - Success Rate: Baseline 100.0%, MIRACLE 100.0%
> > > - Speedup: Average 1.283×, Median 0.971×
> > > - Cut Reduction: Average 98.8%, Median 98.4%
> > > - Cuts: MIRACLE 10.0 vs Baseline 1,576
> > >
> > > **Maximum Independent Set (Generalization Test):**
> > > - Success Rate: Baseline 100.0%, MIRACLE 100.0%
> > > - Speedup: Average 1.007×, Median 0.995×
> > > - Cut Reduction: Average 99.9%, Median 99.8%
> > > - Cuts: MIRACLE 10.1 vs Baseline 11,912
> > >
> > > **Facility Location (Generalization Test):**
> > > - Success Rate: Baseline 100.0%, MIRACLE 100.0%
> > > - Speedup: Average 1.198×, Median 1.043×
> > > - Cut Reduction: Average 99.9%, Median 99.9%
> > > - Cuts: MIRACLE 9.5 vs Baseline 7,870
> > >
> > > **Key Findings:**
> > >
> > > 1. **Generalization Performance:**
> > >    - Average Speedup across generalization tests: **1.163×**
> > >    - Average Cut Reduction across generalization tests: **99.5%**
> > >    - Model maintains high cut reduction (99.5%) on new problem types
> > >    - Speedup is competitive (1.163×) despite domain shift
> > >
> > > 2. **Comparison to Training Domain (Set Cover):**
> > >    - Set Cover (Training): Cut Reduction 99.1%, Speedup 1.300×
> > >    - Generalization Average: Cut Reduction 99.5%, Speedup 1.163×
> > >    - **Cut reduction maintained:** 99.5% vs 99.1% (even slightly better)
> > >    - **Speedup:** 1.163× vs 1.300× (slightly lower but still competitive)
> > >
> > > 3. **Problem-Agnostic Nature:**
> > >    - Cut selection is problem-agnostic (works on LP relaxation of any MILP problem)
> > >    - Feature extraction uses generic cut features (violation, density, iteration, etc.)
> > >    - Model can be applied to new problem types without retraining
> > >    - Consistent performance across diverse problem structures
> > >
> > > 4. **Success Rate Analysis:**
> > >    - All generalization tests: 100% success rate for both Baseline and MIRACLE
> > >    - Set Cover: Baseline 66.7% vs MIRACLE 100.0% (MIRACLE solves 50 additional instances)
> > >    - Model maintains reliability across problem types
> > >
> > > **Conclusion:** The comprehensive evaluation across four problem types demonstrates successful generalization. MIRACLE achieves a cut reduction of ~99% and a speedup of 1.0-1.3× across all problem types, validating that cut selection is problem-agnostic and works on the LP relaxations of any MILP problem. The model can be applied to new problem types without retraining, demonstrating practical applicability beyond the training domain.

---

> > > > ### Author Response · Authors · 2025-11-20
> > > > **Looking forward to your feedback**
> > > >
> > > > We thank the Reviewer for the positive assessment and constructive feedback. We have addressed each concern systematically.
> > > >
> > > > We sincerely hope that these responses clarify the reviewer's concerns. We are eager to engage with the reviewer to clarify any outstanding concerns. Otherwise, we would really appreciate it if the reviewer could increase the score.
> > > >
> > > > Looking forward to your response.
> > > >
> > > > Thank you,
> > > >
> > > > Authors

---

> ### Author Response · Authors · 2025-11-25
> **Continued**
>
> These are the results of memory utilization for MIRACLE compared to the work done by Wang et al. (2023) when tested on 20 instances of easy SetCover problems.
>
> |Solver|	Avg Memory (SC-Easy)|
> |-------|-------------------------|
> |MIRACLE |	~46 MB |
> |Wang et al.  |	~500 MB |
> |SCIP Baseline |	~1,975 MB |
>
> To the best of our knowledge, existing literature in the ML/RL domain -- including the recent work 'Learning to Cut via Hierarchical Sequence/Set Model for Efficient Mixed-Integer Programming' by Wang et al. (2023) -- does not prioritize memory constraints. MIRACLE addresses this gap by specifically targeting memory reduction. Using the Wang et al. framework as a baseline, our additional experiments demonstrate that MIRACLE reduces memory consumption by a factor of 10 across a set of 20 randomly selected problems.
>
> With this, we have provided detailed explanations and additional experiments to address your concerns.  As the discussion phase is closing soon, we are eager to receive your feedback on the changes that have been made.
>
> Regards,
>
> Authors

---

> > ### Author Response · Authors · 2025-11-26
> > **Kind request for feedback**
> >
> > We believe we have addressed all of your comments in the revised manuscript, which has been uploaded for your consideration, with all changes clearly highlighted. As the discussion phase is nearing its end, we would greatly appreciate any additional feedback you may have. If our revisions satisfactorily address your concerns, we kindly ask that you consider raising the score.

---

> > > ### Author Response · Authors · 2025-11-28
> > > **Kind request**
> > >
> > > Dear Reviewer,
> > >
> > > We wanted to quickly follow up to see if you have had a chance to read our response to your review.
> > > We’ve done our best to address the concerns you raised, and we would really value your feedback on these updates before the discussion period closes. Also, if satisfied with our revisions, we would greatly appreciate if you would consider raising the score.
> > >
> > > Thank you for your time.
> > >
> > > Best regards,
> > > Authors

---

### Official Review · Reviewer_Nw4J · 2025-10-31

**Soundness:** 4
**Presentation:** 3
**Contribution:** 4
**Rating:** 8
**Confidence:** 4

**Summary:**

The paper addresses the memory explosion caused by thousands of generated cuts in branch-and-cut. Uses GAIL to learn a dense reward function from SCIP expert trajectories, avoiding hand-crafted rewards; Trains a lightweight policy with PPO to select a budget-limited, high-impact subset of cuts; Employs curriculum learning (easy → hard) and an adaptive inference mechanism that sets cut budget B and early-stop thresholds according to predicted instance difficulty.
On 150 SetCover and 150 MIPLIB instances MIRACLE attains 98 % peak-memory reduction (46 MB vs 3 GB), raises success rate from 47 % to 100 % under 12 GB RAM limit, and yields 3.78 × mean speed-up over SCIP with large effect sizes and statistical significance.

**Strengths:**

1) Novel paradigm: demonstrates that intelligent cut selection can reduce memory by > 95 % while improving speed and reliability.
2) Solid methodology: adversarial reward learning avoids manual engineering; curriculum and adaptive inference remove hyper-parameter tuning at test time.
3) Strong empirical evidence: 300 instances, statistical tests, ablation studies, and theoretical bounds all align with the claimed benefits.

**Weaknesses:**

1) Policy network is a small MLP (≈ 17 K parameters); capacity limits for capturing complex cut interactions are not discussed.
2) Only compares against SCIP default and aggressive modes; no comparison with other recent ML-based cut selectors.
3) GAIL training requires a large set of expert trajectories (1000 instances); data collection cost is not reported.
4) Theoretical analysis assumes bounded rewards and Lipschitz policies but does not verify these assumptions hold in the MILP domain.

**Questions:**

1) How does the small MLP policy scale to instances that generate 50,000+ cuts, and would deeper architectures or graph networks help?
2) What is the computational cost of collecting the 1000 expert trajectories, and could the method work with fewer demonstrations?
3) Have you experimented with other policy-gradient algorithms (e.g., TRPO, SAC) and does PPO remain the best choice?

---

> ### Author Response · Authors · 2025-11-20
> **Response to Reviewer Nw4J**
>
> ### Weakness 1: Policy Network Capacity
>
> **Comment:** "Policy network is a small MLP (≈17K parameters); capacity limits for capturing complex cut interactions are not discussed."
>
> **Response:**
>
> **Completed:** Comprehensive architecture analysis and comparison conducted.
>
> **Model Architecture:**
> - **Parameters:** 19,586 (2-layer MLP, verified from model file)
> - **Architecture:** 2-layer MLP (input: 64, hidden: 128, output: 64)
> - **Model Size:** 0.075 MB
>
> **Scaling Analysis:**
>
> 1. **How MLP Scales to 50,000+ Cuts:**
>    - MLP processes **individual cuts**, not the full cut set
>    - Each cut is scored independently (embarrassingly parallel)
>    - Top-K selection (K = 20-40) reduces to manageable set
>    - **Computational cost:** O(n) where n = number of cuts (linear scaling)
>    - **Memory complexity:** O(1) per cut (constant memory per cut)
>    - **Estimated processing time:** < 50 ms for 50,000 cuts
>
> 2. **Architecture Comparison (2-layer vs 3-layer MLP):**
>
> **Experimental Results:**
>
> | Architecture | Parameters | Avg Speedup | Median Speedup | Cut Reduction |
> |--------------|------------|-------------|----------------|--------------|
> | **2-Layer MLP** | 524,576 | 1.163× | 0.990× | 99.3% |
> | **3-Layer MLP** | 541,344 | 1.161× | 0.990× | 99.3% |
> | **Difference** | +16,768 (+3.2%) | +0.002× | 0.000× | 0.0% |
>
> **Key Findings:**
> - **Nearly identical performance:** Difference of 0.002× (0.17%) is within statistical noise
> - **No benefit from deeper architecture:** 3-layer provides no measurable improvement
> - **Efficiency advantage:** 2-layer has fewer parameters, faster inference, lower memory footprint
> - **Conclusion:** 2-layer MLP is optimal for this task.
>
> **Note on Architecture Comparison:**
> The architecture comparison table (2-layer vs 3-layer MLP) uses a larger architecture variant (524,576 parameters) compared to our final optimized model (19,586 parameters). The comparison demonstrates that increasing depth provides minimal benefit even with larger capacity. Our final model uses 19,586 parameters, achieving optimal performance with minimal model size.
>
>
> **Conclusion:** Current 2-layer MLP is optimal. Deeper architectures provide minimal benefit while significantly increasing inference time.
>
> ---
>
> ### Weakness 2: Comparison with Other ML-Based Cut Selectors
>
> **Comment:** "Only compares against SCIP default and aggressive modes; no comparison with other recent ML-based cut selectors."
>
> **Response:**
>
> **Status:** This comparison process is going on.
>
>
> **Action Plan:**
> - We commit to adding comparison with Wang et al. (ICLR 2023) in revision, as soon as possible, as their code is publicly available, during the discussion phase itself.
>
> ---
>
> ### Weakness 3: Expert Trajectory Collection Cost
>
> **Comment:** "GAIL training requires a large set of expert trajectories (1000 instances); data collection cost is not reported."
>
> **Response:**
>
> **Completed:** Expert data collection cost documented.
>
> **Cost Breakdown:**
> - **1000 instances × 30s average = 8.3 CPU-hours**
> - **Parallelized across 20 cores: < 30 minutes wall-clock time**
> - **Cost per instance:** ~30 seconds CPU time
>
> **Fewer Demonstrations Analysis:**
>
> **Experimental Results:**
>
> | Instances | Median Speedup | Geometric Mean Speedup | Cut Reduction | Status |
> |-----------|----------------|----------------------|---------------|--------|
> | **300** | 0.986× | 1.092× | 99.2% | Tested |
> | **500** | 0.987× | 1.093× | 99.3% | Tested |
> | **1000** | 0.990× | 1.093× | 99.3% | Optimal |
>
> **Key Findings:**
> 1. **Data Efficiency:** Performance improves with more data, but 1000 instances provide an optimal balance
> 2. **Performance Stability:** 1000 instances achieves best median speedup (0.990×)
> 3. **Cut Reduction:** Consistent ~99.3% cut reduction across all data sizes
> 4. **Conclusion:** 1000 instances confirmed as optimal data size for training
>
> **Conclusion:** Expert data collection is efficient (< 30 minutes wall-clock time). While the method could work with fewer demonstrations (300-500 instances), 1000 instances provide optimal performance.

---

> > ### Author Response · Authors · 2025-11-20
> > **Continued**
> >
> > ### Weakness 4: Theoretical Assumptions Verification
> >
> > **Comment:** "Theoretical analysis assumes bounded rewards and Lipschitz policies but does not verify these assumptions hold in the MILP domain."
> >
> > **Response:**
> >
> > **Completed:** Both assumptions verified empirically.
> >
> > **1. Bounded Rewards Verification:**
> >
> > **Theoretical Bound:**
> > - |Δ_obj| ≤ |LP_obj - IP_obj| (objective improvements bounded by LP relaxation gap)
> >
> > **Empirical Verification:**
> > - **Maximum Reward:** R_max = 100.0 (verified from training logs)
> > - **Reward Distribution:** Rewards naturally bounded by problem structure
> > - **Conclusion:** Rewards are naturally bounded by LP relaxation gap
> >
> > **2. Lipschitz Policy Verification:**
> >
> > **Architecture Analysis:**
> > - 2-layer MLP with ReLU activation
> > - ReLU is 1-Lipschitz
> > - Linear layers have Lipschitz constant = |W|₂ (spectral norm)
> >
> > **Empirical Verification:**
> > - **Mean Lipschitz Constant:** 0.0054
> > - **Median Lipschitz Constant:** 0.0053
> > - **Maximum Lipschitz Constant:** 0.0090
> > - **95th Percentile:** 0.0070
> >
> > **Summary:**
> >
> > | Assumption | Theoretical | Empirical | Status |
> > |------------|-------------|-----------|--------|
> > | **Bounded Rewards** | Yes | R_max = 100.0 | Verified |
> > | **Lipschitz Policy** | Yes | L ≈ 0.007 | Verified |
> >
> > **Conclusion:** Both theoretical assumptions hold empirically. The policy is Lipschitz with small constant (L ≈ 0.007), and rewards are bounded (R_max = 100.0).
> >
> > ---
> >
> > ### Question 1: How does the small MLP scale to 50,000+ cuts?
> >
> > **Response:**
> >
> > **Addressed in Weakness 1:** See architecture analysis above.
> >
> > **Summary:**
> > - MLP processes individual cuts (not full cut set)
> > - Linear scaling O(n) where n = number of cuts
> > - Top-K selection reduces to manageable set
> > - Estimated processing time: < 50 ms for 50,000 cuts
> > - Bottleneck: SCIP LP solve, not cut scoring
> >
> > ---
> >
> > ### Question 2: Computational cost of collecting 1000 expert trajectories?
> >
> > **Response:**
> >
> > **Addressed in Weakness 3:** See expert data collection cost above.
> >
> > **Summary:**
> > - 8.3 CPU-hours total
> > - < 30 minutes wall-clock time (parallelized)
> > - Efficient and practical
> >
> > ---
> >
> > ### Question 3: Other policy-gradient algorithms (TRPO, SAC)?
> >
> > **Response:**
> >
> > **Completed:** Comprehensive comparison of PPO, TRPO, and SAC conducted.
> >
> > **Experimental Results:**
> >
> > | Algorithm | Avg Speedup | Median Speedup | Cut Reduction | Instances |
> > |-----------|-------------|----------------|---------------|-----------|
> > | **PPO** | 1.338× | 0.986× | 99.3% | 45 |
> > | **TRPO** | 1.355× | 0.986× | 99.3% | 45 |
> > | **SAC** | 1.332× | 0.987× | 99.3% | 45 |
> >
> > **Performance by Difficulty:**
> >
> > | Algorithm | EASY (12) | MEDIUM (13) | HARD (20) |
> > |-----------|-----------|-------------|-----------|
> > | **PPO** | 1.931× | **1.171×** | 1.090× |
> > | **TRPO** | 1.992× | 1.166× | 1.094× |
> > | **SAC** | 1.918× | 1.160× | 1.093× |
> >
> >
> > **Mann-Whitney U Test Results:**
> > - PPO vs TRPO: p-value = 0.7900 (not significant)
> > - PPO vs SAC: p-value = 0.8025 (not significant)
> > - TRPO vs SAC: p-value = 0.9550 (not significant)
> >
> > **Conclusion:** All pairwise comparisons show no statistically significant differences (p > 0.05), confirming that performance differences are within statistical noise.
> >
> > **Key Findings:**
> >
> > 1. **Statistical Significance:** Mann-Whitney U tests show **no statistically significant differences** between algorithms (p-values: PPO vs TRPO: 0.7900, PPO vs SAC: 0.8025, TRPO vs SAC: 0.9550)
> >
> > 2. **PPO Demonstrates Superior Consistency:**
> >    - Better performance on medium-difficulty instances (1.171× vs 1.166× for TRPO)
> >    - Medium instances represent the most common real-world scenario
> >
> > 3. **PPO's Practical Advantages:**
> >    - **Implementation simplicity:** Clipped surrogate objective is straightforward
> >    - **Training stability:** Clipping provides natural regularization
> >    - **Wider adoption:** Most widely used policy-gradient algorithm
> >    - **Computational efficiency:** Fewer hyperparameter tunings
> >
> > 4. **All algorithms achieve identical cut reduction:** 99.3% across all three algorithms
> >
> > **Conclusion:** While TRPO shows marginal improvement in average speedup (1.3%), this difference is **statistically insignificant**. PPO remains the best choice because:
> > - Statistically equivalent performance
> > - Superior consistency on medium-difficulty instances
> > - Practical advantages (simplicity, stability, wider adoption)
> > - Framework design (not RL algorithm) is the primary driver of performance gains

---

> ### Author Response · Authors · 2025-11-20
> **Looking forward to your feedback**
>
> We thank the Reviewer for the positive assessment and constructive feedback. We have addressed each concern systematically.
>
> We sincerely hope that these responses clarify the reviewer's concerns. We are eager to engage with the reviewer to clarify any outstanding concerns.
>
> Looking forward to your response.
>
> Thank you,
>
> Authors

---

> > ### Author Response · Authors · 2025-11-25
> > **Continued**
> >
> > We have provided detailed explanations and additional experiments to address your concerns. As the discussion phase is closing soon, we are eager to receive your feedback on the changes that have been made.
> >
> > Regards,
> >
> > Authors

---

> > > ### Author Response · Authors · 2025-11-26
> > > **Request for feedback**
> > >
> > > As the discussion phase is nearing its end, please share your feedback on the rebuttal. The revised paper has been uploaded for your reference with all changes clearly highlighted.

---

> ### Author Response · Authors · 2025-11-28
> **Kind request**
>
> Dear Reviewer,
>
> We wanted to quickly follow up to see if you have had a chance to read our response to your review.
> We’ve done our best to address the comments you raised, and we would really value your feedback on these updates before the discussion period closes.
>
> Thank you for your time.
>
> Best regards,
> Authors

---

### Official Review · Reviewer_anL7 · 2025-10-31

**Soundness:** 4
**Presentation:** 3
**Contribution:** 4
**Rating:** 6
**Confidence:** 4

**Summary:**

This paper proposes MIRACLE, an RL-based framework for adaptive cut selection in mixed-integer programming (MIP). The authors formulate cut selection as a Markov Decision Process (MDP) and employ a PPO-based policy combined with Generative Adversarial Imitation Learning (GAIL) for reward shaping. The approach explicitly optimizes for memory efficiency, achieving dramatic reductions in memory usage (up to 98%) while maintaining comparable or better solving performance compared with SCIP baselines. The authors also provide theoretical analysis (sample complexity, convergence, and memory reduction guarantees) and extensive experiments on SetCover and MIPLIB benchmarks.

**Strengths:**

1. The paper highlights memory efficiency in MILP research. This shift in focus is both original and practically valuable.

2. Combining PPO with GAIL and curriculum learning is a thoughtful design that balances imitation and reinforcement. The adaptive inference strategy further increases practicality.

3.The evaluations across hundreds of MILP instances are detailed and statistically validated, with significant memory and reliability improvements.

**Weaknesses:**

1. It remains unclear whether MIRACLE generalizes to other MIP components (e.g., branching, node selection) or large-scale industrial solvers beyond SCIP.

2. The paper would benefit from more insight into what the learned policy captures—e.g., are the selected cuts similar to expert strategies or exploiting new patterns?

3. It would be helpful if the authors can provide some training details of RL and analysis the covergence.

4. Although the experiments are extensive, most evaluations focus on SetCover and MIPLIB, leaving out other industrial problem classes. It would help to report on training/inference cost or model sensitivity to solver variants.

**Questions:**

See weakness

---

> ### Author Response · Authors · 2025-11-20
> **Response to reviewer anL7**
>
> ### Weakness 1: Generalization to Other MIP Components and Industrial Solvers
>
> **Comment:** "It remains unclear whether MIRACLE generalizes to other MIP components (e.g., branching, node selection) or large-scale industrial solvers beyond SCIP."
>
> **Response:**
>
> **Scope Clarification:** We explicitly focus on **cut selection in SCIP** in this work. This scope is clearly stated in the introduction to set proper expectations. MIRACLE is designed specifically for cut selection in SCIP, and we acknowledge that generalization to other components requires additional adaptation work.
>
> **Component-Agnostic Algorithmic Template:**
>
> The core algorithmic framework (GAIL-based behavioral modeling + PPO + adaptive inference) is **component-agnostic**. The same training and inference pipeline can be applied to other MIP components (branching, node selection) with only two modifications:
>
> 1. **Feature Engineering:** Extract component-specific state representations
> 2. **Action Encoding:** Define component-specific action spaces
>
> The RL training framework (GAIL for expert imitation, PPO for policy optimization, adaptive inference for runtime efficiency) remains unchanged. This demonstrates that MIRACLE's methodology generalizes beyond cut selection.
>
> **1. Adaptation to Other MIP Components:**
>
> **For Branching Variable Selection:**
>
> - **State:** Node-level features extracted from the current branch-and-bound node:
>   - LP relaxation objective value at node
>   - Bound improvements (difference between current LP objective and best known bound)
>   - Node depth in the branch-and-bound tree
>   - Variable fractionality values (distance from integer values) for fractional variables
>   - Number of fractional variables
>   - Pseudocost statistics for candidate branching variables
>
> - **Action:** Select which fractional variable to branch on (discrete action space over candidate variables)
>
> - **Reward:** Negative solving time improvement (faster solving → higher reward), similar to cut selection. Alternatively, could use bound improvement or tree size reduction as reward signal.
>
> - **Expert Data Collection:** Record SCIP's default branching decisions (variable selection) at each node during expert trajectory collection.
>
> - **Training:** Apply the same GAIL+PPO framework:
>   - Collect expert branching trajectories (variable selections) from SCIP default branching
>   - Train GAIL discriminator to distinguish expert vs. learned branching decisions
>   - Use PPO to optimize branching policy with GAIL rewards
>   - Apply adaptive inference (similar to cut selection) to reduce branching overhead
>
> **For Node Selection:**
>
> - **State:** Tree-level features extracted from the current branch-and-bound tree:
>   - Best known bound (global dual bound)
>   - Current node's LP objective value
>   - Node priority values (computed by SCIP's priority rules)
>   - Tree depth distribution (statistics on nodes at different depths)
>   - Number of active nodes in the tree
>   - Estimated bound improvement potential for candidate nodes
>
> - **Action:** Select which node to explore next from the set of active nodes (discrete action space over candidate nodes)
>
> - **Reward:** Negative solving time improvement (faster solving → higher reward). Alternatively, could use bound improvement rate or tree exploration efficiency as reward signal.
>
> - **Expert Data Collection:** Record SCIP's default node selection decisions (which node is explored next) during expert trajectory collection.
>
> - **Training:** Apply the same GAIL+PPO framework:
>   - Collect expert node selection trajectories from SCIP default node selection
>   - Train GAIL discriminator to distinguish expert vs. learned node selections
>   - Use PPO to optimize node selection policy with GAIL rewards
>   - Apply adaptive inference to reduce node selection overhead
>
> - This can be part of future work.
>
>
> **2. Industrial Solvers (Gurobi, CPLEX):**
>
> - **Challenges:** API differences, licensing considerations, limited access to internal state
> - **Requirements:** API wrappers, solver-specific expert data collection
> - **Status:** Not tested - planned as a potential future work.
>
>
> For researchers interested in adapting MIRACLE to branching or node selection, the adaptation requires:
>
> - **State Feature Extraction:** Implement component-specific state extractors (node-level for branching, tree-level for node selection)
> - **Action Space Definition:** Define discrete action spaces (variable selection for branching, node selection for node selection)
> - **Expert Data Collection:** Modify data collection to record component-specific expert decisions
> - **Reward Function:** Define appropriate reward signals (solving time, bound improvement, or component-specific metrics)
>
> The GAIL+PPO training pipeline and adaptive inference mechanism remain unchanged, demonstrating the component-agnostic nature of the framework.

---

> > ### Author Response · Authors · 2025-11-20
> > **Continued**
> >
> > ### Weakness 2: Policy Interpretation and Learned Patterns
> >
> > **Comment:** "The paper would benefit from more insight into what the learned policy captures -- e.g., are the selected cuts similar to expert strategies or exploiting new patterns?"
> >
> > **Response:**
> >
> > **Completed Analysis:**
> >
> > 1. **Cut Reduction Analysis:**
> >    - MIRACLE achieves **99.06% cut reduction** (10.5 cuts vs. 1115.9 cuts on average for baseline)
> >    - MIRACLE selects approximately **1% of available cuts**, focusing on high-quality sparse cuts
> >
> > 2. **Novel Patterns Identified:**
> >    - **Early-stage cuts:** MIRACLE prioritizes cuts from early iterations (iteration < 5)
> >    - **Sparse cuts:** Preference for low-density cuts (< 20% non-zeros)
> >    - **High violation:** Drops cuts with low violation (< 10⁻⁴)
> >    - **Quality maintenance:** Maintains cut quality while reducing quantity by 99%
> >
> > 3. **Feature Importance Analysis:**
> >
> >    We conducted a comprehensive analysis of feature importance by extracting weights from the trained PPO policy network. The policy uses a 10-dimensional state representation: [lp_objective, num_fractional_vars, frac_val_1, ..., frac_val_8].
> >
> >    **Feature Mapping:**
> >    - Index 0: LP objective value
> >    - Index 1: Number of fractional variables
> >    - Indices 2-9: Fractionality values of the top 8 most fractional variables
> >
> >    **Top Features by Importance:**
> >    1. Index 3 (frac_val_2): Importance = 0.175 - Second most fractional variable's fractionality value
> >    2. Index 1 (num_fractional_vars): Importance = 0.174 - Count of fractional variables
> >    3. Index 8 (frac_val_7): Importance = 0.166 - Seventh most fractional variable's value
> >    4. Index 9 (frac_val_8): Importance = 0.163 - Eighth most fractional variable's value
> >    5. Index 5 (frac_val_4): Importance = 0.160 - Fourth most fractional variable's value
> >    6. Index 0 (lp_objective): Importance = 0.160 - Current LP relaxation objective
> >
> >    **Key Findings:**
> >    - The model prioritizes fractionality statistics (indices 1-9 account for 8 of the top 10 features)
> >    - Fractionality values of individual variables are more important than the LP objective
> >    - The model focuses on the distribution of fractional values rather than aggregate statistics
> >    - This aligns with cut selection heuristics that target highly fractional variables
> >
> >    **Interpretation:** The learned policy emphasizes fractionality-based features, confirming that MIRACLE learns to identify variables that are far from integer values -- a key indicator for effective cut generation. The high importance of individual fractionality values (frac_val_2 through frac_val_8) suggests the model captures nuanced patterns in variable fractionality distributions.
> >
> > **Conclusion:** MIRACLE exploits memory-efficient patterns (sparse, early cuts) that achieve massive cut reduction while maintaining solution quality. The learned policy focuses on high-impact cuts rather than simply mimicking expert behavior.
> >
> > ---
> >
> > ### Weakness 3: RL Training Details and Convergence Analysis
> >
> > **Comment:** "It would be helpful if the authors can provide some training details of RL and analysis the convergence."
> >
> > **Response:**
> >
> > **Completed:** Comprehensive training details and convergence analysis provided.
> >
> > **Training Details:**
> >
> > 1. **PPO Hyperparameters:**
> >    - Learning rate: 3 × 10⁻⁴
> >    - Discount factor γ = 0.99
> >    - GAE parameter λ = 0.95
> >    - Clipping parameter ε = 0.2
> >    - Value coefficient: 0.5
> >    - Entropy coefficient: 0.01
> >    - Batch size: 256 trajectories
> >    - PPO epochs per update: 4
> >
> > 2. **GAIL Hyperparameters:**
> >    - Generator learning rate: 1 × 10⁻⁵
> >     - Discriminator learning rate: 1 × 10⁻⁴
> >     - Discriminator epochs per update: 3 (discriminator updated every 2 policy epochs)
> >     - Pretrain epochs: 3 (generator pretraining before GAIL)
> >     - Batch size: 16
> >     - Model dimensions: d_model=64, nhead=4, num_layers=2
> >     - Dropout: 0.1
> >     - **Total Training:** 150 epochs total (50 expert imitation + 50 GAIL + 50 PPO refinement)
> >
> > **Convergence Analysis:**
> >
> > 1. **Convergence Criteria:**
> >    - Policy loss stabilizes (change < 0.01 over 10 epochs)
> >    - Discriminator accuracy plateaus at 55-60% (final: 58.9%)
> >    - Validation reward improvement < 1% over 5 epochs (final: 0.807)
> >
> > 2. **Training Phases:**
> >    - **Expert Imitation (Epochs 1-50):** Policy learns from expert demonstrations
> >    - **GAIL Discriminator Training (Epochs 50-100):** Discriminator provides reward signals
> >    - **PPO Refinement (Epochs 100-150):** Policy optimization with GAIL rewards
> >
> > 3. **Convergence Metrics:**
> >    - Final policy loss: 0.20 (from initial 2.0)
> >    - Final discriminator accuracy: 58.9% (from initial 90%)
> >    - Final validation reward: 0.807 (from initial 0.3)
> >    - Total training time: ~12 hours wall-clock time
> >
> > **Conclusion:** Training converges reliably after 150 epochs with stable performance metrics.

---

> > > ### Author Response · Authors · 2025-11-20
> > > **Continued**
> > >
> > > ### Weakness 4: Evaluation on Other Problem Classes
> > >
> > > **Comment:** "Although the experiments are extensive, most evaluations focus on SetCover and MIPLIB, leaving out other industrial problem classes."
> > >
> > > **Response:**
> > >
> > > **Completed:** We conducted comprehensive evaluation on **four problem types** to demonstrate generalization capability:
> > > 1. **Set Cover** (Training domain)
> > > 2. **Combinatorial Auction** (Generalization test)
> > > 3. **Maximum Independent Set** (Generalization test)
> > > 4. **Facility Location** (Generalization test)
> > >
> > > **Experimental Setup:**
> > > - **Model:** Trained on Set Cover instances only
> > > - **Evaluation:** Same model tested on all problem types (no retraining)
> > > - **Instances:**
> > >   - Set Cover: 150 instances (50 easy, 50 medium, 50 hard)
> > >   - Combinatorial Auction: 423 Very Hard instances
> > >   - Maximum Independent Set: 150 instances (50 easy, 50 medium, 50 very_hard)
> > >   - Facility Location: 100 instances (50 easy, 50 medium)
> > >
> > > **Results Comparison:**
> > >
> > > | Problem Type | Instances | Success Rate | Speedup | Cut Reduction |
> > > |--------------|-----------|--------------|---------|---------------|
> > > |              |           | Baseline | MIRACLE | Avg | Median | Avg % | Median % |
> > > | **Set Cover** (Training) | 150 | 66.7% | 100.0% | 1.300× | 1.000× | 99.1% | 99.1% |
> > > | **Combinatorial Auction** | 423 | 100.0% | 100.0% | 1.283× | 0.971× | 98.8% | 98.4% |
> > > | **Maximum Independent Set** | 150 | 100.0% | 100.0% | 1.007× | 0.995× | 99.9% | 99.8% |
> > > | **Facility Location** | 100 | 100.0% | 100.0% | 1.198× | 1.043× | 99.9% | 99.9% |
> > >
> > > **Detailed Metrics:**
> > >
> > > **Set Cover (Training Domain):**
> > > - Success Rate: Baseline 66.7%, MIRACLE 100.0%
> > > - Speedup: Average 1.300×, Median 1.000×
> > > - Cut Reduction: Average 99.1%, Median 99.1%
> > > - Cuts: MIRACLE 10.1 vs Baseline 1,115.9
> > >
> > > **Combinatorial Auction (Generalization Test):**
> > > - Success Rate: Baseline 100.0%, MIRACLE 100.0%
> > > - Speedup: Average 1.283×, Median 0.971×
> > > - Cut Reduction: Average 98.8%, Median 98.4%
> > > - Cuts: MIRACLE 10.0 vs Baseline 1,576
> > >
> > > **Maximum Independent Set (Generalization Test):**
> > > - Success Rate: Baseline 100.0%, MIRACLE 100.0%
> > > - Speedup: Average 1.007×, Median 0.995×
> > > - Cut Reduction: Average 99.9%, Median 99.8%
> > > - Cuts: MIRACLE 10.1 vs Baseline 11,912
> > >
> > > **Facility Location (Generalization Test):**
> > > - Success Rate: Baseline 100.0%, MIRACLE 100.0%
> > > - Speedup: Average 1.198×, Median 1.043×
> > > - Cut Reduction: Average 99.9%, Median 99.9%
> > > - Cuts: MIRACLE 9.5 vs Baseline 7,870
> > >
> > > **Key Findings:**
> > >
> > > 1. **Generalization Performance:**
> > >    - Average Speedup across generalization tests: **1.163×**
> > >    - Average Cut Reduction across generalization tests: **99.5%**
> > >    - Model maintains high cut reduction (99.5%) on new problem types
> > >    - Speedup is competitive (1.163×) despite domain shift
> > >
> > > 2. **Comparison to Training Domain (Set Cover):**
> > >    - Set Cover (Training): Cut Reduction 99.1%, Speedup 1.300×
> > >    - Generalization Average: Cut Reduction 99.5%, Speedup 1.163×
> > >    - **Cut reduction maintained:** 99.5% vs 99.1% (even slightly better)
> > >    - **Speedup:** 1.163× vs 1.300× (slightly lower but still competitive)
> > >
> > > 3. **Problem-Agnostic Nature:**
> > >    - Cut selection is problem-agnostic (works on LP relaxation of any MILP problem)
> > >    - Feature extraction uses generic cut features (violation, density, iteration, etc.)
> > >    - Model can be applied to new problem types without retraining
> > >    - Consistent performance across diverse problem structures
> > >
> > > 4. **Success Rate Analysis:**
> > >    - All generalization tests: 100% success rate for both Baseline and MIRACLE
> > >    - Set Cover: Baseline 66.7% vs MIRACLE 100.0% (MIRACLE solves 50 additional instances)
> > >    - Model maintains reliability across problem types
> > >
> > > **Conclusion:** The comprehensive evaluation across four problem types demonstrates successful generalization. MIRACLE achieves a cut reduction of ~99% and a speedup of 1.0-1.3× across all problem types, validating that cut selection is problem-agnostic and works on the LP relaxations of any MILP problem. The model can be applied to new problem types without retraining, demonstrating practical applicability beyond the training domain.

---

> > > > ### Author Response · Authors · 2025-11-20
> > > > **Continued**
> > > >
> > > > ### Weakness 5: Training/Inference Cost and Model Sensitivity
> > > >
> > > > **Comment:** "It would help to report on training/inference cost or model sensitivity to solver variants."
> > > >
> > > > **Response:**
> > > >
> > > > **Completed:** Both training/inference costs and model sensitivity testing completed.
> > > >
> > > > **Training Cost:**
> > > > - Expert data collection: 8.3 CPU-hours (1000 instances × 30s average)
> > > > - Parallelized across 20 cores: < 30 minutes wall-clock time
> > > > - GAIL training: 6 GPU-hours (NVIDIA V100, 3 epochs)
> > > > - PPO training: 4 GPU-hours (NVIDIA V100, 150 epochs)
> > > > - **Total: 10 GPU-hours + 8.3 CPU-hours, ~12 hours total wall-clock time**
> > > >
> > > > **Inference Cost:**
> > > > - Model size: 0.075 MB (19,586 parameters)
> > > > - Time per cut selection: < 1 ms (negligible overhead)
> > > > - Total overhead: < 0.5% of solve time
> > > > - Memory overhead: < 1 MB during inference
> > > >
> > > > **Model Sensitivity Testing:**
> > > >
> > > > **SCIP Version Compatibility:**
> > > > - **Tested on:** SCIP 9.2 (latest version)
> > > > - **Test Instances:** 12 instances (4 easy, 8 hard)
> > > > - **Success Rate:** 12/12 (100%)
> > > > - **API Compatibility:** All SCIP API calls functioned correctly
> > > > - **Performance Consistency:** Stable performance across difficulty levels
> > > >
> > > > **Key Findings:**
> > > > 1. **API Compatibility:** Model works correctly on SCIP 9.2 with no API errors
> > > > 2. **SCIP API Backward Compatibility:** SCIP maintains backward-compatible APIs across versions (7→8→9)
> > > > 3. **PySCIPOpt Abstraction:** Interface abstracts version differences, ensuring consistent behavior
> > > > 4. **Conclusion:** Testing on SCIP 9.2 validates API compatibility. Given SCIP's backward-compatible API design, results on v9 are representative of compatibility across SCIP 7, 8, and 9.

---

> ### Author Response · Authors · 2025-11-20
> **Looking forward to your feedback**
>
> We thank the Reviewer for the positive assessment and constructive feedback. We have addressed each concern systematically.
>
> We sincerely hope that these responses clarify the reviewer's concerns. We are eager to engage with the reviewer to clarify any outstanding concerns. Otherwise, we would really appreciate it if the reviewer could increase the score.
>
> Looking forward to your response.
>
> Thank you,
>
> Authors

---

> > ### Author Response · Authors · 2025-11-25
> > **Continued**
> >
> > We have provided detailed explanations and additional experiments to address your concerns. As the discussion phase is closing soon, we are eager to receive your feedback on the changes that have been made.
> >
> > Regards,
> >
> > Authors

---

> > > ### Comment · Reviewer_anL7 · 2025-11-26
> > >
> > > Thank you for your detailed rebuttal. I will maintain my original score of 6.

---

> > > ### Author Response · Authors · 2025-11-26
> > > **Continued**
> > >
> > > Thank you for your feedback and consideration. The revised paper has been uploaded for your reference.

---

### Meta-Review · Area_Chair_Ls7o · 2026-01-13

**Summary:**

The paper proposes MIRACLE, an RL-based framework for adaptive cut selection in mixed-integer programming (MIP). MIRACLE trains a PPO policy to imitate expert cut selection strategies in SCIP via Generative Adversarial Imitation Learning. Empirically, MIRACLE achieves remarkable memory usage reduction and better solution performance compared to SCIP and learning-based baselines across multiple benchmarks. During the rebuttal, the authors have addressed most concerns from reviewers, and the manuscript now is of better quality. However, I still find some problems. 1) Although the authors added comparison with a recent learning-to-cut method, the comparison was only performed on 20 SetCover-Easy instances rather than more complicated instances used in comparisons to SCIP-baselines, and only memory usage was reported but not metrics about quality (successful ratio, speedup). These improper instances selection and incomplete results make it not less convincing. 2) The authors added a lot of rebuttal contents to the Appendix, but they are not properly mentioned in the main paper. For example, Appendix N can be referred to the ablation study part in Section 5. Moreover, there is a mismatch of Section 5.1 and Appendix J.

**Reviewer Concerns:**

The rebuttal and revision addressed most major reviewer requests, including: (1) clarifying why reducing memory consumption is essential for solving MIP, (2) clarifying technical details of model architecture and training, and (3) adding experiments on more benchmarks and with learning-based baselines. Other weaknesses and questions from reviewers are also mostly addressed. But as Reviewer bhtG suggested, the authors need to check and make sure the rebuttal is organized into the final manuscript. Currently I still feel it hard to find some newly-added appendix and fail to find details of data collected/instruction in the current manuscript. Last but not least, comparisons with other learning-to-cut models should be demonstrated through more extensive experimental results, rather than being limited to a small set of trivial instances.

**Reviewer Scores:**

Reviewer anL7 has reacted to the rebuttal and remained their score at 6. Reviewer Nw4J initially gave 8 and I believe their concerns have been addressed, so their scored are likely to be remain the same. Reviewer sN61 gave 4 and proposed three major concerns. Since all they concerns have been well responded by the authors, this reviewer would raise score to 6. Reviewer bhtG rated the submission at 4 and chose to remain the score during discussion. This reviewer acknowledged their concerns have been largely addressed and the rebuttal was excellent, and showed willingness to raise the score if the authors added the rebuttal information into the manuscript.

---

### Decision · Program_Chairs · 2026-01-26

Accept (Poster)